# Scalable and multiplexed recorders of gene regulation dynamics across weeks

Lirong Zheng[1,2,14], Dongqing Shi[1,2,14], Yixiao Yan[1,2,14], Bingxin Zhou[3], Jormay Lim[1,2], Yongjie Hou[1,2], Bobae An[4], Jason K. Adhinarta[5], Michael Lin[5], BumJin Ko[6], William C. Joesten[1,2], Mehul Gautam[1,2], Elie D. M. Huez[7], Eung Chang Kim[4], Emily G. Klyder[8], Boxuan Chang[9], Sethuramasundaram Pitchiaya[10,11], Michael T. Roberts[7], Denise J. Cai[6], Edward S. Boyden[4,12], Donglai Wei[5], Pietro Liò[3] & Changyang Linghu[1,2,13 ✉]

Gene expression is dynamically controlled by gene regulatory networks comprising multiple regulatory components to mediate cellular functions[1]. An ideal tool for analysing these processes would track multi-component dynamics with both spatiotemporal resolution and scalability within the same cells, a capability not yet achieved. Here we present CytoTape, a genetically encoded, physiologically compatible, modular protein tape recorder for multiplexed and spatiotemporally scalable recording of gene regulation dynamics continuously for up to 3 weeks, with single-cell, up to minutes-scale resolution. CytoTape uses a flexible, thread-like, elongating intracellular protein self-assembly engineered via computationally assisted rational design, built on our earlier XRI technology[2]. We demonstrate its utility across multiple mammalian cell types, achieving simultaneous recording of five transcription factor activities and gene transcriptional activities. CytoTape reveals that divergent transcriptional trajectories correlate with transcriptional history and signal integration, and that distinct immediate early genes (IEGs) exhibit complex temporal correlations within single cells. We further extended CytoTape into CytoTape-vivo for scalable, spatiotemporally resolved single-cell recording in the living brain, enabling simultaneous weeks-long recording of doxycycline-dependent and IEG promoter-dependent gene expression histories across up to 14,123 neurons spanning multiple brain regions per mouse. Together, the CytoTape toolkit establishes a versatile platform for scalable and multiplexed analysis of cell physiological processes in vitro and in vivo.

Gene regulatory networks (GRNs) integrate intrinsic programs and extrinsic signals to establish cellular identity, govern cellular function and dynamic responses, and when dysregulated, contribute to disease[1]. These networks comprise dynamically interacting components that collectively produce complex regulatory behaviours. Such dynamic interactions remain inaccessible to static snapshots or time-resolved measurements of individual network components. Recent studies have suggested that GRNs exhibit complex dynamical rules, including intricate activation–inhibition patterns and temporal sequences[3]. Dissecting GRN principles requires a technology that provides simultaneous, time-resolved recording of multiple interacting components within the same cell. Furthermore, GRNs often drive behaviours and disease processes that unfold over days to weeks in vivo[1], necessitating long-term, tissue-scale, single-cell measurements.

Existing technologies fall short of meeting the combined demands of days-to-weeks-long, multiplexed and scalable single-cell recording. Genetically encoded fluorescent reporters allow real-time imaging and multiplexing via spectral[4], spatial[5] and temporal[6] schemes, but require continuous microscopy access and stability, and are constrained in spatiotemporal scale by limited three-dimensional (3D) fields of view, tissue absorption and scattering of light, and photobleaching. Nucleic acid-based recorders encode intracellular events into nucleotide sequences for subsequent readout[7–14], but do not yet support single-cell multiplexed, continuous temporal recording and typically require cell lysis and/or tissue dissociation that may compromise spatial and molecular information in intact tissues. Post-mortem methods provide multiplexed spatial analysis of intact tissues but lack temporal resolution[15]. The recently proposed 'protein tape recorder' concept (XRI[2] and

[1]Department of Cell and Developmental Biology, Medical School, University of Michigan, Ann Arbor, MI, USA. [2]Michigan Neuroscience Institute, University of Michigan, Ann Arbor, MI, USA. [3]Department of Computer Science and Technology, University of Cambridge, Cambridge, UK. [4]McGovern Institute for Brain Research, Massachusetts Institute of Technology, Cambridge, MA, USA. [5]Department of Computer Science, Boston College, Boston, MA, USA. [6]Nash Department of Neuroscience, Icahn School of Medicine at Mount Sinai, New York, NY, USA. [7]Kresge Hearing Research Institute and Department of Otolaryngology-Head and Neck Surgery, University of Michigan, Ann Arbor, MI, USA. [8]Department of Chemistry, University of Michigan, Ann Arbor, MI, USA. [9]Department of Electrical Engineering and Computer Science, University of Michigan, Ann Arbor, MI, USA. [10]Department of Urology, University of Michigan, Ann Arbor, MI, USA. [11]Department of Pathology, University of Michigan, Ann Arbor, MI, USA. [12]Yang Tan Collective and Howard Hughes Medical Institute, Massachusetts Institute of Technology, Cambridge, MA, USA. [13]Department of Biomedical Engineering, University of Michigan, Ann Arbor, MI, USA. [14]These authors contributed equally: Lirong Zheng, Dongqing Shi, Yixiao Yan. ✉e-mail: linghu@umich.edu

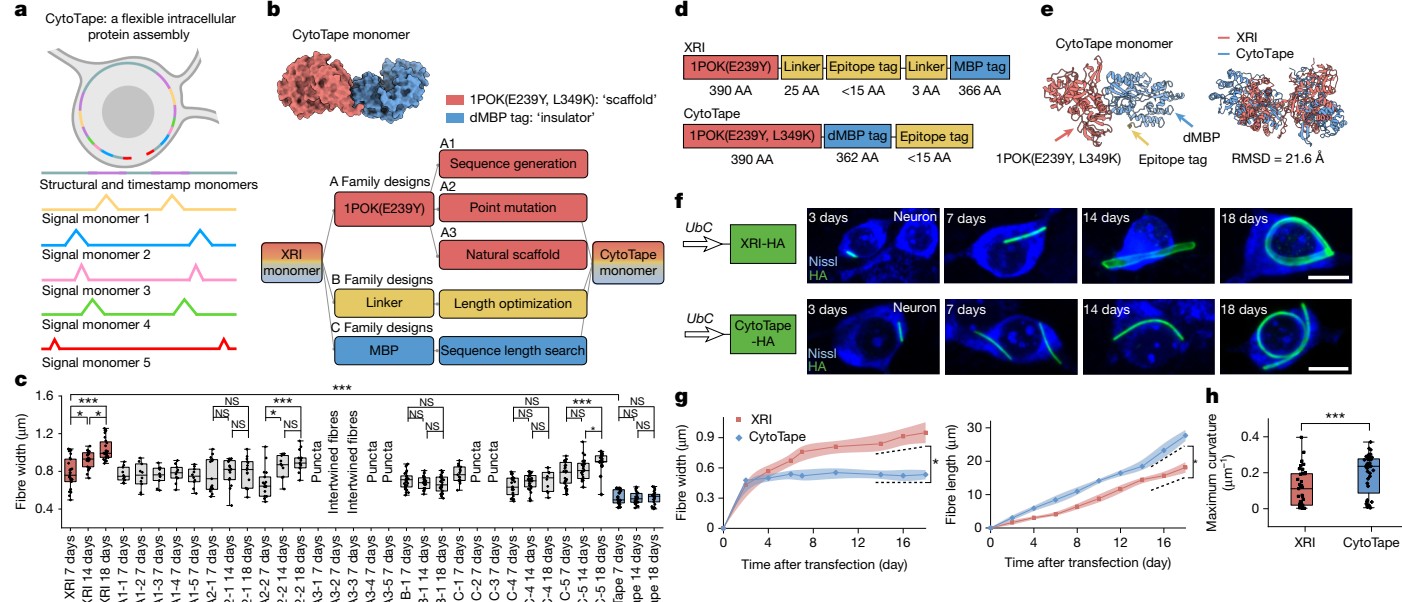

**Fig. 1 | Design of CytoTape. a**, Schematic of the CytoTape system. Schematic created in BioRender; Zheng, L. https://biorender.com/3uq51z7 (2026). **b**, AlphaFold3-predicted structure of the CytoTape monomer (top), and the design strategy using the XRI monomer as a template (bottom). **c**, Comparison of fibre assembly width among protein designs across time in cultured neurons. Sample size ($n$) denotes $n = X$ assemblies from $Y$ neurons from $Z$ cultures. XRI: 7 days (31, 25 and 5), 14 days (26, 20 and 5) and 18 days (32, 30 and 5). A1-1: 7 days (10, 5 and 1). A1-2: 7 days (10, 6 and 1). A1-3: 7 days (10, 7 and 1). A1-4: 7 days (10, 6 and 1). A1-5: 7 days (10, 7 and 1). A2-1: 7 days (15, 10 and 2), 14 days (15, 15 and 2) and 18 days (15, 12 and 3). A2-2: 7 days (17, 10 and 2), 14 days (10, 10 and 2) and 18 days (15, 15 and 2). B-1: 7 days (31, 28 and 4), 14 days (25, 20 and 6) and 18 days (25, 25 and 3). C-1: 7 days (15, 12 and 3). C-4: 7 days (28, 20 and 3), 14 days (30, 30 and 4) and 18 days (10, 10 and 2). C-5: 7 days (31, 30 and 2), 14 days (24, 20 and 2) and 18 days (18, 15 and 3). CytoTape: 7 days (16, 16, 3), 14 days (28, 20 and 5) and 18 days (25, 20 and 5). NS, not significant, *$P < 0.05$ and ***$P < 0.001$; for each design across time, two-sided Kruskal–Wallis analysis with Dunn's post-hoc tests were used; between XRI and CytoTape, a two-sided Mann–Whitney $U$-test was used. **d**, Schematic constructs of XRI (top) and CytoTape (bottom) monomers. AA, amino acids. **e**, AlphaFold3-predicted structure of the CytoTape monomer (left), and superposition of XRI (red) and CytoTape (blue) monomers (right). **f**, Confocal images of cultured neurons expressing HA-tagged XRI (top) and CytoTape (bottom) across time. Scale bars, 10 µm. **g**, Kinetics of protein assembly width (left) and length (right) for XRI and CytoTape in cultured neurons ($n = 21$ XRIs or CytoTapes from 20 neurons, 4 cultures). The dashed line denotes the slope, the thick centre line indicates the mean and the boundary shows the s.d. *$P < 0.05$, two-sided Mann–Whitney $U$-test. **h**, Maximum curvature of XRI and CytoTape fibres in cultured neurons on day 18 ($n = 21$ XRIs or CytoTapes from 20 neurons, 4 cultures). ***$P < 0.001$, two-sided Mann–Whitney $U$-test. For the boxplots, the middle line indicates the median, the box boundary denotes the interquartile range, the whiskers show the minimum and maximum, and the black dots denote the individual data points (**c**,**h**).

iPAK4 (ref. 16)) allows recording of cellular activities along intracellular linear protein self-assemblies for post-fixation scalable readout. However, these protein tape recorders do not support simultaneous recording of multiple cellular signals or time-resolved recording over weeks because long-term growth of linear protein assemblies is constrained by cell size. Thus, a scalable technology enabling long-term, multiplexed measurements of cellular activities with spatiotemporal resolution remains critically needed.

Here we introduce CytoTape, an intracellular protein tape recorder for scalable and multiplexed continuous recording with minute-scale temporal precision and weeks-long duration, built on our earlier XRI system[2]. CytoTape uses flexible, thread-like, physiologically compatible protein self-assembly that can grow longer than the cell size with multiple molecular tags, each encoding a distinct cellular signal, to achieve long-term, multiplexed recording. CytoTape achieves simultaneous, spatiotemporally scalable recording of five plasticity-associated transcription factor and IEG transcriptional activities, including phosphorylated cAMP response element-binding protein (pCREB), *Fos*, *Arc*, *Egr1* (also known as *Zif268*) and NPAS4, within the same assemblies in individual live cells. Using CytoTape, we observed that CREB activation and downstream FOS expression can temporally decouple in cells exhibiting active FOS regulation before CREB activation, with more cells showing decoupling upon MEK–ERK inhibition, highlighting the roles of transcriptional memory and signal integration in shaping cellular responses. In cultured neurons, *Arc*-promoter-driven and *Egr1*-promoter-driven expressions display complex, multi-peak activation following a single stimulation, revealing distinct stimulus-dependent temporal correlations. To enable this capability in vivo, we optimized CytoTape into CytoTape-vivo for spatiotemporally resolved, scalable single-cell recording in the living brain, achieving weeks-long, simultaneous recording of doxycycline (Dox)-dependent and *Fos*-promoter-dependent gene expression across up to 14,123 hippocampal and cortical neurons per mouse during seizure, demonstrating its utility in vivo.

## Design of CytoTape

Recording more kinds of biological events over extended durations requires protein tape recorders with larger information storage capacity. Because information is encoded in molecular tags along the protein assembly, the storage capacity is in principle proportional to the product between the maximum number of tags along the recording axis (that is, maximum axial length) and the logarithm of the number of tag variants (that is, alphabet size). We first worked towards increasing the maximum axial length of the protein assembly in live cells. As rigid assemblies either distort cell membrane when exceeding cell size, risking physiological perturbations[17] or halt growth at membrane contact, we reasoned that a flexible, thread-like assembly could elongate beyond the cell size without perturbing cellular integrity, providing a suitable molecular substrate for long-term recording (Fig. 1a). Minimizing

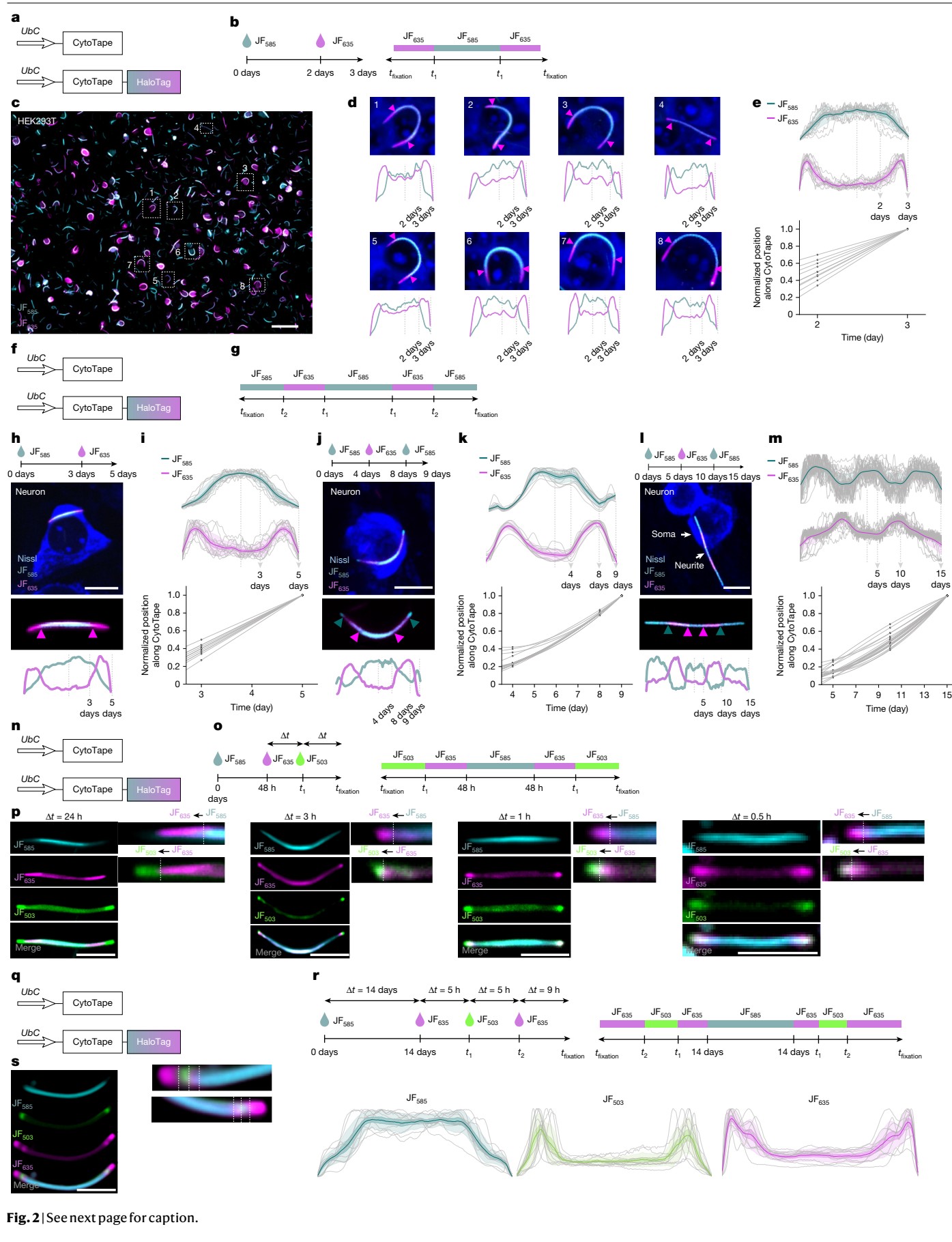

**Fig. 2** | See next page for caption.

**Fig. 2 | CytoTape resolves and encodes timestamps with minutes-to-days intervals in single cells. a–e**, Characterization of timestamps of CytoTape in HEK cells. **f–m**, Characterization of timestamps of CytoTape in cultured neurons. **q–s**, Characterization of temporal resolution of CytoTape for long-term recording in cultured neurons. **a,f,n,q**, Schematics of constructs transfected into HEK cells (**a,n**) and cultured neurons (**f,q**). **b,g,o,r**, Time points of JF dyes addition (left) and the expected dye distributions along the protein fibre (right). **c**, Low-magnification image of CytoTape labelled with JF dyes in HEK cells; the timestamp transitions appear less distinct in some fibres owing to intercellular variations in optimal colour-channel contrasts. Scale bar, 100 μm. **d**, Enlarged confocal images of HEK cells indicated in panel **c**. Fluorescence line profiles of JF dyes are shown below each image. **e**, Population analysis (top) of fluorescence line profiles from the experiments described in panel **b** (*n* = 10 CytoTapes from 10 HEK cells, 2 cultures) and interpolation between the spatial axis along the fibre and the time axis (bottom). **h,j,l**, Timestamps for different

temporal scales and resolutions in cultured neurons. Scale bars, 10 μm. **i,k,m**, Population analysis of fluorescence line profiles from the experiments described in panel **h** (*n* = 8 CytoTapes from 8 neurons, 2 cultures (**i**); *n* = 10 CytoTapes from 10 neurons, 2 cultures (**k**) and *n* = 18 CytoTapes from 18 neurons, 2 cultures (**m**); top). Interpolation between the spatial axis along the fibre and the time axis is also shown (bottom). **p**, Confocal images of timestamped fibres labelled with different dye-switching intervals in HEK cells. Scale bars, 5 μm. **s**, Confocal images of a timestamped fibre in a cultured neuron from the dye-switching experiment described in panel **r** (left). Scale bar, 5 μm. Population analysis of fluorescence line profiles (*n* = 10 CytoTapes from 10 neurons, 2 cultures) is also shown (right). The arrowheads in panels **d,h,j,l** and the dashed lines in panels **p,s** indicate the transient dye switching. In panels **e,i,k,m,s**, the thick centre line denotes the mean, the darker boundary shows the s.e.m., the lighter boundary indicates the s.d., and the light grey thin lines represent data from individual CytoTapes.

---

lateral growth perpendicular to the elongation axis is also critical, as extensive uncontrolled lateral growth disrupts temporal order along the axis[2]. We hypothesized that reducing the thickness of XRI assembly, perhaps by minimizing lateral monomer interactions, would enhance its mechanical flexibility for long-term temporal recording.

We applied a design and screening workflow with combined computational and rational protein engineering for new monomers, independently optimizing the scaffold (1POK(E239Y))[18], the linker (with a haemagglutinin (HA) epitope tag) and the insulator (MBP) domains in the XRI monomer (Fig. 1b). For the scaffold domain, we used three strategies: (1) sequence generation via CPDiffusion[19], yielding five variants (A1 family designs, A1-1 to A1-5) with 70–80% identity to the original 1POK(E239Y) sequence; (2) single-site mutagenesis (A2 family designs, A2-1 and A2-2) guided by ProtSSN[20], identifying two top mutations (E149I and L349K); and (3) replacement with natural self-assembling proteins (A3 family designs, A3-1 to A3-5; Fig. 1c and Supplementary Tables 1 and 8). As control, the XRI assembly maintained fibre-like morphology but progressively thickened over 18 days in primary cultured mouse hippocampal neurons, indicating lateral monomer incorporation (Fig. 1c and Supplementary Fig. 2). CPDiffusion variants formed short, low-aspect-ratio fibres unsuitable for long-term recording (Fig. 1c and Supplementary Fig. 3). Among ProtSSN variants, L349K produced thinner fibres and reduced lateral growth, whereas E149I did not (Fig. 1c). Natural scaffolds failed to form regular fibres (Fig. 1c and Supplementary Fig. 3). For the linker domain, we first relocated the HA tag to the C terminus and then shortened the linker length to two amino acids (B family design, B-1), which yielded thinner fibres with suppressed lateral growth (Fig. 1c and Supplementary Fig. 3). For the insulator domain, we conducted homology-based sequence searches (BLAST) to identify MBP homologues with different radii of gyration ($R_g$) and also performed truncation and duplication of the original MBP (C family designs, C-1 to C-5). Removing the first four unstructured residues (dMBP, C-4) produced thinner fibres than B-1 and suppressed lateral growth (Fig. 1c and Supplementary Figs. 3 and 4). Combining all effective elements, including L349K mutation, linker removal, C-terminal tag relocation and dMBP, produced a monomer design that robustly forms thin, stable fibres while maintaining the same thickness over weeks, termed CytoTape (Fig. 1d). Structural modelling (Fig. 1e, Extended Data Fig. 1a and Supplementary Fig. 5) and molecular dynamics simulations (Extended Data Fig. 1b,c) revealed that CytoTape monomers exhibit markedly increased conformational stability compared with XRI monomers, which may suppress non-specific lateral monomer interactions.

## Characterization across cell types

We characterized CytoTape structure and kinetics in cultured neurons, HEK and HeLa cells, in comparison with XRI and iPAK4. In neurons, iPAK4 forms rigid, crystalline fibres[16] that induced cell membrane distortion

within 3 days (Supplementary Fig. 6a), suggesting its limited capability to support multi-week intracellular recording. XRI fibres bent when longer than the soma size without distorting cell morphology but progressively thickened after 7 days (Fig. 1c,f). By contrast, CytoTape fibres remained thin after 18 days and exhibited substantial, thread-like bending without altering cell morphology (Fig. 1f and Supplementary Fig. 7). CytoTape exhibited self-limiting lateral growth, faster elongation and greater flexibility than XRI (Fig. 1g,h). In addition, over 95% of CytoTape-positive neurons successfully formed fibres in the somata, and over 80% of all CytoTape fibres localized exclusively in the somata, with the remaining fibres extending into neurites (Supplementary Fig. 6f,g).

To test the recording capability of CytoTape over multi-week timescales, we co-expressed a V5-tagged CytoTape monomer under the *Egr1* promoter[21] (activity-dependent promoter-driven monomer is referred to as the 'signal monomer') together with a human ubiquitin C (*UbC*) promoter-driven HA-tagged monomer ('structural monomer') in cultured neurons. After 15 days of expression, CytoTape reported clear KCl-induced V5 signal peaks along the fibre, whereas XRI did not (Extended Data Fig. 1d and Supplementary Fig. 8). When neurons received two KCl stimulations 12 h apart, CytoTape captured both events as distinct peaks along the fibre (Extended Data Fig. 1e), demonstrating hours-scale temporal resolution for sequential transcriptional events.

We next evaluated CytoTape performance in HEK and HeLa cells. In both cell types, CytoTape produced thin, flexible and well-structured fibres without altering cellular morphology (Extended Data Fig. 1f and Supplementary Figs. 7 and 9b). However, XRI formed thick fibres and puncta in HEK cells and intertwined fibre structures in HeLa cells (Extended Data Fig. 1f), whereas iPAK4 produced rigid fibres that caused noticeable distortions in cell morphology[2,16] (Supplementary Figs. 6a and 9a). By contrast, over 90% of CytoTape-positive HEK and HeLa cells contained fibres rather than puncta (Supplementary Fig. 6h,i), indicating structural stability and cross-cell-type compatibility that may stem from the low free-energy landscape of CytoTape monomer (Extended Data Fig. 1b,c). We also confirmed that CytoTape can record the heat shock-responsive human *HSPA1A* promoter activity[22] in HeLa cells under heat shock (Supplementary Fig. 6c). Furthermore, CytoTape does not alter the electrophysiological integrity, synaptic transmission, and calcium and network dynamics of cultured neuron populations (Extended Data Fig. 2) or the viability, proliferation, physiological state, transcriptional activity or signalling activity of HEK cells (Extended Data Fig. 3 and Supplementary Fig. 6b). Collectively, these results establish CytoTape as a versatile long-term recording platform compatible across cell types while preserving normal cellular physiology.

## Time axis recovery from timestamps

Precise encoding of time into space is critical for recovering the time axis from protein tape recorders. In XRI, a chemically inducible Cre

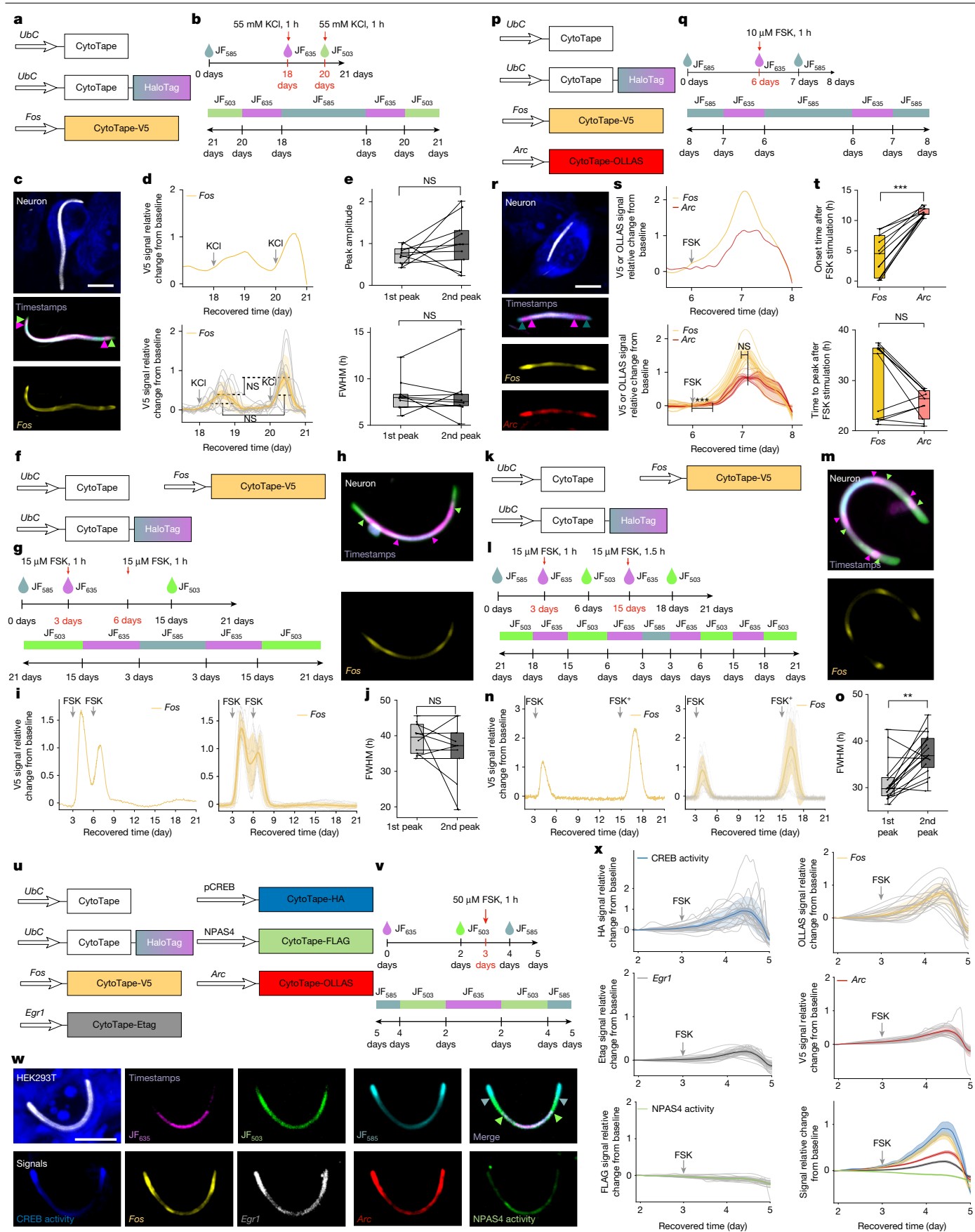

**Fig. 3 |** See next page for caption.

**Fig. 3 | CytoTape enables multiplexed, weeks-long recording of gene regulation dynamics. a–e**, Weeks-long recording in cultured neurons with two sequential KCl stimulations. **f–j**, Weeks-long recording in cultured neurons with two sequential FSK stimulations separated by 3 days. **k–o**, Weeks-long recording in cultured neurons with two sequential FSK stimulations separated by 12 days. **p–t**, Multiplexed recording in cultured neurons. **u–x**, Multiplexed recording in HEK cells. **a,f,p,k,u**, Schematics of constructs transfected into cultured neurons (**a,f,k**) and HEK cells (**u**). **b,g,l,q,v**, Experimental timelines (top) and expected dye distributions along CytoTape (bottom). **c**, Images of cultured neurons expressing the constructs shown in panel **a**. Scale bar, 10 μm. **d**, *Fos* signal (from panel **c**) relative change from baseline plotted against recovered time (top). The population analysis (*n* = 10 CytoTapes from 10 neurons, 2 cultures) is also shown (bottom). **e**, Comparison of peak amplitudes (top) and widths (FWHM; bottom; from panel **d**) between sequential V5 signals. **h–j,m–o**, Three-week-long continuous analogue recording of *Fos* signal in cultured neurons. Images of cultured neurons expressing constructs shown in panel **f** (**h**). *Fos* signal relative change from baseline plotted against recovered time (left), and population analysis (*n* = 10 CytoTapes from 10 neurons, 5 cultures; right; **i**). Comparison of FWHM (from panel **i**) between sequential V5 signals (**j**). Images of cultured neurons expressing the constructs shown in panel **k** (**m**). *Fos* signal

relative change from baseline plotted against recovered time (left), and population analysis (*n* = 16 CytoTapes from 16 neurons, 5 cultures; right; **n**). Comparison of FWHM (from panel **n**) between sequential V5 signals (**o**). **r**, Images of neurons expressing the constructs shown in panel **p**. Scale bar, 10 μm. **s**, *Fos* and *Arc* signal (from panel **r**) relative change from baseline plotted against recovered time (top). Population analysis (*n* = 10 CytoTapes from 10 neurons, 2 cultures) is also shown (bottom). **t**, Comparison of onset time (top) and time to peak (bottom; from panel **s**) between *Fos* and *Arc* signals. **w**, Images of a representative HEK cell expressing the constructs shown in panel **u**. Scale bar, 10 μm. **x**, Signal relative change from baseline plotted against recovered time (*n* = 20 CytoTapes from 20 HEK cells, 2 cultures). For the population analysis in panels **d,i,n,s,x**, the thick centre line indicates the mean, the darker boundary denotes the s.e.m., the lighter boundary represents the s.d., and the light thin lines show data from individual CytoTapes. For the boxplots in panels **e,j,o,t**, the middle line denotes the median, the box boundary shows the interquartile range, the whiskers indicate the minimum and maximum, and the black dots represent the individual data points. Throughout this figure, **\*\****P* < 0.01 and \*\*\**P* < 0.001; two-sided Wilcoxon signed-rank test (no multiple-comparison adjustment for distinct waveform features). The arrowheads in panels **c,h,m,r,w** indicate the transient dye switching.

---

system enabled global time calibration across cells. However, a single global time axis cannot account for cell-to-cell variabilities in nucleation onset and elongation kinetics of protein assembly (Fig. 1g and Supplementary Fig. 6d,e), limiting the precision of space-to-time conversion. The HaloTag system has been shown to provide hour-scale timestamp precision along the iPAK4 assembly over 22 h via HaloTag ligand dye labellings[16], and we therefore tested whether the HaloTag system supports precise timestamps along CytoTape assembly in the long-term recording context.

We first tested such HaloTag-based timestamps in HEK and HeLa cells, creating a 'timestamp monomer' by fusing HaloTag to the C terminus of the structural monomer (Fig. 2a). In HEK cells, the switching event between Janelia Fluor 585 ($JF_{585}$) and $JF_{635}$ dyes (Fig. 2b) was clearly recorded as a timestamp along individual fibres (Fig. 2c and Supplementary Fig. 10), producing a decline in $JF_{585}$ intensity and a simultaneous rise in $JF_{635}$ intensity upon dye switch (Fig. 2d). Cell-to-cell variation in timestamp positions along fibres highlights fibre elongation variability and the benefit of single-fibre-level timestamps (Fig. 2e). To convert discrete timestamps along each fibre from dye-switching events, together with fibre termini as the final timestamp upon cell fixation, into a continuous time axis, we performed spatiotemporal interpolation along fibres under one dye-switching event (Fig. 2e) and two dye-switching events, which provide a more precise time axis, in HeLa (Supplementary Fig. 11a–d) and HEK cells (Supplementary Fig. 11e,f). To assess temporal scalability, we further validated time axis recovery in cultured neurons under three distinct timestamping schedules spanning 5, 9 and 15 days with one and two dye-switching events (Fig. 2f–m). These results demonstrate that, via timestamps, users can flexibly define the temporal precision and scale of the recovered time axis based on experimental needs. In addition, over 90% of CytoTape fibres in cultured neurons and HEK cells, and over 80% in HeLa cells, successfully incorporated timestamps (Supplementary Fig. 11g). By contrast, XRI cannot robustly resolve dye-switching events in neurons over 15 days and in HEK cells over 3 days (Supplementary Fig. 12). We further characterized the intrinsic temporal resolution of CytoTape, showing that it resolves dye-switching events separated by 30 min in HEK cells (Fig. 2n–p). Moreover, CytoTape provides hours-scale resolution while maintaining weeks-long elongation in cultured neurons (Fig. 2q–s).

### Development of transcriptional recorders

We tested CytoTape with IEG and IEG-dependent promoters for *Fos*[23], *Egr1* (ref. 21), *Arc*[24] and NPAS4 (ref. 25) that are widely used to link reporter expression to IEG activities, as IEGs are key regulators

of many cellular processes[26], as well as the pCREB-dependent promoter[27], because CREB activity regulates IEG expression and other cellular processes[28] (transcription factor activities measured using transcription factor-responsive promoters are written in non-italic font, whereas gene transcriptional activities measured via their respective gene promoters are italicized to distinguish between the two). We co-expressed the V5-tagged signal monomer driven by an IEG-dependent or pCREB-dependent promoter and the structural monomer in cultured neurons (Extended Data Fig. 4a), which were stimulated by KCl, to induce depolarization and activate IEG activity[29], or forskolin (FSK), to raise intracellular cAMP level and induce CREB activity[5] (Extended Data Fig. 4b,c). CytoTape recorded peak-like V5 intensity profiles symmetrically located on both sides of the fibre midpoint in stimulated neurons (Extended Data Fig. 4d–m), but not in unstimulated neurons (Supplementary Fig. 14), in agreement with results from conventional GFP reporter assay under live-cell timelapse imaging[2] (Supplementary Fig. 13). Under identical stimulation, many neurons displayed a single-peak waveform, whereas other neurons exhibited more complex waveforms (Extended Data Fig. 4d–m), highlighting temporal heterogeneity of transcriptional activities across cells[30]. Single-fibre recordings closely mirrored each other when a neuron contains multiple fibres (Supplementary Fig. 14n). After titrating doses and durations of stimulations, we observed that stronger and longer stimulation produced a higher and steeper peak (or peaks) in V5 signal waveforms (Extended Data Fig. 4n–r), demonstrating CytoTape as an analogue recorder capturing both the timing and amplitude of transcriptional events. CytoTape also captured IEG transcriptional activities in primary hippocampal glial cells co-cultured with neurons, showing its potential for mapping coordinated gene regulation dynamics between neuronal and glial cell populations (Supplementary Fig. 15).

### Multiplexed and multi-week recording

To test whether CytoTape could jointly record cellular activity and time information, we co-expressed structural, timestamp and signal monomers in HEK cells and cultured neurons, which produced fibres with clear dye switching-induced timestamps and stimulus-induced transcriptional signals, allowing alignment of cellular activity to the recovered time axis (Supplementary Figs. 16 and 17), even after 21 days of expression in neurons (Fig. 3a–e). After 21 days, CytoTape formed long, flexible fibres (Fig. 3c), and reconstruction of the time axis revealed two distinct peaks corresponding to two sequential, identical KCl stimulations, with the V5 signal onset occurring after each stimulus (Fig. 3d), with similar peak amplitudes (Fig. 3e) and full width

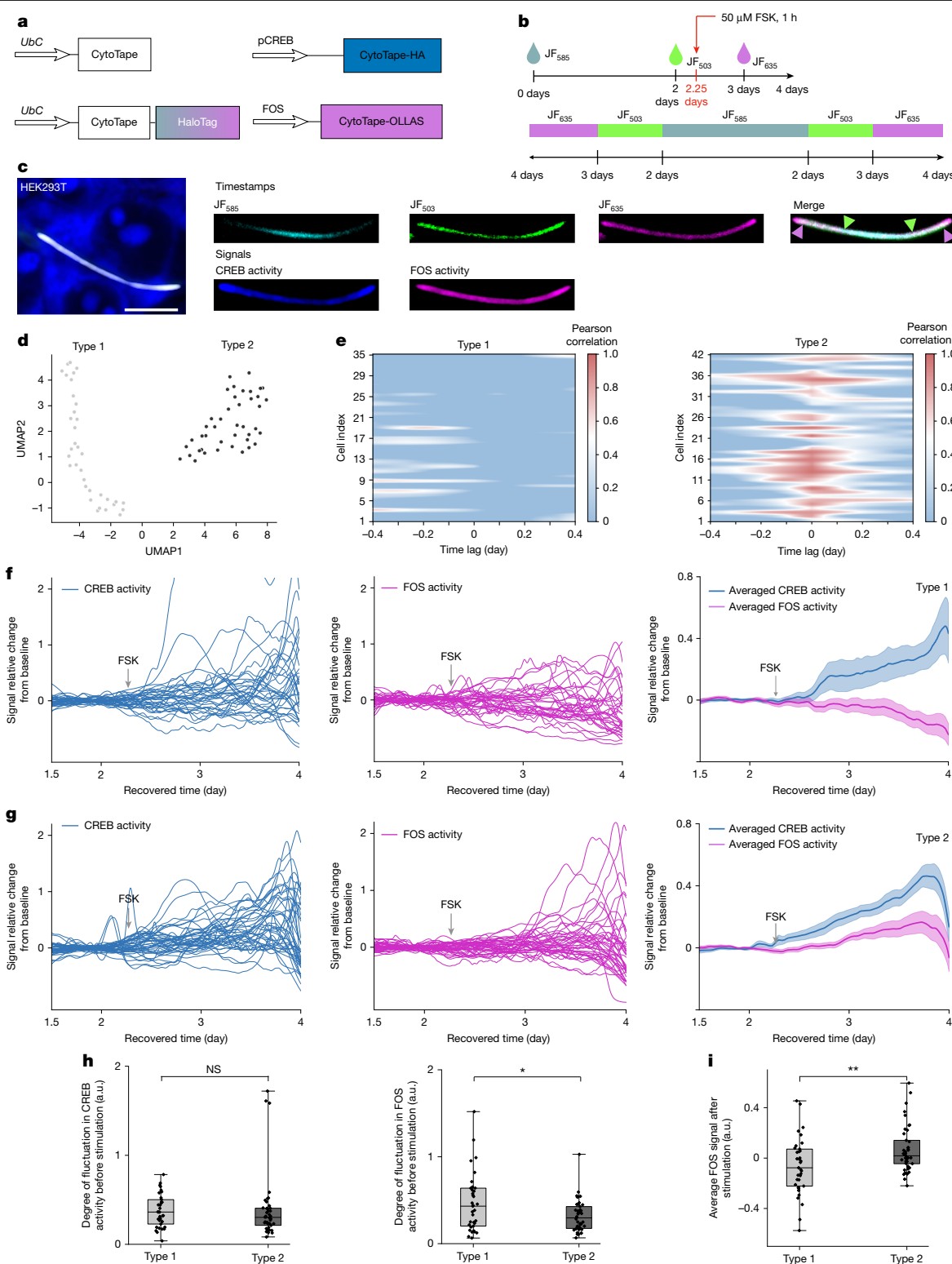

**Fig. 4** | See next page for caption.

at half maximum (FWHM) of V5 signal peaks (Fig. 3e) between the two stimulation events. FWHM was on the order of several hours, indicating that CytoTape can resolve hours-scale features of transcriptional activity. Under identical 15 μM FSK for 1-h stimulations on both days 3 and 5, representing temporally close events during the same 21-day period, CytoTape resolved both events with similar temporal features in cultured neurons (Fig. 3f–j). Under 15 μM FSK for 1 h on day 3 and for 1.5 h on day 15, representing temporally distant, non-identical events,

CytoTape resolved both events and reported distinct FWHMs of signal peaks that correlated with stimulation strengths (Fig. 3k–o). These results demonstrate that CytoTape robustly resolves temporal features of multiple transcriptional events over 3 weeks.

To evaluate whether CytoTape can simultaneously record multiple IEG-promoter-driven signals, a key capability for studying complex processes[31], we co-expressed the structural monomer, the timestamp monomer, the *Fos* signal monomer fused to the V5 tag and the *Arc* signal

**Fig. 4 | CytoTape reveals temporal decoupling between CREB and FOS activities in single cells. a**, Schematics of constructs transfected into HEK cells. **b**, Experimental timelines (top) and expected distributions of JF dyes along the CytoTape (bottom). **c**, A representative confocal image of CytoTape in HEK cells expressing constructs shown in panel **a** (left); timestamp channels (right, top row); and CREB and FOS activity channels (right, bottom row). The arrowheads indicate the transient dye switching. HEK cells were stimulated with 50 μM FSK for 1 h at day 2.25. Scale bar, 10 μm. **d**, Uniform manifold approximation and projection (UMAP) plot of time-lagged correlations between pCREB and FOS signals across single cells ($n = 77$ CytoTapes from 77 cells, 5 cultures). **e**, Heatmaps showing time-lagged correlations between pCREB and FOS signals for type 1 (left; $n = 35$ CytoTapes from 35 cells) and type 2 (right; $n = 42$ CytoTapes from 42 cells) HEK cells. **f,g**, Single traces of pCREB (left) and

FOS (middle) signals relative change from baseline, plotted against recovered time post-transfection for type 1 (**f**) and type 2 (**g**) cells; the averaged curves of pCREB and FOS signals relative change from baseline plotted against recovered time post-transfection are also shown (right). The thick centre line denotes the mean, and the darker boundary shows the s.e.m. **h**, Quantification of baseline fluctuations in pCREB (left) and FOS (right) signals before stimulation for type 1 and type 2 cells from panel **e**. Baseline fluctuation is quantified by line length (sum of absolute differences between consecutive points) between day 1.5 and day 2.25. **i**, Average post-stimulation FOS signal in type 1 and type 2 cells. Throughout this figure, $*P < 0.05$ and $**P < 0.01$; two-sided Mann–Whitney $U$-test. For the boxplots in panels **h,i**, the middle line shows the median, the box boundary denotes the interquartile range, the whiskers indicate the minimum and maximum, and the black dots represent individual data points.

---

monomer fused to the OLLAS tag in cultured neurons (Fig. 3p–t). We observed that both *Fos*-promoter-driven and *Arc*-promoter-driven signals increased after FSK stimulation, confirming that CytoTape can simultaneously record multiple distinct gene regulation dynamics along the same fibre and within the same cell (Fig. 3r,s). Signal waveform analysis revealed that although the *Fos*-promoter-driven signal initiated earlier than the *Arc*-promoter-driven signal, the two signals exhibited similar peak timing (Fig. 3t). To further assess multiplexing capacity, we co-expressed five signal monomers for *Fos*, *Arc*, *Egr1*, NPAS4 and CREB activities, each carrying a unique epitope tag, together with structural and timestamp monomers in HEK cells (Fig. 3u–x). CytoTape simultaneously recorded the timestamps and the five distinct gene regulation dynamics along single fibres (Fig. 3w,x). Unlike other signals, the NPAS4 signal showed no response to FSK treatment, consistent with previous reports in human induced pluripotent stem cell-derived organoids[32] (Fig. 3w,x).

## Temporal principles of gene regulation

CREB and FOS are two transcription factors involved in signalling-dependent gene expression programs in mammalian cells[33]. We applied CytoTape to investigate how CREB activation is transduced into FOS induction over time (Fig. 4a–c), a process that informs how cells integrate signalling to control gene expression[34]. In HEK cells, FSK-induced CREB activation preceded FOS expression, consistent with the upstream role of CREB in the transcriptional cascade (Extended Data Fig. 5a). We computed time-lagged correlations between the two temporal trajectories, and dimensional reduction of the correlation profiles revealed two distinct cell clusters (Fig. 4d): type 1 cells ('decoupled mode') exhibited little or no correlation between CREB and FOS activities at any time lag, whereas type 2 cells ('coupled mode') displayed strong positive correlations near zero lag, indicating coordinated CREB–FOS dynamics (Fig. 4e). Correspondingly, type 2 cells showed a canonical CREB–FOS cascade (Fig. 4g), whereas type 1 cells showed irregular FOS fluctuations despite comparable CREB activation (Fig. 4f).

We hypothesized that this divergence arises from differences in both transcriptional state and signal integration[35,36]. Supporting this hypothesis, FOS activity fluctuated modestly more in type 1 cells before CREB activation, suggesting that a history of active FOS regulation may place the gene in a refractory state, limiting its response to new CREB activation (Fig. 4h,i). Immunostaining against pCREB and FOS proteins (Extended Data Fig. 5b,c) and fluorescent protein reporter assays (Extended Data Fig. 5d,e) at 1 h and 24 h after FSK stimulation likewise showed heterogeneous and often uncorrelated pCREB and FOS signals, confirming that CREB phosphorylation alone is insufficient to ensure FOS induction.

To investigate the molecular basis of CREB–FOS decoupling, we asked whether other signalling inputs influence whether CREB phosphorylation is transduced into FOS transcription (Extended Data Fig. 6). Because the *Fos* gene promoter contains *cis*-regulatory elements including the

cAMP response element (regulated by cAMP→PKA→pCREB signalling) and the serum response element (regulated by MEK→ERK→pElk-1/SRF signalling), we hypothesized that CREB–FOS decoupling arises when ERK signalling is absent or ineffective to integrate inputs[37]. Consistent with this hypothesis, CytoTape showed that inhibition of ERK signalling by U0126, a MEK inhibitor, abolished FOS induction under FSK stimulation, whereas activation of ERK signalling by epidermal growth factor (EGF) resulted in markedly strong FOS induction under FSK stimulation (Extended Data Fig. 6c,d). In both conditions, CREB remained robustly activated by FSK. These results were independently validated by protein immunostaining against pCREB and FOS (Extended Data Fig. 6e,f). Compared with FSK alone, FSK + U0126 increased the portion of cells in the decoupled state, whereas FSK + EGF did not decrease this portion (Extended Data Fig. 6g,h), indicating that ERK signalling is necessary but not sufficient for robust FOS induction and that other mechanism (or mechanisms) beyond ERK signalling may contribute to CREB–FOS decoupling. These results suggest a signal-dependent and state-dependent mechanism in IEG regulation, demonstrating the utility of CytoTape in dissecting temporal cellular signals.

We also investigated how distinct IEGs are coordinately and differentially regulated across time by simultaneously recording the transcriptional dynamics driven by the *Arc* promoter and the *Egr1* promoter in cultured neurons. We observed that *Arc*-promoter-driven and *Egr1*-promoter-driven transcriptional dynamics exhibit complex, multi-peak activation waveforms following a single upstream stimulation, revealing stimulus-dependent temporal correlations between distinct IEGs (Extended Data Fig. 7 and Supplementary Fig. 18).

## Spatiotemporal in vivo recording

To extend the utility of CytoTape for in vivo applications, we evaluated its performance in the living mouse brain. CytoTape formed thin and flexible fibres in vivo (Extended Data Fig. 8a), significantly thinner than XRI fibres (Extended Data Fig. 8b), demonstrating our design strategy works in both cultured cells and in vivo. However, CytoTape formed multiple fibres within each soma (more than 70% of neurons formed over three fibres per soma) that are closely spaced (Extended Data Fig. 8c), whereas cultured neurons typically contain one or two soma-localized CytoTape fibres (Supplementary Fig. 6g). This close fibre spacing could make conventional fluorescence microscopy insufficient to accurately resolve and quantify individual fibres in intact tissue. We therefore optimized CytoTape for in vivo applications by retaining the L349K mutation in 1POK(E239Y), and testing variants with linker lengths of 6–18 amino acids between 1POK(E239Y, L349K) and MBP (Extended Data Fig. 8b and Supplementary Table 8). The resulting construct, CytoTape-vivo with a six-residue linker, produced only one or two fibres per soma in over 70% of neurons (Extended Data Fig. 8d), while maintaining thin morphology comparable with CytoTape and significantly thinner than XRI (Extended Data Fig. 8b). Immunohistochemical characterizations showed that weeks-long expression of

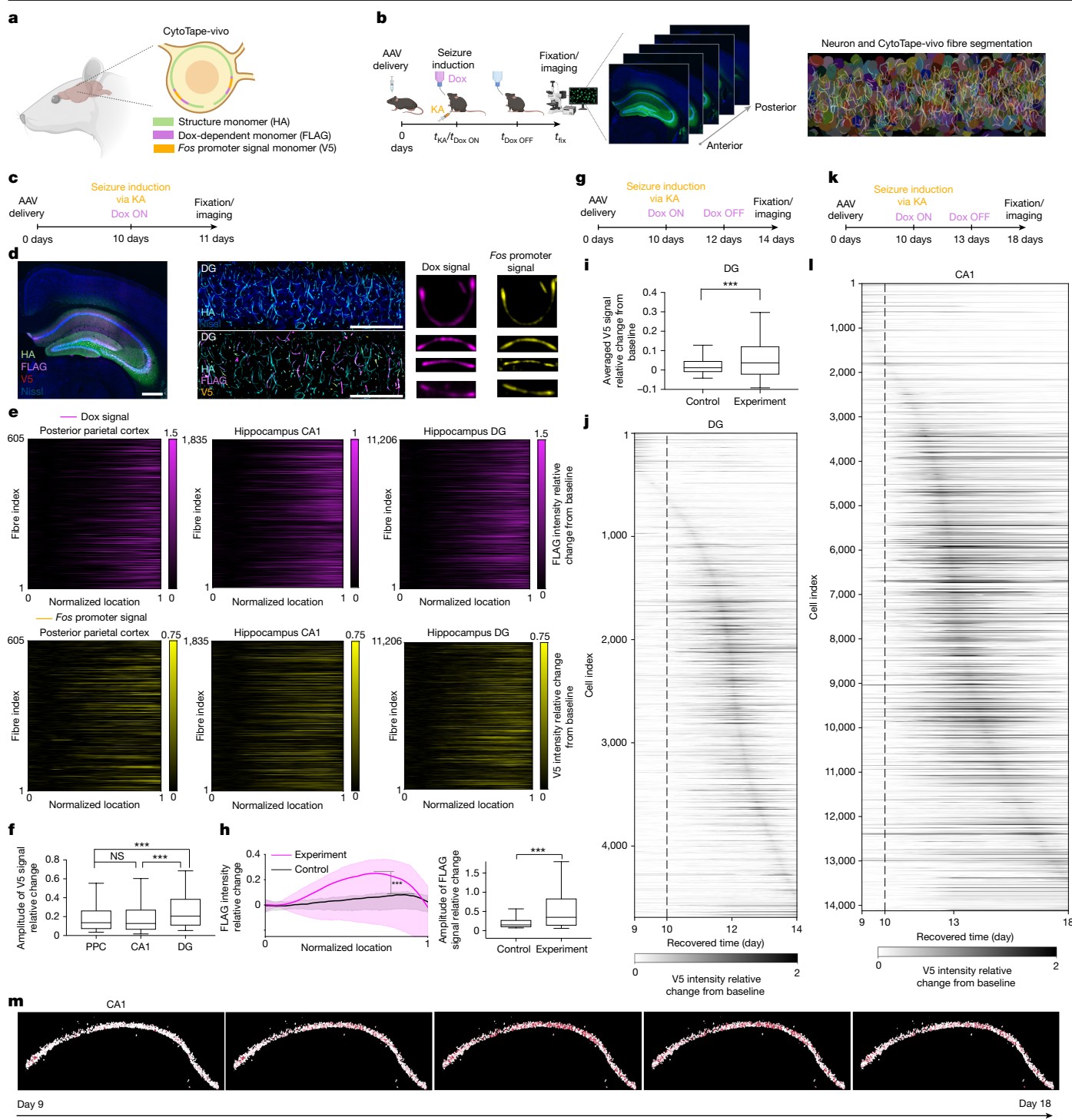

**Fig. 5 | See next page for caption.**

CytoTape-vivo in the mouse brain does not alter cellular physiology and synaptic state markers (Extended Data Fig. 9). Behavioural tests, including open field test, novel object recognition and contextual fear conditioning, further confirmed that mice with hippocampus-wide, weeks-long CytoTape-vivo expression in both hemispheres retained normal hippocampal function in vivo (Extended Data Fig. 10). Thus, we used CytoTape-vivo as the structural monomer for in vivo applications.

We next evaluated the multiplexed recording capability of CytoTape-vivo by co-expressing a Dox-inducible (Tet-On)[38] signal monomer and a *Fos*-promoter-driven signal monomer in the living mouse brain (Fig. 5a,b). Seizure was induced via kainic acid injection on

day 10, concurrent with Dox administration via drinking water, and the mice were perfused on day 11 for immunofluorescence readout under confocal microscopy (Fig. 5c–f). Both Dox-dependent monomers and *Fos*-promoter-driven signal monomers accumulated at fibre termini (Fig. 5d), indicating successful simultaneous in vivo recording. Analysis of 8,639 neurons across hippocampal CA1, the dentate gyrus and the posterior parietal cortex via Tape Reader, a custom high-throughput image analysis platform (Supplementary Video 1), demonstrated that CytoTape-vivo enables large-scale, single-cell readouts of multiplexed gene expression histories from a single confocal dataset (Fig. 5e). Dentate gyrus neurons displayed higher *Fos*-promoter signal amplitudes

**Fig. 5 | CytoTape-vivo simultaneously records Dox-dependent and *Fos* signal monomer expression histories over weeks in single cells across multiple brain regions in vivo. a**, Schematic of the CytoTape-vivo system for multiplexed, scalable recording of cellular signals in living mouse brains. **b**, Experimental pipeline for in vivo application (left); low-magnification coronal view confocal images from a mouse brain expressing CytoTape-vivo (middle); and computational segmentation of neuronal soma and CytoTape-vivo fibres from high-magnification dentate gyrus volumes (right). Schematics in panels **a**,**b** created in BioRender; Zheng, L. https://biorender.com/3uq51z7 (2026). **c**, Experimental timeline for multiplexing in vivo. KA, kainic acid. **d**, Low-magnification image of a representative brain slice (left). Scale bar, 500 μm. A 3.2-μm-thick maximum-intensity projection in the dentate gyrus (DG; middle). Scale bars, 50 μm. Confocal images showing *Fos*-promoter-driven and Dox-dependent signal monomer expression in DG neurons (right). **e**, Dox and *Fos* promoter signals in the posterior parietal cortex (left), hippocampal CA1 (middle) and hippocampal DG (right); $n$ = 605, 1,835 and 11,206 fibres from 600, 1,697 and 6,342 neurons, respectively, from one mouse. **f**, *Fos* promoter signal amplitudes from panel **e**. PPC, posterior parietal cortex. **g**, Experimental timeline for 14-day *Fos*-promoter

signal recording in vivo. **h**, Dox signal changes over 14 days in the DG in control (no Dox and no KA; $n$ = 718 neurons from one mouse) and experimental (described in panel **g**; $n$ = 4,692 neurons from one mouse) mice (left). The centre line shows the mean, and the shaded area denotes the interquartile range. Dox signal amplitudes change in control and experimental mice (right). **i**, Averaged *Fos* promoter signal changes in control and experimental mice from panel **h**. **j**, Recorded *Fos* promoter signal changes in DG neurons under a KA-induced seizure in the experimental mice from panel **h**. The dashed line indicates seizure induction. **k**, Experimental timeline for 18-day *Fos* promoter signal recording in vivo. **l**, Recorded *Fos* promoter signal intensity changes in CA1 neurons under KA-induced seizure (described in panel **k**); $n$ = 14,123 neurons from one mouse. The dashed line indicates seizure induction. **m**, Spatiotemporal mapping of *Fos* promoter signals over weeks from the recording in panel **l**. The red colour intensity represents the *Fos* promoter signal intensity, spatially mapped across time. For the boxplots in panels **f**,**h**,**i**, the middle line indicates the median, the box boundary shows the interquartile range, and the whiskers denote the 10th–90th percentile. Throughout this figure, \*\*\**P* < 0.001; Kruskal–Wallis analysis with Dunn's post-hoc tests.

---

than CA1 or posterior parietal cortex neurons (Fig. 5f), which is in accordance with previous immunostaining-based observations of endogenous FOS induction under kainic acid seizure[39], although the endogenous FOS activity is regulated by additional factors beyond promoter activity, such as enhancer activity[40], chromatin accessibility[41] and histone modifications[42]. Unlike immunostaining, which only provides a static snapshot in time, CytoTape-vivo preserves temporal trajectories to support causal inference.

To recover time axes in vivo, we utilized Dox-dependent monomers as timestamps, capitalizing on their reversible ON and OFF switching, widespread use in in vivo applications, simple and non-invasive oral administration and physiological compatibility[43]. Mice were injected with kainic acid on day 10, with Dox in drinking water switched ON on day 10 and OFF on day 12, and fixed on day 14 (Fig. 5g–j). Control mice lacking Dox or kainic acid exhibited minimal Dox or *Fos* promoter signals (Fig. 5h,i and Extended Data Fig. 8e). Using our established computational framework for time recovery from timestamps (Fig. 2), we aligned Dox ON and OFF and fixation times with locations along each fibre corresponding to the onset of Dox signal rise, the onset of Dox signal decay and fibre termini (Fig. 5h), and reconstructed the time axis for recorded *Fos* promoter signals across 4,692 dentate gyrus neurons (Fig. 5j and Extended Data Fig. 8f). To demonstrate temporal scalability, we repeated kainic acid injection on day 10, but with Dox-ON on day 10, Dox-OFF on day 13 and fixation on day 18 (Fig. 5k and Extended Data Fig. 8g), where CytoTape-vivo successfully recorded *Fos* promoter activity across 14,123 CA1 neurons (Fig. 5l and Extended Data Fig. 8h–i) and generated single-cell spatiotemporal maps of weeks-long transcriptional histories in CA1 (Fig. 5m). These observations provide examples of the unique type of readout CytoTape-vivo empowers, and do not represent a full scientific study that involves extensive experimentation beyond the scope of this technology development work. Together, these results demonstrate the power of CytoTape-vivo for multiplexed, spatiotemporally resolved, large-scale, single-cell recording of gene regulation dynamics in vivo.

## Discussion

In this work, we developed CytoTape, a genetically encoded, modular protein tape recorder for multiplexed, continuous recording of cellular activities with spatiotemporal resolution and scalability (Supplementary Table 9 and Supplementary Fig. 24). CytoTape embeds temporal signal along spatial dimension, continuously recording transcriptional dynamics over time as analogue waveforms, with minutes-scale intrinsic temporal resolution and up to 3-week-long recording duration. Its flexible, thread-like structure allows elongation beyond the cell size for long-term recording, whereas the diversity of epitope tags expands the alphabet size, enabling multiplexed recording of three-colour

timestamps and five-colour transcriptional dynamics on a single fibre. CytoTape exhibits symmetrical bidirectional growth, providing two copies of recording per fibre for cross-validation. CytoTape has no detectable sequence homology to eukaryotic genes and preserves cell physiological integrity and intact brain function in vivo. Correspondingly, a recent study has reported that 1POK(E239Y)-based protein self-assembly is inert in live yeast[44]. CytoTape functions across diverse cell types, including mouse neurons and glial cells, and human embryonic and cancer cell lines, and uses activity-dependent *cis*-regulatory elements to record transcriptional and transcription factor activities under various stimuli. CytoTape can be delivered transiently via standard DNA transfection or adeno-associated virus (AAV) transduction without genome editing, and readout requires only routine immunostaining and conventional microscopy. Building on this platform, CytoTape-vivo enables scalable, spatiotemporally resolved single-cell recording in vivo, incorporating Dox-dependent timestamps delivered non-invasively via drinking water for multiplexed recording of transcriptional activities across multiple brain regions.

Molecular recorders based on CRISPR integrases, base editors and prime editors have enabled transcriptional events to be recorded into genomic DNA for scalable sequencing readout[13,45,46] or spatially resolved, in situ hybridization-based imaging readout in cultured cells[47]. However, these approaches do not support time-continuous recording and are primarily optimized for prokaryotes or mammalian cell cultures. Existing live-imaging techniques provide spatially resolved, time-continuous recording but are inherently limited by a spatial trade-off between resolution and scale. The CytoTape toolkit breaks through this trade-off by stably encoding cellular activities in situ, enabling post-hoc imaging readout with both high resolution and large scale. It also provides a record that moves with the cell (Supplementary Figs. 19–21).

Gene regulation dynamics mediated by mechanisms beyond *cis*-regulatory elements are not directly accessible with the CytoTape system, and its five-signal multiplexing capacity remains insufficient to fully dissect the complexity of large gene regulatory networks. We envision that future efforts will integrate CytoTape with various cellular activity reporter systems[48–50] to expand its recorder repertoire. In addition, multi-round immunostaining techniques[51] may greatly enhance the multiplexing capacity of CytoTape. CytoTape may also benefit from systematic gene delivery methods[52] and high-throughput imaging modalities to achieve organ-wide and even organism-wide recording scales.

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

# Methods

## CytoTape design strategy and pipeline

CytoTape structure monomer design uses XRI structure monomer as a template. Artificial intelligence-assisted protein design (A1 and A2 family designs; CPDiffusion and ProtSSN were used for sequence generation and site mutation, respectively, using 1POK(E239Y) as the template; the Methods for CPDiffusion and ProtSSN are described in Supplementary Information) and rational design (A3, B and C family designs; 1POK(E239Y) was replaced with other assembly proteins, and the linker and MBP were optimized by tuning their sequence lengths, adjusting the number of their tandem repeats, and replacing them with homologous sequences) were combined to develop the CytoTape monomer. After in silico design, design candidates were screened in cultured primary mouse hippocampal neurons. The *UbC* promoter was used to drive the steady expression of candidate protein monomers (Fig. 1c).

## Molecular dynamics simulations

The structure of the protein monomer for simulations was predicted by AlphaFold3 (ref. 53). Supplementary Fig. 5 shows the confidence analysis of CytoTape and XRI monomer structures (Fig. 1b,e, Extended Data Fig. 1c and Supplementary Fig. 3). The $C_\alpha$ structural alignment of CytoTape and XRI protein monomers was performed in PyMOL. The simulations were replicated three times. A cubic box was filled with one protein monomer. Water molecules were randomly inserted into the box to reach the protein that is fully covered by water molecules in the simulation. Chlorine counter ions were added to keep the system neutral in charge. The CHARMM27 force field was used for the complex and the CHARMM-modified TIP3P model was chosen for water. The simulations were carried out at 310 K (37 °C). After the 5,000-step energy-minimization procedure, the systems were heated and equilibrated for 100 ps in the NVT ensemble and 500 ps in the NPT ensemble. Production simulations (500 ns) were conducted with a trajectory saving frequency of 10 ps. The final 300 ns (30,000 frames) was extracted for subsequent analysis. The integration step was set to 2 fs, and only the covalent bonds with hydrogen atoms were constrained by the LINCS algorithm. Lennard–Jones interactions were truncated at 12 Å with a force-switching function from 10 to 12 Å. The electrostatic interactions were calculated using the particle mesh Ewald method with a cut-off of 12 Å on an 1 Å grid with a fourth-order spline. The temperature and pressure of the system were controlled by the velocity rescaling thermostat and the Parrinello–Rahman algorithm, respectively. All molecular dynamics simulations were performed using GROMACS 2021.1 packages.

The free-energy landscape (Extended Data Fig. 1b) was constructed using the trajectory data from the last 300 ns of the production simulations. Root-mean-square deviation (RMSD) from the initial structure and radius of gyration ($R_g$) were used to represent the conformational space of the protein. Trajectory frames were extracted at 10-ps intervals and aligned to the reference structure to remove overall translational and rotational motions. The two-dimensional histogram of RMSD and $R_g$ was computed using bins of 0.05 nm × 0.05 nm, and the corresponding free energy $G$ was calculated using the Boltzmann inversion: $G(\text{RMSD}, R_g) = -k_B T/\ln P(\text{RMSD}, R_g)$, where $k_B$ is the Boltzmann constant, T is the absolute temperature (310 K) and $P(\text{RMSD}, R_g)$ is the normalized probability distribution of the system in the two-dimensional conformational space. The analysis was performed using GROMACS tools (gmx rms, gmx gyrate and gmx sham), and the free-energy landscape was visualized as a contour map using Origin2019b.

To evaluate the relative motion between two structural domains within the same protein, the centre-of-mass (COM) distance between the domains was calculated over the course of the molecular dynamics simulation (Extended Data Fig. 1c). The simulation trajectory was preprocessed using gmx trjconv to remove periodic boundary conditions and align the protein to a common reference frame. The interdomain distance at each time point was determined as the Euclidean distance between the COM coordinates. Analysis was performed on the final 300 ns of the production simulation, sampled every 10 ps, and the resulting distance profile was visualized using Origin2019b.

## Molecular cloning

The DNAs encoding the protein motifs used in this work were mammalian-codon optimized and synthesized by Epoch Life Science, and then cloned into mammalian expression backbones: pAAV-*UbC* (for constitutive expression), pAAV-*Fos* (for expression driven by the *Fos* promoter), pAAV-*Arc* (for expression driven by the SARE-ArcMin promoter), pAAV-*Egr1* (for expression driven by the *Egr1* promoter), pAAV-NPAS4 (for expression driven by the *N*-RAM promoter), pAAV-FOS (for expression driven by the *F*-RAM promoter), pAAV-pCREB (for 6×*CRE*-CMVMin-dependent expression) or pAAV-*HSPA1A* (for expression driven by the HSE promoter with the heat shock element from the human *HSPA1A* promoter) for DNA transfection in cultured neurons, glial cells, HEK293T clone 17 (referred to as HEK throughout this paper; American Type Culture Collection CRL-11268; authenticated by short tandem repeat profiling; tested negative for mycoplasma contamination) cells, and HeLa cells (American Type Culture Collection CCL-2; authenticated by short tandem repeat profiling; tested negative for mycoplasma contamination). *Fos*, *Egr1*, SARE-ArcMin, HSE, *N*-RAM, *F*-RAM and 6×*CRE*-CMVMin promoters that are responsive to transcriptional activities of *Fos*, *Egr1*, *Arc*, *HSPA1A* genes and transcription factor activities of NPAS4, FOS and pCREB proteins, respectively, were used. pAAV-*UbC* was used to drive HaloTag-based timestamp monomer, which consists of CytoTape-HA with a C-terminal HaloTag (the HA tag was removed from this construct for experiments in Figs. 3u–x and 4 to accommodate the use of an HA-tagged signal monomer). pAAV-*hSyn* and pAAV-*TRE3G* (TakaraBio) were used to drive rtTA3G (TakaraBio) and Dox-based timestamp monomer, respectively, for Dox-dependent expression in vivo. See Supplementary Table 1 for sequences of the motifs, Supplementary Tables 7 and 8 for all constructs of protein monomer designs, and Supplementary Tables 2–6 for all designed signal constructs. The plasmids and the corresponding sequence maps of constructs reported in this paper are available at Addgene (plasmid IDs: 239423–239430, 239616, 250670–250672 and 252514). The sequences of CytoTape and CytoTape-vivo are available at GenBank (accession numbers: PX843332–PX843334).

## Animals

All animal procedures were conducted in accordance with the US National Institutes of Health Guide for the Care and Use of Laboratory Animals and were approved by the Institutional Animal Care and Use Committee of the institution where each procedure was conducted (University of Michigan, Icahn School of Medicine at Mount Sinai or Massachusetts Institute of Technology). Mice were maintained on a 12-h light–dark cycle (lights on at 06:00 EST) in a temperature-controlled environment at 22 ± 1 °C, with a relative humidity of 30–50%. Mice were group housed, except for pregnant females (individually housed before delivery) and mice post-surgery (individually housed for the remainder of the experiment). Preparation of primary neuron–glia co-cultures were conducted using both male and female neonatal mice delivered by pregnant females; adult mouse experiments were conducted using male mice. For behavioural tests, the number of mice per experiment was determined based on expected variance and effect size from previous studies[23,25]; no statistical method was used to predetermine sample size; mice were randomly allocated to either the experimental or control groups; experimenters were not blinded to group identity.

## Preparation of primary mouse hippocampal neuron–glia co-cultures

Cultured hippocampal neurons with glial cells were prepared from neonatal (postnatal day 0 or 1) Swiss Webster mice (Taconic; both

male and female mice were used) as previously described[2]. In brief, the brains of ice-anaesthetized neonatal mice were dissected out, and the hippocampal tissue was further dissected from the brains in an ice-cold dissection buffer (1 mM kynurenic acid, 10 mM $MgCl_2$ and 35 mM D-glucose in HBSS). The hippocampal tissue was subsequently treated with papain for 10 min at 37 °C, followed by a 5-min ovomucoid inhibition. The dissociated tissues were triturated, and a single-cell suspension was prepared using neuronal culture media (minimum essential medium (MEM); Gibco) supplemented with 10 mM HEPES, 10% heat-inactivated fetal bovine serum (HI-FBS; Corning), 2% B27 supplement (Gibco), 2 mM L-glutamine, 1 µM transferrin and 25 mM D-glucose; pH adjusted to 7.4). Cells were plated at a density of 40,000 cells per 100 µl on Matrigel (Corning)-coated coverslips (Carolina) and then incubated in a humidified cell culture incubator at 37 °C with 5% $CO_2$. After the cells had settled and attached to the coverslips, 1 ml of neuronal culture medium was added. When glial cells reached approximately 80% confluency (usually 2 days after plating), 1 ml neuronal culture medium supplemented with 4 µM AraC was added to the existing medium in each well.

## CytoTape expression in cultured neurons and glial cells

Cultured neurons and glial cells were transfected at 7 days in vitro (DIV7) for protein assembly design screening (Fig. 1 and Supplementary Figs. 3, 4 and 7), CytoTape with timestamps (Fig. 2 and Supplementary Fig. 12), and CytoTape-based transcriptional recorders (Extended Data Fig. 4 and Supplementary Figs. 14 and 15). For recording the time course of *Fos*, *Arc*, *Egr1*, NPAS4 and pCREB-promoter-driven expression (Fig. 3, Extended Data Fig. 7 and Supplementary Figs. 17 and 18), cells were transfected at DIV4. A commercial calcium phosphate transfection kit (Invitrogen) was used as previously described[2,5], with the following transfection parameters. For protein assembly design screening, each well of a 24-well glass-bottom plate received 50 ng of the structural monomer plasmid and 1,450 ng of the pUC19 plasmid (a 'dummy' plasmid without a mammalian open reading frame to maintain the optimal mass ratio between DNA and calcium phosphate for the formation of co-precipitates for transfection). In the timestamped CytoTape expression experiment, each well received 50 ng of the structural monomer plasmid (*UbC*-CytoTape), 10 ng of the timestamp monomer plasmid (*UbC*-CytoTape-HaloTag) and 1,440 ng of pUC19 plasmid. For the CytoTape-based transcriptional recorder experiment, each well was transfected with 50 ng of the structural monomer plasmid (*UbC*-CytoTape), 12.5 ng of CytoTape-HaloTag plasmid (*UbC*-CytoTape-HaloTag) and 1,437.5 ng of the pUC19 plasmid. In the single signal recording experiment, each well received 300 ng of the structural monomer plasmid (*UbC*-CytoTape), 60 ng of the timestamp monomer plasmid (*UbC*-CytoTape-HaloTag), 75 ng of the signal monomer plasmid (*Fos*, *Egr1*, *Arc* or NPAS4-CytoTape-V5) and 1,065 ng of the pUC19 plasmid. For the two-signal recording experiment, each well was transfected with 300 ng of the structural monomer plasmid (*UbC*-CytoTape), 60 ng of the timestamp monomer plasmid (*UbC*-CytoTape-HaloTag), 2 × 75 ng of the signal monomer plasmids (*Fos*-CytoTape-V5 and *Arc*-CytoTape-OLLAS) and 990 ng of pUC19 plasmid. After a 45-min incubation with DNA-calcium phosphate precipitates, cells were washed with a preformulated acidic buffer, HMEM (that is, the MEM buffer supplemented with 15 mM HEPES and then adjusted to a final pH of 6.70–6.74 with acetic acid (Millipore Sigma) over the time course of 6 h) to remove residual precipitates. Cells were then placed back in the cell culture incubator.

## GFP expression in cultured neurons

Cultured neurons were transfected at DIV7 (Supplementary Fig. 13). A commercial calcium phosphate transfection kit (Invitrogen) was used as previously described[2,5], with the following transfection parameters. Each well of a 24-well glass-bottom plate received 50 ng of the GFP plasmid and 1,450 ng of the pUC19 plasmid (a dummy plasmid without a mammalian open reading frame to maintain the optimal mass ratio between DNA and calcium phosphate for the formation of co-precipitates for transfection). After a 45-min incubation with DNA–calcium phosphate precipitates, cells were washed with a preformulated acidic buffer, HMEM to remove residual precipitates. Cells were then placed back in the cell culture incubator.

## CytoTape expression in HEK and HeLa cells

HEK and HeLa cells at a low passage number (less than 10 passages) were plated at 40% confluence onto a 200-mm tissue culture dish. Dulbecco's modified eagle's medium (DMEM; Gibco; containing high glucose, GlutaMAX supplement and pyruvate) supplemented with 10% HI-FBS (Corning) and penicillin–streptomycin (Gibco) was used as the HEK and HeLa cell culture medium. Cells were then grown in a cell culture incubator at 37 °C with 5% $CO_2$. After reaching 50–70% confluence, cells were transferred to 24-well glass-bottom plates via trypsin treatment. The 24-well glass-bottom plates were pre-treated with Matrigel at room temperature for 30 min before cell plating. Twenty-four hours after cell plating, genes were delivered using the TransIT-X2 Dynamic Delivery System kit (Mirus Bio).

For the structural monomer expression experiment (Extended Data Fig. 1 and Supplementary Figs. 6, 7 and 9), 100 ng structural monomer plasmid (*UbC*-CytoTape-HA) and 200 ng pUC19 plasmid were first diluted in 50 µl Opti-MEM medium (Gibco), followed by the addition of 0.9 µl TransIT-X2 reagent. The mixture was incubated at room temperature for 25 min and then added to the single well. The cells were further incubated at 37 °C with $CO_2$ for 24 h. For timestamped CytoTape expression experiment (Fig. 2 and Supplementary Figs. 10–12), 100 ng structural monomer plasmid (*UbC*-CytoTape-HA), 20 ng timestamp monomer plasmid (*UbC*-CytoTape-HaloTag) and 180 ng pUC19 plasmid were first diluted in 50 µl Opti-MEM medium, followed by the addition of 0.9 µl TransIT-X2 reagent. The mixture was incubated at room temperature for 25 min and then added to the single well. The cells were further incubated at 37 °C with $CO_2$ for 24 h. For single-signal recording experiment (Supplementary Figs. 16, 19 and 21), 100 ng structural monomer plasmid (*UbC*-CytoTape-HA), 20 ng timestamp monomer plasmid (*UbC*-CytoTape-HaloTag), 25 ng signal monomer plasmid (*Fos*, pCREB or *HSPA1A*-CytoTape-V5) and 155 ng pUC19 plasmid were first diluted in 50 µl Opti-MEM medium, followed by the addition of 0.9 µl TransIT-X2 reagent. The mixture was incubated at room temperature for 25 min and then added to the single well. The cells were further incubated at 37 °C with $CO_2$ for 24 h. For two-signal recording experiment (Fig. 4 and Extended Data Figs. 5 and 6), 100 ng structural monomer plasmid (*UbC*-CytoTape), 20 ng timestamp monomer plasmid (*UbC*-CytoTape-HaloTag), 2 × 25 ng signal monomer plasmids (pCREB-CytoTape-HA and FOS-CytoTape-OLLAS) and 130 ng pUC19 plasmid were first diluted in 50 µl Opti-MEM medium, followed by the addition of 0.9 µl TransIT-X2 reagent. The mixture was incubated at room temperature for 25 min and then added to the single well. The cells were further incubated at 37 °C with $CO_2$ for 24 h. For five-signal recording experiments (Fig. 3), 180 ng structural monomer plasmid (*UbC*-CytoTape), 36 ng CytoTape-HaloTag plasmid (*UbC*-CytoTape-HaloTag) and 5 × 25 ng signal monomer plasmid (pCREB-CytoTape-HA, *Egr1*-CytoTape-Etag, *Fos*-CytoTape-OLLAS, *Arc*-CytoTape-V5 and NPAS4-CytoTape-FLAG) were first diluted in 50 µl Opti-MEM medium, followed by the addition of 1.5 µl TransIT-X2 reagent. The mixture was incubated at room temperature for 25 min and then added to the well. The cells were further incubated in the cell culture incubator at 37 °C with 5% $CO_2$ for 24 h.

## Fluorescence protein expression in HEK cells

HEK cells at a low passage number (less than 10 passages) were plated at 40% confluence onto a 200-mm tissue culture dish. DMEM supplemented with 10% HI-FBS and penicillin–streptomycin was used as the HEK cell culture media. Cells were then grown in a cell culture incubator

at 37 °C with 5% $CO_2$. After reaching 50–70% confluence, cells were transferred to 24-well glass-bottom plates via trypsin treatment. The 24-well glass-bottom plates were pre-treated with Matrigel at room temperature for 30 min before cell plating. Twenty-four hours after cell plating, genes were delivered using the TransIT-X2 Dynamic Delivery System kit. 100 ng plasmid (FOS–eGFP, pCREB–eGFP or FOS–mRuby3) and 200 ng pUC19 plasmid were first diluted in 50 μl Opti-MEM medium, followed by the addition of 0.9 μl TransIT-X2 reagent. The mixture was incubated at room temperature for 25 min and then added to the single well. The cells were further incubated at 37 °C with $CO_2$ for 24 h (Extended Data Fig. 5 and Supplementary Fig. 19–21).

### CytoTape timestamps labelling

CytoTape was labelled with cell-permeant HaloTag ligand JF dyes (Promega; Figs. 2–4, Extended Data Figs. 5a, 6a–d,g,h and 7 and Supplementary Figs. 10–12, 16, 17, 19 and 21). $JF_{503}$, $JF_{585}$ and $JF_{635}$ were used in this study for timestamps. The JF dyes were in lyophilized powder form and stored at −20 °C before use. The JF dye powder was first dissolved in 50 μl dimethyl sulfoxide (DMSO) and then diluted into 10 ml of HEK and HeLa cell culture media (for HEK and HeLa cells) or into 10 ml of neuronal culture media (for cultured neurons and glial cells) at a final concentration of 0.1 μM. The dyed medium was filtered using a sterile 0.22-μm syringe filter before being added to the culture. It was then used to replace the original media in cell cultures at designated time points. During the dye-switching process, the medium containing the original dye was fully removed, followed by thorough washes of the cell cultures five times with 37 °C DMEM for HEK and HeLa cells, and MEM for cultured neurons and glial cells. Finally, the cells in each well of the 24-well plate were cultured in 500 μl of fresh HEK and HeLa cell culture media (for HEK and HeLa cells) or 1 ml neuronal culture media (for cultured neurons and glial cells) supplemented with a new JF dye at a final concentration of 0.1 μM in the cell culture incubator at 37 °C with 5% $CO_2$.

### Chemical stimulation of cultured neurons

For KCl stimulation (Fig. 3, Extended Data Figs. 1, 4 and 7 and Supplementary Figs. 12–15 and 17), a KCl stock buffer was prepared, containing 170 mM KCl, 2 mM $CaCl_2$, 1 mM $MgCl_2$ and 10 mM HEPES. Then, a KCl depolarization medium was prepared by mixing the KCl stock buffer with fresh neuronal culture medium, ensuring that the final concentration of $K^+$ after mixing was 55 mM or 30 mM (accounting for the $K^+$ already present in the fresh neuronal culture medium). The original culture medium from neuron cultures was transferred into a fresh 24-well plastic-bottom plate, where the medium from different neuron cultures was stored in separate wells and kept in the neuronal incubator until the end of the KCl-induced depolarization treatment. Then, 500 μl of KCl depolarization medium was added to each well containing neuron cultures. The neuron cultures were placed back into the incubator and incubated for 10 min, 30 min, 1 h, 2 h or 3 h. Finally, the KCl depolarization medium was removed, and the original neuronal culture medium was transferred back into the corresponding wells. The neuron cultures were then returned to the cell culture incubator.

For FSK stimulation (Fig. 3, Extended Data Figs. 4 and 7 and Supplementary Fig. 13), FSK powder was dissolved in DMSO, with a final concentration of 5 mM. Then, a FSK stimulation medium was prepared by mixing the 5 mM FSK solution with a fresh neuronal culture medium, ensuring that the final concentration of FSK after mixing was 5 μM, 10 μM, 20 μM and 25 μM. The FSK stimulation medium was filtered using a sterile 0.22-μm syringe filter (VWR, Avantor) before being added to neurons. The original culture medium from neuron cultures was transferred into a fresh 24-well plastic-bottom plate, where the medium from different neuron cultures was stored in separate wells and kept in the neuronal incubator until the end of the FSK stimulation. Then, 500 μl of FSK stimulation medium was added to each well containing neuron cultures. The neuron cultures were placed back into the incubator and incubated for 1 h. Finally, the FSK stimulation medium was

removed, and the original neuronal culture medium was transferred back into the corresponding wells. The neuron cultures were then returned to the cell culture incubator.

### Chemical stimulation of cultured mouse hippocampal glial cells

For KCl stimulation (Supplementary Fig. 15), a KCl stock buffer was prepared, containing 170 mM KCl, 2 mM $CaCl_2$, 1 mM $MgCl_2$ and 10 mM HEPES. Then, a KCl depolarization medium was prepared by mixing the KCl stock buffer with fresh neuronal culture medium, ensuring that the final concentration of $K^+$ after mixing was 55 mM (accounting for the $K^+$ already present in the fresh neuronal culture medium). The original culture medium was transferred into a fresh 24-well plastic-bottom plate, where the medium from different cultures was stored in separate wells and kept in the neuronal incubator until the end of the KCl-induced depolarization treatment. Then, 500 μl of KCl depolarization medium was added to each well containing cultures. The cultures were placed back into the incubator and incubated for 1 h. Finally, the KCl depolarization medium was removed, and the original culture medium was transferred back into the corresponding wells. The cultures were then returned to the cell culture incubator.

### Chemical stimulation of HEK cells

For FSK stimulation (Figs. 3 and 4, Extended Data Fig. 5 and Supplementary Figs. 16 and 19–21), FSK powder was dissolved in DMSO to a final concentration of 5 mM. Then, an FSK stimulation medium was prepared by mixing the 5 mM FSK solution with fresh HEK cell culture medium, ensuring that the final concentration of FSK after mixing was 10 μM, 20 μM and 50 μM. The FSK stimulation medium was filtered using a sterile syringe filter before being added to HEK cells. The original culture medium from HEK cell cultures was removed. Then, 500 μl of FSK stimulation medium was added to each well and incubated for 1 h. Finally, the FSK stimulation medium was removed, and fresh culture medium was transferred into the corresponding wells. The HEK cell cultures were then returned to the cell culture incubator.

For FSK + U0126 or FSK + EGF stimulation (Extended Data Fig. 6), FSK powder was dissolved in DMSO to 5 mM, U0126 powder was dissolved in DMSO to 20 mM, EGF powder was dissolved in water with 0.1% BSA to 25 mM. Then, a stimulation medium was prepared by mixing the 5 mM FSK solution with either 20 mM U0126 or 25 mM EGF with fresh HEK cell culture medium, yielding final concentrations of 50 μM FSK and 10 μM U0126 or 20 μM EGF. The stimulation medium was filtered using a sterile syringe filter before being added to HEK cells. The original culture medium from HEK cell cultures was removed. Then, 500 μl of stimulation medium was added to each well and incubated for 1 h. Finally, the stimulation medium was removed, and fresh culture medium was transferred into the corresponding wells. The HEK cell cultures were then returned to the cell culture incubator.

### Heat treatments of HEK and HeLa cells

HEK (Supplementary Fig. 16) and HeLa cells (Supplementary Fig. 6) were cultured in cell culture media in a cell culture incubator set to 5% $CO_2$ at 37 °C. Cells were plated in 24-well glass-bottom plates to achieve 90–100% confluence before the heat shock treatment. Then, the plates containing the cells were transferred into a mini-incubator preheated to 42 °C and incubated there for 1 h. After the heat shock, the plates were returned to the original 37 °C cell culture incubator for recovery.

### Antibodies, stains and dyes

Primary antibodies (1:500 for immunofluorescence of cultured cells and brain slices unless specified below): anti-HA (3724, Cell Signaling Technology; PA5-33243, Invitrogen), anti-V5 (R960-25, Invitrogen), anti-FLAG (740001, Invitrogen), anti-OLLAS (MA5-16125, Invitrogen), anti-Etag (MA5-38276, Invitrogen), FOS monoclonal antibody (226009, Synaptic Systems), pCREB(Ser133) monoclonal antibody (9198, Cell Signaling Technology), anti-pElk-1 (9186, Cell Signaling

Technology), anti-Egr1 (4153, Cell Signaling Technology), anti-p-p44/42 MAPK (Erk1/2)(Thr202/Tyr204) (4370, Cell Signaling Technology), anti-GFAP (12389, Cell Signaling Technology), anti-cleaved caspase-3 (9664, Cell Signaling Technology, 1:250 for brain slices), anti-Hsp70 (4872, Cell Signaling Technology, 1:200 for brain slices), anti-Hsp27 (2402, Cell Signaling Technology, 1:50 for brain slices), anti-$\gamma$H2AX (05-636, Sigma), anti-synaptophysin (S5768, Sigma), anti-NeuN (266004, Synaptic Systems, 1:1,000 for brain slices), anti-Iba1 (019-19741, Wako Chemicals), anti-Ki67 (ab279653, Abcam), anti-GRP78 BiP (ab21685, Abcam) and anti-TOMM20 (ab186735, Abcam).

Fluorescent secondary antibodies (1:500 for immunofluorescence of cultured cells and brain slices): goat anti-mouse IgG1 ATTO 425 (610-151-040, Rockland), goat anti-mouse IgG2a Alexa Fluor 488 (A-21131, Invitrogen), goat anti-guinea pig IgG (H + L) Alexa Fluor 488 (A-11073, Invitrogen), goat anti-rabbit IgG (H + L) Alexa Fluor 488 (A-11008, Invitrogen), goat anti-rabbit IgG (H + L) Alexa Fluor Plus 488 (A-32731, Invitrogen), goat anti-mouse IgG2a Alexa Fluor 546 (A-21133, Invitrogen), goat anti-rat IgG (H + L) Alexa Fluor 546 (A-11081, Invitrogen), goat anti-rabbit IgG (H + L) Alexa Fluor 594 (A-11012, Invitrogen), goat anti-chicken IgY (H + L) Alexa Fluor Plus 647 (A-32933, Invitrogen) and goat anti-mouse IgG1 Alexa Fluor 647 (A-21240, Invitrogen).

**Nissl stain.** NeuroTrace 435/455 blue fluorescent Nissl stain (N21479, Invitrogen, 1:1,000 for immunofluorescence of cultured cells and 1:500 for immunofluorescence of brain slices).

**DAPI stain.** DAPI (4′,6-diamidino-2-phenylindole) nuclear stain (62248, Thermo Scientific, 1:1,000 for immunofluorescence of cultured cells).

**CellMask stain.** CellMask plasma membrane stain (C56129 or C37608, Thermo Scientific, 1:1,000 for immunofluorescence of cultured cells).

**Fluorescent dyes.** $JF_{503}$ HaloTag ligand (Promega; 0.1 μM in cultured cells), $JF_{585}$ HaloTag Ligand (Promega; 0.1 μM in cultured cells), $JF_{635}$ HaloTag Ligand ($JF_{635}$; Promega; 0.1 μM in cultured cells), ethidium homodimer-1 (E1169, Invitrogen, 1:1,000 in cultured cells) and calcein AM (ab270788, Biotium, 1:1,000 in cultured cells).

Additional details of primary antibodies, secondary antibodies, stains, dyes and other reagents used in this study are listed in Supplementary Table 10.

### Immunostaining of epitope tags in cultured cells

Cells (neurons, glial, HEK and HeLa cells) were fixed in 10% buffered formalin (Fisher Scientific) at room temperature for 10 min, followed by three 5-min washes in 1× PBS at room temperature. Blocking was performed in MAXBlock blocking medium (Active Motif) supplemented with 0.1% Triton X-100 and 100 mM glycine for 20 min at room temperature, followed by three additional 5-min washes in MAXwash washing medium (Active Motif) at room temperature. Cells were then incubated with primary antibodies diluted in MAXbind staining medium (Active Motif) for 1 h at room temperature. Afterwards, cells underwent three 5-min washes in MAXwash washing medium at room temperature. Secondary antibodies were applied in MAXbind staining medium and incubated for 1 h at room temperature. Cells were then washed three times with MAXwash washing medium for 5 min each at room temperature. Finally, cells were incubated with NeuroTrace 435/455 blue fluorescent Nissl stain (Invitrogen) or CellMask (Thermo Scientific) for 10 min and stored in 1× PBS at 4 °C until imaging.

### Immunostaining of pCREB, FOS, EGR1, pElk-1 and pERK1/2 protein in HEK cells

HEK cells were fixed in 10% buffered formalin (Fisher Scientific) at room temperature for 10 min, followed by three 5-min washes in 1× PBS at room temperature. Blocking was performed in MAXBlock blocking medium (Active Motif) supplemented with 0.1% Triton X-100 and 100 mM glycine for 20 min at room temperature, followed by three additional 5-min washes in MAXwash washing medium (Active Motif) at room temperature. Cells were then incubated with primary antibodies diluted in MAXbind staining medium (Active Motif) for 24 h at 4 °C. Afterwards, cells underwent three 5-min washes in MAXwash washing medium at room temperature. Secondary antibodies were applied in MAXbind staining medium and incubated for 24 h at 4 °C. Cells were then washed three times with MAXwash washing medium for 5 min each at room temperature. Finally, cells were incubated with DAPI nuclear stain (Thermo Scientific) for 10 min and stored in 1× PBS at 4 °C until imaging (Extended Data Figs. 3, 5 and 6 and Supplementary Fig. 20).

### Fluorescence microscopy

Fluorescence microscopy was performed using a spinning disk confocal microscope (Yokogawa CSU-W1 Confocal Scanner Unit on a Nikon Eclipse Ti2 inverted microscope) equipped with a 40 × 1.15 numerical aperture water immersion objective (Nikon MRD77410), a ×20 objective, a ×10 objective and a ×4 objective, a Hamamatsu ORCA-Fusion BT sCMOS camera controlled by the NIS-Elements AR software, and laser/filter sets for 405-nm, 488-nm, 561-nm, 594-nm and 640-nm optical channels. Multi-channel volumetric imaging was conducted at 0.4 μm per Z-step for each field of view under the ×40 objective. No optical crosstalk between 561-nm and 594-nm channels was observed for the fluorescent dye combinations used in this work, as shown in Supplementary Fig. 23. Imaging parameters remained consistent across all samples within each experimental set.

### Photobleaching of JF dyes

For the experiments described in Figs. 3 and 4 and Extended Data Fig. 7, after imaging the JF dyes in the samples by confocal microscopy, we exposed the samples to light to completely photobleach the fluorescence of JF dyes and free up the optical channels for subsequent immunofluorescence staining. Samples in 24-well plates were placed approximately 10 cm away from a broadband white LED light source (MNWHLP1, ThorLabs) to ensure uniform exposure to LED illumination. We checked the fluorescence of JF dyes in the samples after every 10 min of LED exposure and continued the exposures until the fluorescence of JF dyes dropped below the detection limit of the confocal microscope. The total LED exposure duration typically ranged from 60 to 120 min, depending on the initial fluorescence of JF dyes in the samples before LED exposure. Example confocal images of $JF_{585}$ and $JF_{635}$ before and after LED bleaching are shown in Supplementary Fig. 23.

### Re-identify the protein fibre after photobleaching

We recorded the microscope stage coordinates of the fields of view when imaging the JF dyes. We also acquired brightfield images of the fields of view to capture morphological and spatial features of imaged cells. After immunofluorescence staining, we located the previously imaged fibres by combining stage-coordinate guidance with brightfield-based visual registration of the cells and fields of views.

### Electrophysiology

For Extended Data Fig. 2a–e (membrane properties), cultured neurons were transduced with AAVs at DIV7 for electrophysiology at DIV15–18. Neurons were bathed in a room temperature Tyrode solution containing 125 mM NaCl, 2 mM KCl, 3 mM CaCl$_2$, 1 mM MgCl$_2$, 10 mM HEPES, 30 mM glucose and the synaptic blockers 0.01 mM NBQX and 0.01 mM GABAzine, pH adjusted to 7.3 with NaOH, osmolality adjusted to 300 mOsm with sucrose. Borosilicate glass pipette (Warner Instruments) with an outer diameter of 1.2 mm and a wall thickness of 0.255 mm was pulled to a resistance of 5–10 MΩ with a Flaming/Brown micropipette puller (P-97, Sutter Instruments) and filled with an internal solution containing 155 mM K-gluconate, 8 mM NaCl, 0.1 mM CaCl$_2$, 0.6 mM MgCl$_2$, 10 mM HEPES, 4 mM Mg-ATP and 0.4 mM Na-GTP, pH adjusted to 7.3 with KOH, osmolality adjusted to

298 mOsm with sucrose. Fluorescence imaging was performed on an upright fluorescence microscope (Olympus BX51WI) equipped with a SPECTRA X Light Engine and a ×40 NA 1.15 objective lens. Whole-cell patch-clamp recordings were performed using Multiclamp 700B amplifiers, a Digidata 1550 A digitizer and a personal computer running pClamp (Molecular Devices).

For Extended Data Fig. 2f,g (synaptic transmission properties), cultured neurons were transduced with AAVs at DIV4 for electrophysiology at DIV7, DIV9 and DIV11. Neurons were placed in a bath chamber and perfused with an approximately 34 °C Tyrode solution containing 125 mM NaCl, 30 mM glucose, 3 mM KCl, 2 mM $CaCl_2$, 2 mM $MgCl_2$ and 10 mM HEPES, pH adjusted to 7.3 with NaOH, osmolality adjusted to 299–300 mOsm with sucrose. Recording pipettes were pulled on a P-1000 microelectrode puller (Sutter Instrument) from borosilicate glass capillaries (outer diameter = 1.5 mm and inner diameter = 0.86 mm). Pipette resistances ranged from 5 to 8 MΩ and were filled with an internal solution containing 115 mM K-gluconate, 7.73 mM KCl, 0.5 mM EGTA, 10 mM HEPES, 10 mM $Na_2$-phosphocreatine, 4 mM MgATP, 0.3 mM NaGTP and 0.1% biocytin (w/v), pH adjusted to 7.3 with KOH, osmolality adjusted to 290 mOsm with sucrose. Cells were visualized using a Nikon FN1 microscope equipped with Dodt gradient contrast imaging, a 0.8 NA ×40 water-immersion objective and a Retiga Electro CCD camera (Teledyne). Whole-cell voltage-clamp recordings were made with a Double Integrated Patch Clamp Amplifier (Double IPA, Sutter Instrument). Data were sampled at 50 kHz and low-pass filtered at 10 kHz, and acquisition and analyses were performed using SutterPatch (Sutter Instrument). Recordings with series resistance of more than 20 MΩ were discarded. Spontaneous excitatory postsynaptic currents were recorded with cells held at the calculated $Cl^-$ reversal potential (−76 mV) to isolate excitatory events. Series resistance and whole-cell capacitance were compensated. Excitatory postsynaptic currents were detected and analysed using the template-matching event-detection algorithm in SutterPatch.

### Calcium imaging and analysis of spontaneous network dynamics in cultured neurons
For Extended Data Fig. 2h–m, cultured neurons were transduced with AAVs at DIV4 by adding the AAV stocks into neuronal culture medium. Calcium imaging was performed at DIV15. Coverslips were gently washed twice in washing solution (25 mM HEPES, 140 mM NaCl, 5 mM KCl, 1 mM $MgCl_2$, 10 mM glucose, 2 mM $CaCl_2$ and 10 μM glycine, pH adjusted to 7.2 with NaOH, pre-warmed to 37 °C), and placed into a glass-bottom 24-well plate containing imaging solution (25 mM HEPES, 140 mM NaCl, 8 mM KCl, 1 mM $MgCl_2$, 10 mM glucose, 4 mM $CaCl_2$ and 10 μM glycine, pH adjusted to 7.2 with NaOH, pre-warmed to 37 °C). GCaMP6f fluorescence was recorded for 3 min at a frame rate of 8.38 frames per second using a ×10 objective. One to four fields with non-overlapping soma were imaged per coverslip. Image analysis and signal processing were performed in MATLAB (MathWorks).

### CytoTape-vivo structural monomer design strategy and pipeline
CytoTape-vivo structure monomer design uses CytoTape structure monomer as a template. The design strategy optimized the linker between 1POK(E239Y, L349K) and MBP by tuning their sequence lengths. Design candidates (the constructs for CytoTape-vivo designs are listed in Supplementary Table 8) were screened in living mouse brain. The *UbC* promoter was used to drive the steady expression of candidate protein monomers.

### Mouse surgery
C57BL/6J mice (2–5 months of age; male; Jackson Laboratory) were anaesthetized with 5% isoflurane during induction and placed on a heating pad in a stereotactic frame (RWD Instruments) with 1.5–2% isoflurane throughout surgery to maintain deep anaesthesia. Ophthalmic ointment was applied to both eyes. Hair was removed with

a hair removal cream and the surgical site was cleaned with ethanol and betadine. Following this, an incision was made to expose the skull. Craniotomy was performed by drilling through the skull above the injection site using a 0.5-mm diameter drill bit. The AAV mixture was injected into the dorsal CA1 (anteroposterior −2.0 mm from bregma; mediolateral ±1.5 mm; and dorsolateral −1.5 mm from brain surface) using a pulled glass capillary with a pressure microinjector (RWD) at a rate of 100 nl min$^{-1}$. Following injection, the needle remained at the target site for 15 min to facilitate AAV diffusion into brain tissue. Mice received 5 mg kg$^{-1}$ carprofen intraperitoneally following surgery and were placed on a heating pad for recovery.

For Fig. 5 and Extended Data Fig. 8, the AAV mixture was prepared by mixing the AAV stocks (serotype AAV9; UNC NeuroTools) at the following parameters per mouse (the final viral genome copy (GC) of the structural monomer (CytoTape-vivo-HA) AAV in the mixture was 10-fold and 100-fold greater than those for the timestamp AAV and the signal monomer AAV, respectively, to ensure that the fibres were predominantly composed of structural monomers): AAV9-*UbC*-CytoTape-vivo-HA (titre of $7.54 \times 10^{12}$ GC ml$^{-1}$, volume of 0.5 μl and final viral genome copies of $3.8 \times 10^9$ GC), AAV9-*Fos*-CytoTape-V5 (titre of $1.28 \times 10^{12}$ GC ml$^{-1}$, volume of 0.03 μl and final viral genome copies of $3.8 \times 10^7$ GC), AAV9-*hSyn*-rtTA3G (titre of $3.56 \times 10^{12}$ GC ml$^{-1}$, volume of 0.27 μl and final viral genome copies of $1.0 \times 10^9$ GC) and AAV9-*TRE3G*-XRI-FLAG (titre of $2.67 \times 10^{12}$ GC ml$^{-1}$, volume of 0.14 μl and final viral genome copies of $3.8 \times 10^8$ GC).

For Extended Data Figs. 8a–d–10, the following AAVs were injected individually in separate mice: AAV9-*UbC*-XRI-HA (titre of $1.58 \times 10^{13}$ GC ml$^{-1}$, volume of 0.3 μl and final viral genome copies of $4.74 \times 10^9$ GC), AAV9-*UbC*-CytoTape-HA (titre of $3.06 \times 10^{13}$ GC ml$^{-1}$, volume of 0.5 μl and final viral genome copies of $1.5 \times 10^{10}$ GC) and AAV9-*UbC*-CytoTape-vivo-HA (titre of $7.54 \times 10^{12}$ GC ml$^{-1}$, volume of 0.5 μl and final viral genome copies of $3.8 \times 10^9$ GC).

### Seizure induction in mice
Kainic acid (Sigma) was dissolved in sterile saline (Thermo Fisher) at 3.75 mg ml$^{-1}$ and administered intraperitoneally to mice at 15 mg kg$^{-1}$. Following injection, mice were continuously monitored in the home cage for 3 h to assess seizure severity using the Racine scale. Mice were then returned to the housing room and monitored daily for recovery. For the data shown in Fig. 5 and Extended Data Fig. 8g–i, a level 3 seizure was observed for the mouse fixed at day 11, and level 5 seizures were observed for the mice fixed at days 14 and 18, respectively. After monitoring, mice were then returned to the housing room and monitored daily for recovery.

### Dox oral delivery in mice
Dox hyclate (Sigma) was dissolved in standard drinking water at 1 mg ml$^{-1}$ and supplemented with 5% (w/v) sucrose. Standard drinking water in home cages was replaced with this Dox water. Following the treatment period, Dox water was replaced with standard drinking water to terminate Dox delivery.

### Histology
Mice were perfused transcranially with 1× PBS followed by 4% paraformaldehyde (Electron Microscopy Sciences) in 1× PBS. The brain was gently extracted from the skull and post-fixed in 4% paraformaldehyde in 1× PBS overnight at 4 °C. The brain was then incubated in 100 mM glycine in 1× PBS for 1 h at room temperature, and then the brain was transferred into 1× PBS and stored at 4 °C until slicing. The brain was sectioned coronally at 45-μm thickness using a vibratome (Leica VT1000 S), and then stored in 1× PBS at 4 °C until immunofluorescence staining.

### Contextual fear conditioning
CytoTape-vivo-expressing mice received bilateral injections of CytoTape-vivo AAV in the hippocampi. The control group was wild-type

mice of the same age with no AAV injection. Mice were handled for approximately 1 min per day over the course of at least 1 week. On at least 4 of those days, they were transported to the testing room and handled there, whereas on the remaining days, handling took place in the vivarium. During contextual fear conditioning (10 days after AAV injection of CytoTape-vivo), mice were placed in a sound-attenuating chamber (Med Associates; W × H × D, 32 cm × 25 cm × 25 cm) equipped with a stainless-steel grid floor for foot-shock delivery and transparent polycarbonate front and top panels. After a 2-min baseline period, each mouse received three 2-s foot shocks at an intensity of 1.5 mA, separated by 1-min intervals. Thirty seconds after the final shock, mice were removed and returned to their home cages. Two days later, the mice were re-exposed to the same conditioning context for 5 min to assess memory retrieval. Freezing behaviour was analysed and quantified through open-source video analysis pipeline: ezTrack (https://github.com/denisecailab/ezTrack; Extended Data Fig. 10b,c).

## Open field test

The open field test was conducted in a 40 × 40 × 30 cm arena (arena 1) made of white acrylic (Extended Data Fig. 10d–j). CytoTape-vivo-expressing mice received bilateral injections of CytoTape-vivo AAV in the hippocampi 10 days before the open field test. The control group was wild-type mice of the same age with no AAV injection. Before the experiment, mice were handled by hand for 5 min daily and habituated to the procedure room for 30 min on 3 consecutive days. During testing, each mouse was placed individually in the centre of arena 1 and allowed to freely explore for 10 min while being video recorded. The centre zone was defined as the central 36% of the total area. The environment was kept quiet and dimly lit, and the arena was cleaned with 70% ethanol between trials. Data were analysed with Python and LabGym (https://github.com/umyelab/LabGym).

## Novel object recognition

The novel object recognition test was conducted in a 30 × 30 × 40 cm arena (arena 2) made of white acrylic (Extended Data Fig. 10k–n). Following the open field test, the same cohort of mice underwent a four-day rest period in their home cages before beginning habituation on day 15 (15 days after AAV injection of CytoTape-vivo) for the downstream novel object recognition test. For habituation, each mouse was placed in the empty arena 2 for 10 min daily for 2 days (day 15 and day 16). In the training session on day 17, two identical objects (made of toy blocks) were placed along the diagonal, near the centre of arena 2, and each mouse was released from a corner equidistant from both objects to explore for 10 min. After training, mice were returned to their home cages for 1 h, immediately followed by a testing session in which one object was replaced with a novel object of similar size and texture but different shape and colour. The mouse was reintroduced to arena 2 and allowed to explore both objects for 5 min. Object exploration was defined as approaching, sniffing or touching the object within 2.5 cm of its surface. Only the first 5 min of recording in the training session were used for analysis. Only mice with a total exploration time of more than 20 s in the first 5 min were included in the analysis. The test room was kept quiet and dimly lit, and the arena was cleaned with 70% ethanol between trials. Data were analysed with Python and LabGym. The discrimination index was calculated as (time_NO − time_FO)/(time_NO + time_FO), where time_NO was time exploring the novel object and time_FO was time exploring the familiar object.

## Immunofluorescence of brain tissue

Brain tissue sections were blocked overnight at 4 °C in MAXBlock blocking medium, followed by four washes for 30 min each at room temperature in MAXWash washing medium. Next, tissues were incubated with primary antibodies in MAXbind staining medium overnight at 4 °C, and then washed in MAXWash washing medium four times for 30 min each at room temperature. Next, tissues were incubated with fluorescently labelled secondary antibodies and NeuroTrace blue fluorescent Nissl stain (Invitrogen) in MAXbind staining medium overnight at 4 °C, and then washed in MAXWash washing medium four times for 15 min each at room temperature. Tissues were then stored in 1× PBS at 4 °C until imaging.

## Confocal images

For Fig. 1f, the confocal images of cultured primary mouse hippocampal neurons expressing XRI (top row) and CytoTape (bottom row) fused to the HA tag were taken after fixation at different time points (7, 14 and 18 days after calcium phosphate transfection) and then Nissl staining and immunostaining against the HA tag. For Fig. 2c,d, low-magnification images were taken of CytoTape labelled with $JF_{585}$ and $JF_{635}$ in HEK cells, which were taken after fixation on day 3, and Nissl staining and immunostaining against the HA tag. For Fig. 2h,j,l, timestamps for different temporal scales with varying temporal resolutions; for the second row, images were of CytoTape at timescales of 5, 9 and 15 days. For Fig. 2p, the confocal images were taken of timestamped fibres labelled with different dye-switching intervals in HEK cells. The confocal images were taken using a spinning disk confocal microscope with a ×60 objective. For Fig. 2s, the confocal images were taken of timestamped fibres labelled with different dye-switching intervals for weeks-long recording in cultured neurons. For Fig. 3c,h,m, images were captured after fixation, Nissl staining and immunostaining for V5 tag. For the first row, a composite image of all imaged channels showing the CytoTape in the cell; for the second row, the image was of the timestamp channels; for the third row, the image was of the *Fos* signal channel. For Fig. 3r, images were captured after fixation, Nissl staining and immunostaining for V5 and OLLAS tags. For the first row, a composite image of all imaged channels showing the CytoTape in the cell; for the second row, the image was of the timestamp channels; for the third row, the image was of the *Fos* signal channel; and for fourth row, the image was of the *Arc* signal channel. For Fig. 3w, images were captured after three rounds of processing: (1), imaging of JF dyes and the HA tag followed by fixation and immunostaining for HA tag; (2), photobleaching of JF dye signals followed by immunostaining for HA, Etag, OLLAS, V5 and FLAG tags (with the HA tag used for alignment of CytoTape between the two rounds of immunostaining); and (3), Nissl staining. HEK cells were stimulated with 50 µM FSK for 1 h on day 3. For the first row (left), CytoTape within the cell; for the second row (left), $JF_{635}$ dye; for the third row (left), $JF_{503}$ dye; for the fourth row (left), $JF_{585}$ dye; and for the fifth row (left), merge of $JF_{635}$, $JF_{503}$ and $JF_{585}$ dyes. For Fig. 4c, images were captured after fixation, Nissl staining and immunostaining for HA and OLLAS tags. For the left panel, the CytoTape in the HEK cells. For the first row, timestamps; for the second row, CREB activity and FOS activity. For Fig. 5b, the preserved brains were sectioned coronally at 40-µm thickness and stained with anti-HA, anti-FLAG, anti-V5 and Nissl stain. For the left panel in Fig. 5d, a low-magnification multi-channel composite image was taken of a representative brain slice; for the middle panel, maximum intensity projection was taken of a 3.2-µm-thick volume in the region of interest in the dentate gyrus. Some CytoTape-vivo fibres extend beyond this volume in the *Z* (depth) dimension and are therefore not fully visible in these two-dimensional projections. For the right panel, representative confocal images that show *Fos*-promoter driven gene expression and Dox-dependent signal monomer expression in dentate gyrus neurons were taken.

For Extended Data Fig. 1d, confocal images were taken of cultured mouse hippocampal neurons expressing XRI (middle row) and CytoTape (bottom row) fused to the HA tag and the V5 tag, taken after fixation on day 15, followed by Nissl staining and immunostaining against the HA and V5 tags. For Extended Data Fig. 1e, confocal images were taken of cultured mouse hippocampal neurons expressing CytoTape with the HA and V5 tags, taken after fixation on day 7, followed by Nissl

staining and immunostaining against the HA and V5 tags. For Extended Data Fig. 1f, confocal images were taken of XRI and CytoTape in HEK and HeLa cells after fixation, Nissl staining and immunostaining against the HA tag. For Extended Data Fig. 3b–e,h–l, confocal images were taken of CytoTape in HEK cells after fixation, Nissl staining and immunostaining against the physiological markers. For Extended Data Fig. 3f,g, confocal images were taken of CytoTape in live HEK cells stained with fluorescent dyes. For Extended Data Fig. 4d,f,h,j,l, images were captured after fixation, Nissl staining and immunostaining for HA and V5 tags; enlarged views of the CytoTape in the top-row panels are shown in the three rows of rectangular panels below. For Extended Data Figs. 5 and 6, confocal images were taken of pCREB and FOS protein reported by eGFP and mRuby3 or protein immunostaining. For Extended Data Fig. 5, representative confocal images were taken of brain slices from adult mice expressing CytoTape-vivo in the hippocampus in the right cerebral hemisphere following AAV injection. Three-month-old mice were injected with AAV9-CytoTape-vivo-HA at the CA1 region in the hippocampus of the right cerebral hemisphere. When the mice reached 14 days after AAV injection, they were perfused with 4% PFA, and brains were sliced coronally at 40 μm in 1X PBS, and stained with antibodies to cellular and synaptic markers (Extended Data Fig. 9) and to the HA tag to label CytoTape-vivo fibres, together with DAPI to label nuclei. Staining intensities of cellular and synaptic markers in the hippocampus were imaged volumetrically using a ×40 objective on a spinning disk confocal microscope, with identical imaging conditions, measured in ImageJ as the averaged fluorescent intensities of the fluorescent secondary antibodies over imaged fields of view (400 μm × 400 μm × 40 μm for each field of view), and compared between the left hemisphere and the right hemisphere. For Extended Data Fig. 8a, a confocal image was taken of CytoTape fibres in dentate gyrus neurons of the mouse brain 14 days after AAV injection, taken after fixation on day 14, followed by Nissl staining and immunostaining against the HA tag. For Extended Data Fig. 8g, representative confocal images were taken of CytoTape-vivo showing *Fos*-promoter-driven gene expression and Dox-dependent signal monomer expression in CA1 neurons. Images were captured after fixation, Nissl staining and immunostaining for HA, FLAG and V5 tags.

## Software for image analysis
Image analysis was performed in ImageJ (National Institutes of Health), napari[54] (napari contributors) and Python.

## Fluorescent intensity readout and geometrical measurement of fibres in images
To enable automated readout from CytoTape fibres in cultured cells, we developed a custom computational analysis pipeline, Tape Analyzer, which is described below. Three-dimensional fibres were first segmented with an adaptive thresholding approach, and fibre centre lines were extracted using the intrinsic function from the scikit-image (Python) package. Sequentially, fluorescence intensity profiles were obtained by averaging fluorescence values within cylindrical sampling regions positioned along the centre line. Each cylinder had a circular cross-section oriented perpendicular to the local direction vector of the centre line with 0.06-μm intervals. In parallel, key geometrical parameters, including fibre length, thickness and curvature, were calculated based on the 3D morphology of segmented masks of fibres and were used for statistical analysis in Fig. 1. Reliability of the extracted fluorescent intensity profile was benchmarked against ground-truth datasets derived from manual measurements in ImageJ. Details and source code of Tape Analyzer have been provided on GitHub (https://github.com/LinghuLab/TapeAnalyzer).

## Fluorescent intensity profiles analysis
Fluorescence intensity profiles along the fibre were analysed as previously described[38].

## Time recovery with HaloTag–dye-based timestamp
For each fibre, we divided it into two segments based on an optimal split point as previously described[38]. This point was determined by iterating through hypothetical split locations within a window spanning 20% of the total fibre length around its geometrical centre point and selecting the one that maximized the Pearson correlation between the left and right halves. We reasoned that this optimal split best balances the trade-off between the inherent symmetry of CytoTape elongation and the variability in protein assembly introduced by the complex cellular environment.

For each half fibre (representing a complete time axis), we plotted the fluorescent intensities from different HaloTag-based timestamp channels, and manually identified critical points for dye colour switching that represented real-time events along the fibre, serving as a bridge between fibre growth and real-time events. These critical points included the onset rise point of the second dye colour relative to the first dye colour for the first timestamp (reflecting when the second dye was actually added), the onset rise point of the next dye colour (if present) relative to the second colour for the second timestamp (reflecting when the subsequent dye was actually added), and the fibre end point (reflecting the actual cell fixation time). The normalized position along the fibre (dividing the distance from the initial point by its total length) was also calculated for each point, and we termed it as a fraction of fibre length.

Time axis recovery was achieved by fitting a polynomial curve between the the actual time points when dyes were added or cells were fixed (for example, day 2, day 5.5 and day 11) and their corresponding fraction of fibre length values (for example, 0.5, 0.7 and 1). If there was only one dye switch, resulting in two timestamps (the dye switch and the cell fixation time corresponding to the fibre termini), a linear polynomial (first-degree) fitting was performed. If there were two dye switches, resulting in three timestamps (dye switch 1, dye switch 2 and the cell fixation time), a quadratic polynomial (second-degree) fitting or power fitting was used to best characterize the fibre growth kinetics, which has been experimentally characterized in Fig. 2, right panel. After curve fitting, the resulting curve served as a transfer function between the spatial axis along the fibre and the real-world time axis. Fluorescent intensities along the fibres from the signal monomer channel (or channels) were initially plotted along each of the half fibres with respect to the spatial axis representing the fraction of the fibre length. This fractional fibre position was then converted into the real-world time axis using the fitted curve as the transfer function, thereby establishing the time axis for the recorded signal (or signals).

For Fig. 2e, statistical analysis (left panel) was determined of fluorescence line profiles from the experiments described in Fig. 2b, and interpolation was determined between the spatial axis along the fibre and the time axis (right panel), using the timestamps from the left panel. Time axis recovery via linear curve fitting of timestamps and fixation point is shown. Each raw trace on the left was normalized to its peak to show relative changes before averaging. For Fig. 2i,k,m, top panel, statistical analysis was determined of the fluorescence line profiles from the experiments described in Fig. 2h,l,n. For the bottom panel, interpolation was determined between the spatial axis along the fibre and the time axis using the timestamps from the top panel. Time axis recovery via B-spline curve fitting of timestamps and fixation point is shown. Each raw trace was normalized to its peak to show relative changes before averaging. The normalized position along CytoTape was calculated as dividing the distance along the half fibre starting from the optimal split point by the total length of this half fibre. For Fig. 3d, top panel, the *Fos* signal (from Fig. 3c) relative change from baseline was plotted against recovered time (interpolated with timestamps and fixation time) after calcium phosphate transfection. For the bottom panel, statistical analysis was determined of the *Fos* signal relative change from baseline plotted against recovered time after calcium phosphate transfection. For Fig. 3i, left panel, the *Fos* signal relative

change from baseline was plotted against recovered time after calcium phosphate transfection; for the right panel, statistical analysis was determined for the *Fos* signal relative change from baseline plotted against recovered time after calcium phosphate transfection. FSK (15 μM) was added to the culture on days 3 and 5, with an incubation time of 1 h. For Fig. 3n, left panel, the *Fos* signal relative change from baseline was plotted against recovered time after calcium phosphate transfection; for the right panel, statistical analysis was determined of the *Fos* signal relative change from baseline plotted against recovered time after calcium phosphate transfection. FSK (15 μM) was added to the culture on days 3 and 15, with an incubation time of 1 h for the first stimulation and 1.5 h for the second stimulation. For Fig. 3s, top panel, the *Fos* and *Arc* signals (from Fig. 3r) relative change from baseline were plotted against recovered time (interpolated with timestamps and fixation time) after calcium phosphate transfection; for the bottom panel, statistical analysis was determined of the *Fos* and *Arc* signals relative change from baseline plotted against recovered time after calcium phosphate transfection. For Fig. 3x, HA (left, first row), OLLAS (right, first row), Etag (left, second row), V5 (right, second row) and FLAG (left, third row) signal relative change from baseline was plotted against recovered time (interpolated with timestamps and fixation time). For the right, third row, comparison was of *Fos*, *Egr1*, *Arc*, CREB activity and NPAS4 activity. For Fig. 4f,g, type 1 (panel f) and type 2 (panel g), single traces of pCREB (left panel) and FOS (middle panel) signals relative change from baseline were plotted against recovered time (interpolated with timestamps and fixation time) in two responsive types; for the right panel, the averaged curve of pCREB and FOS signals relative change from baseline were plotted against recovered time after calcium phosphate transfection.

For Extended Data Figs. 5a and 6c,d, the averaged curve of pCREB and FOS signals relative change from baseline were plotted against recovered time after TransIT-X2 transfection. For Extended Data Fig. 7c, representative single traces of *Arc* and *Egr1* signals relative change from baseline were plotted against recovered time (interpolated with timestamps and fixation time) after calcium phosphate transfection in three responsive types under KCl stimulation. For Extended Data Fig. 7d, the averaged *Arc* and *Egr1* signals relative change from baseline were plotted against recovered time after calcium phosphate transfection. For Extended Data Fig. 7f, representative single traces of *Arc* and *Egr1* signals relative change from baseline were plotted against recovered time (interpolated with timestamps and fixation time) after calcium phosphate transfection in three responsive types under FSK stimulation. For Extended Data Fig. 7g, the averaged *Arc* and *Egr1* signals relative change from baseline were plotted against recovered time after calcium phosphate transfection.

### Computational readout of protein tape recordings and cell boundaries and locations from microscopy images of mouse brain tissue

We developed Tape Reader, a custom high-throughput image analysis platform with the following components. For cell segmentation: we utilized the Segment Anything for Microscopy (μSAM) model to perform automated cell segmentation of the Nissl-stained neuron channel. Although typically used for interactive segmentation tasks, we leveraged the automatic instance segmentation module to generate high-quality soma segmentations. We used pretrained weights trained on light microscopy imagery using the base vision transformer architecture. For fibre segmentation, we used the U-Net backbone architecture from the PyTorch Connectomics library[55] to obtain instance segmentations from the structure monomer channel. Our model predicts three output channels: a binary segmentation mask, a contour map of instances and the Skeleton-Aware Distance Transform[56]. We subsequently applied an adapted marker-based watershed algorithm to generate instance segmentations from these intermediate representations. Watershed hyperparameters were optimized

using scikit-optimized Gaussian processes, minimizing the adapted Rand error metric on hand-annotated validation volumes. The training dataset comprised six image volumes from mouse brain tissue with manually annotated fibre structures in 3D. We applied multiple quality control criteria to assess whether predicted fibre segments accurately reflected fibre morphology and anatomical positioning. Initially, segments were filtered by geodesic length, excluding all fibres shorter than 8 μm. To ensure morphological accuracy of segmentation, segmented structures with non-fibre-like morphology were removed through principal component analysis. Specifically, we calculated the explained variance of the first principal component of the skeleton of the segmented structure and excluded segmented structures with values below 0.8. In addition, fibres located outside the soma regions detected by Nissl staining were excluded from further analysis. For skeletonization, each predicted fibre segment was converted to a point cloud representation, and principal component analysis was applied to determine optimal fibre alignment axes. A cubic spline was fitted through the central 80% of points to generate a representative centre line, which was subsequently sampled at 1,000 equidistant points. This centre line was extrapolated by up to 20% of the geodesic length in both directions until local minima in the structure monomer channel were identified. For signal extraction, continuous signals were extracted from image volumes with discretized voxels through trilinear interpolation of intensity values at each skeleton point across all imaging channels. To determine optimal skeleton midpoints to split the fibre into two halves for further analysis, we used the Optuna TPESampler hyperparameter optimization framework to maximize Pearson correlation coefficients between signal halves in the Dox-dependent monomer channel. Candidate centre points were constrained to a 5% radius interval around the geometric centre. The optimal centre point was defined as the position yielding maximum Pearson correlation between the resulting signal halves. Signal relative change from the baseline was performed by calculating the mean intensity within a 5% radius interval around the centre point as the baseline signal value, followed by subtraction and division of all signal intensities by this baseline signal value. Each fibre was assigned to a cell by identifying the segmented cell boundary that contained all or the majority of its centre line points with the cell segmentation approach described above. In 14-day and 18-day experiments (Fig. 5 and Extended Data Fig. 8), if there was more than one fibre in a cell, the longest fibre was used for subsequent analysis. Figure 5b shows examples of computational segmentation of neuronal soma and CytoTape-vivo fibres from high-magnification volumetric images in the dentate gyrus. Each colour represents an individual neuron or fibre. Details of the segmentation method are described in Methods. See Supplementary Video 1 for the full field of view. The source code of Tape Reader has been provided on GitHub (https://github.com/LinghuLab/TapeReader).

### Time recovery with TRE or Dox-based timestamp

To convert CytoTape-vivo recorded signals into temporally resolved representations, we leveraged the robust ON and OFF waveform features of the Dox-dependent (FLAG) signal as timestamps (Fig. 5 and Extended Data Fig. 8g–i). For each fibre in the 14-day experiment, the onset of the rise of the FLAG signal, the onset of the decay of the FLAG signal and the end of the FLAG signal were used as timestamps for days 10, 12 and 14, respectively. For each fibre in the 18-day experiment, the onset of the rise of the FLAG signal, the onset of the decay of the FLAG signal and the end of the FLAG signal were used as timestamps for days 10, 13 and 18, respectively. Space-to-time transformation was performed by curve fitting the location coordinates of the timestamps along the fibre and the corresponding real-world timepoints to a piecewise linear function directly connecting the timestamp points. The detailed experimental timelines are listed below.

For Fig. 5j, Dox-containing drinking water was administered on day 10, seizure induction was performed on day 10, Dox-containing drinking

water was replaced with regular water on day 12 and the brain was fixed on day 14. Recorded *Fos* signal monomer intensity relative change from baseline in dentate gyrus neurons under a kainic acid-induced seizure. The time axis was reconstructed for each fibre using Dox ON (onset of Dox signal rise), Dox OFF (onset of Dox signal decay) and fixation times shown in Fig. 5g as timestamps. For Fig. 5k, the experimental timeline for *Fos* signal monomer recording in vivo is shown. Dox-containing drinking water was administered on day 10, seizure induction was performed on day 10, Dox-containing drinking water was replaced with regular water on day 13 and the brain was fixed on day 18. For Fig. 5l, the recorded *Fos* signal monomer intensity relative change from baseline in CA1 neurons under kainic acid-induced seizure is shown. The time axis was reconstructed for each fibre using Dox ON (onset of Dox signal rise), Dox OFF (onset of Dox signal decay) and fixation times shown in Fig. 5k as timestamps.

For Extended Data Fig. 8f, the recorded relative change in Dox signal monomer intensity from baseline in dentate gyrus neurons during administration and subsequent replacement of Dox-containing water with regular water is shown; the dashed vertical lines indicate the time of Dox water administration (day 10, indicated by the dashed line) and its replacement with regular water (day 12, indicated by dashed line). For Extended Data Fig. 8h, the recorded relative change in Dox signal monomer intensity from baseline in CA1 neurons during administration and subsequent replacement of Dox-containing water with regular water is shown; the dashed vertical lines indicate the time of Dox water administration (day 10, indicated by dashed line) and its replacement with regular water (day 13, indicated by dashed line). For Extended Data Fig. 8i, the recorded *Fos* signal monomer intensity relative change from baseline in CA1 neurons under kainic acid-induced seizure is shown (enlarged views of Fig. 5l). The time axis was reconstructed for each fibre using Dox ON (onset of Dox signal rise), Dox OFF (onset of Dox signal decay) and fixation times shown in Fig. 5k as timestamps; the dashed vertical line indicates time of seizure induction.

## Statistical analysis and reproducibility

Statistical analysis was performed in Prism (GraphPad). All statistical tests were two-sided. Statistical parameters are described in the figure legends, and complete statistical tables are provided in Supplementary Table 11. Representative microscopy images are from at least three independent cell culture experiments or two independent in vivo experiments, all yielding similar results.

## Reporting summary

Further information on research design is available in the Nature Portfolio Reporting Summary linked to this article.

## Data availability

The mouse brain in vivo recording datasets generated and analysed in this study are available on Zenodo[57]. Supplementary Information includes additional details on the development and discussion of the CytoTape toolkit, Supplementary Tables 1–11, Supplementary Figs. 1–24, Supplementary Video 1 and Supplementary References. Source data are provided with this paper.

## Code availability

The code developed in this work for the in vivo image analysis, Tape Reader (v1.0), can be found on GitHub (https://github.com/LinghuLab/TapeReader). The code developed in this work for signal extraction and analysis of protein tape recordings, Tape Analyzer (v1.0), can be found on GitHub (https://github.com/LinghuLab/TapeAnalyzer). Video analysis of mouse behaviour was performed by ezTrack (v1.2; https://github.com/denisecailab/ezTrack) and LabGym (v2.9.0; https://github.com/umyelab/LabGym). The ProtSSN model for protein mutation prediction can be found on GitHub (https://github.com/tyang816/ProtSSN). The CPDiffusion model for protein sequence generation prediction can be found on GitHub (https://github.com/bzho3923/CPDiffusion).

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

**Acknowledgements** We thank D. Cable, D. Cai, W. Dong, J. Dudley, F. Levet, C. Ma, W. Wang, M. Widener, J. Yang and B. Ye for discussion. L.Z. thanks the Michigan Neuroscience Institute Postdoctoral Advancement Program and the eLife Ambassadors Program. M.L. and D.W. acknowledge NSF-IIS-2239688. E.S.B. acknowledges National Institutes of Health (NIH) 1R01EB024261, R01AG087374, 1R01AG070831 and R01MH122971, L. Yang and the Howard Hughes Medical Institute. C.L. is supported by the NIH Director's New Innovator Award (DP2MH140133), the Glenn Foundation for Medical Research and American Federation for Aging Research Grant Award for Junior Faculty, the Whitehall Foundation and the Klingenstein Fellowship Award in Neuroscience. C.L. and D.J.C. are supported by the Chan Zuckerberg Initiative Collaborative Pairs Pilot Project Award.

**Author contributions** L.Z., Y.Y., D.S. and C.L. conceived the concept of CytoTape and CytoTape-vivo and made high-level plans for this project. L.Z. performed the rational design of CytoTape and CytoTape-vivo monomers, in silico protein characterization and molecular dynamics simulations. B.Z. and L.Z. performed in silico protein point mutation and protein sequence generation with the help from P.L. L.Z. designed the HaloTag–dye-based timestamps, transcriptional recorders, multiplexing strategy and performed the cell culture experiments with the help and input from Y.Y., B.A., E.S.B. and C.L. D.S. designed the in vivo TRE–Dox-based timestamps, developed the in vivo experimental pipeline and performed the in vivo experiments with help and input from Y.Y., Y.H., L.Z., W.C.J., B.K., B.C., D.J.C. and C.L. L.Z. prepared the HEK293T and HeLa cells with help from Y.Y., E.G.K. and J.L. J.L. prepared the cultured neurons and glial cells. E.D.M.H., E.C.K. and M.T.R. performed the electrophysiology. L.Z., Y.Y., W.C.J., D.S. and B.K. performed the compatibility assays of cellular physiology and mouse behaviour. Y.Y., L.Z., J.K.A., M.L., M.G. and D.W. analysed the data with help from D.S., B.K., D.J.C. and C.L. S.P. helped with the heat shock experiment. J.K.A., M.L. and D.W. developed the algorithm for the in vivo data analysis with help from Y.Y., L.Z. and D.S. L.Z., Y.Y., D.S., B.Z. and C.L. interpreted the data and wrote the manuscript. C.L. supervised the project.

**Competing interests** L.Z. and C.L. declare that they applied for US provisional patent applications (63/801,605 and 63/852,288) based on the cell culture work presented in this paper. D.S., L.Z. and C.L. declare that they applied for a US provisional patent application (63/852,319) based on the in vivo work presented in this paper. The other authors declare no competing interests.

**Additional information**
**Correspondence and requests for materials** should be addressed to Changyang Linghu.

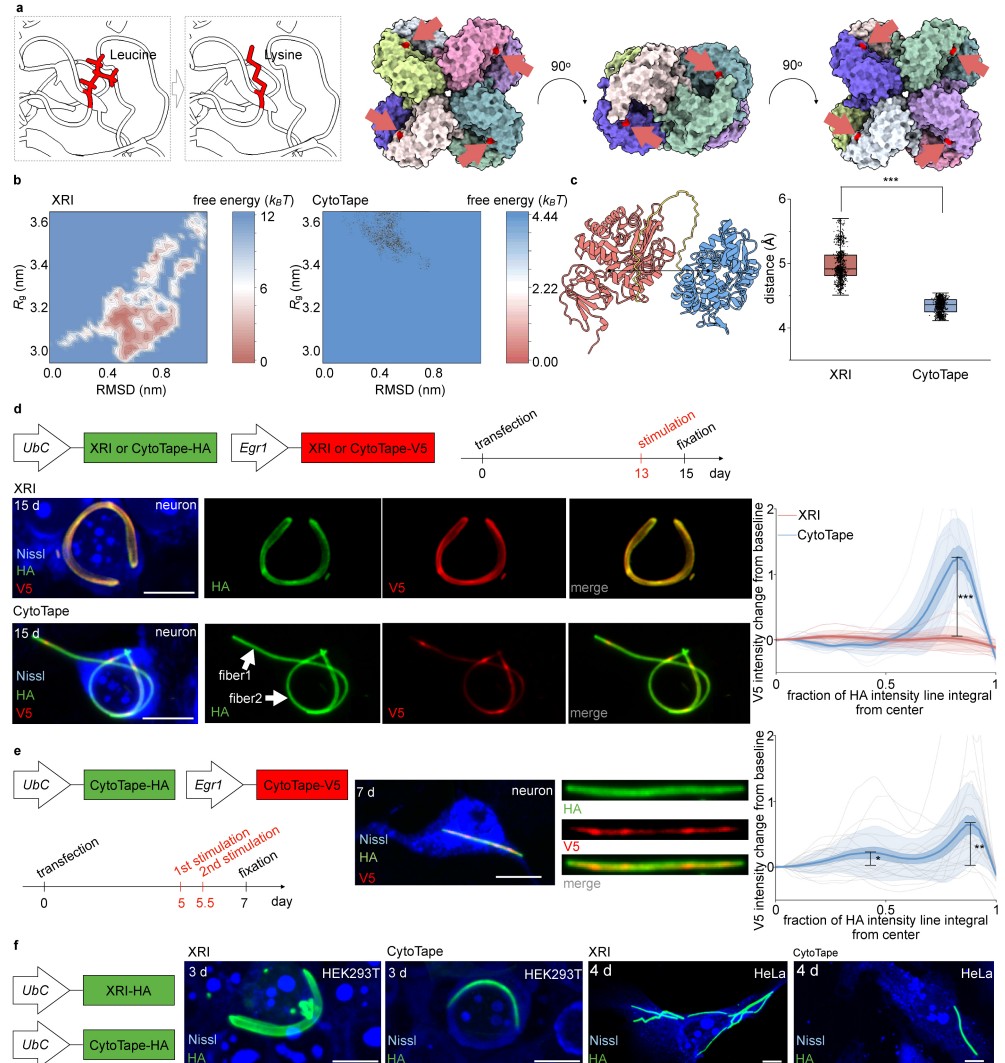

**Extended Data Fig. 1 | CytoTape forms thin and flexible fibers in live mammalian cells for weeks-long molecular recording. (a)** Position of the L349K mutation (highlighted in red and indicated by red arrows) in the octamer structure (PDB ID: 1POK). Each colored domain represents an identical monomer. (**b**) Free energy landscape of a single XRI or CytoTape protein monomer. (**c**) Distance between the centroid of the scaffold (1POK(E239Y) for XRI, 1POK(E239Y, L349K) for CytoTape) and the insulator (MBP for XRI, dMBP for CytoTape). ***, $P < 0.001$; two-sided Mann–Whitney U test. Box plot: middle line, median; box boundary, interquartile range; whiskers, minimum and maximum; black dots, individual data points. (**d**) Left, first row, schematic of the constructs transfected into cultured neurons and the experimental timeline. Confocal images of cultured neurons expressing XRI (middle row) and CytoTape (bottom row) fused to HA and V5 tags, taken after fixation on day 15. Neurons were stimulated with 55 mM KCl for 1 h. Right, V5 signal relative to baseline plotted against the fraction of the HA intensity line integral. XRI: n = 13 XRIs from 13 neurons, 2 cultures. CytoTape: n = 11 CytoTapes from 11 neurons, 2 cultures. ***, $P < 0.001$; two-sided Mann–Whitney U test for the peak amplitude of the V5 signal between XRI and CytoTape. Scale bars, 10 μm. (**e**) Left, schematic of the constructs transfected into cultured neuron and the experimental timeline. Middle, confocal images of cultured neurons expressing CytoTape with the HA and V5 tags, taken after fixation on day 7. Neurons were stimulated twice with 55 mM KCl for 0.5 h. The two stimulations are 12 h apart. Right, V5 signal relative to baseline plotted as a function of the fraction of the HA intensity line integral (n = 20 CytoTapes from 20 neurons, 2 cultures). *, $P < 0.05$; **, $P < 0.01$; two-sided Mann–Whitney U test between the amplitude of each of the two sequential peaks and the baseline of the V5 signal. Scale bars, 10 μm. Thick centerline, mean; darker boundary, s.e.m.; lighter boundary, s.d.; thin light lines, intensity profiles from all individual CytoTapes. (**f**) Confocal images of XRI and CytoTape expressed in HEK and HeLa cells. The constructs transfected into cells are shown in the left row. Scale bars, 10 μm.

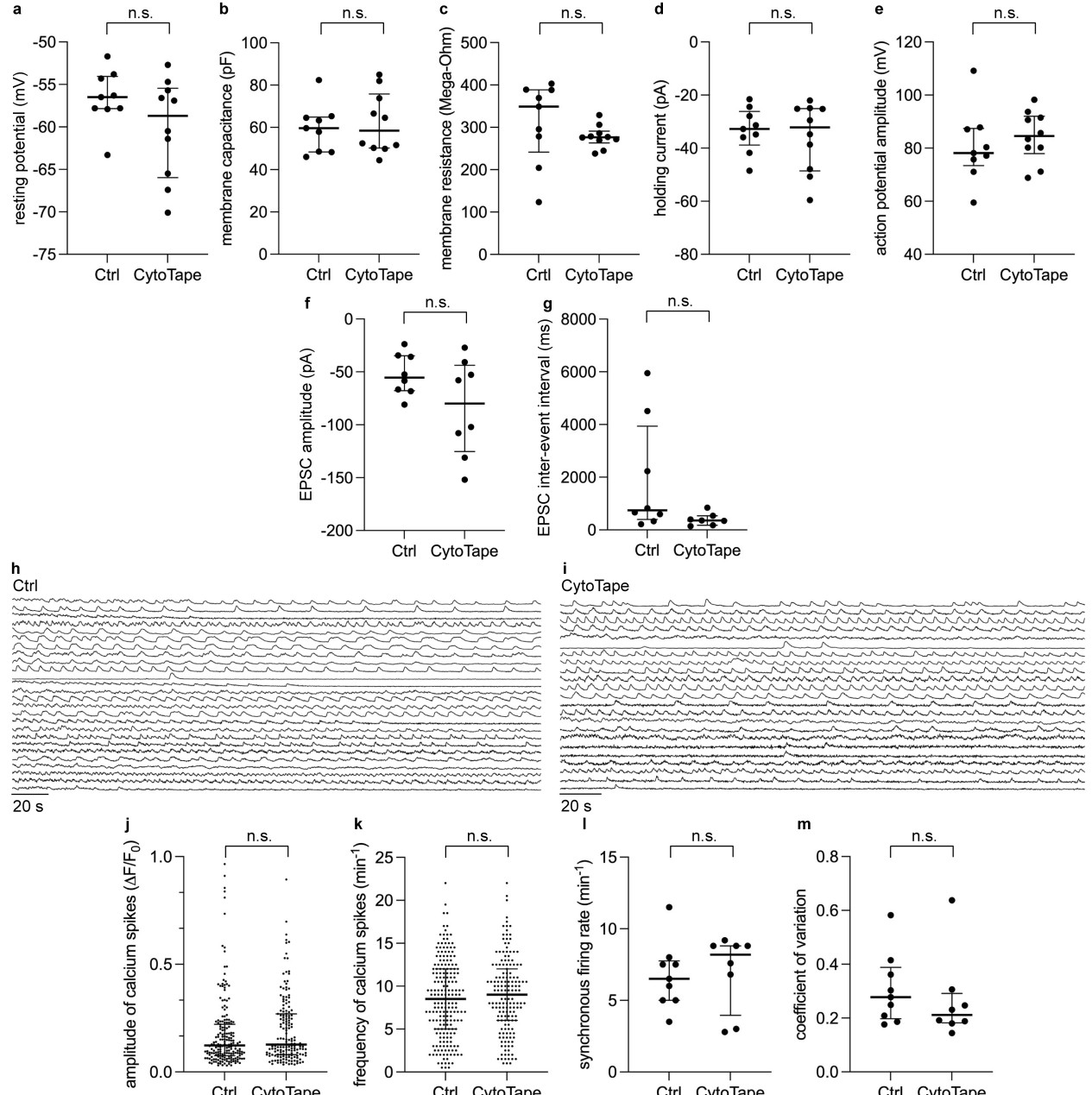

**Extended Data Fig. 2 | Electrophysiological properties, synaptic transmission, and calcium dynamics of cultured neurons expressing CytoTape.** Scatter plots of the electrophysiological properties of cultured neurons with or without CytoTape expression, in terms of (**a**) resting potential, (**b**) membrane capacitance, (**c**) membrane resistance, (**d**) holding current at −65 mV, and (**e**) action potential amplitude. Middle line, median; error bar, interquartile range; black dots, individual data points. n = 9 neurons for the control group (expressing *CAG*-mEGFP for 8-11 days); n = 10 neurons for the CytoTape group (expressing *UbC*-CytoTape-P2A-mEGFP for 8-11 days). Scatter plots of synaptic transmission properties, including (**f**) EPSC amplitude and (**g**) EPSC inter-event interval, of cultured neurons in the control (no gene delivery) and CytoTape (expressing *UbC*-CytoTape for 3-7 days) groups. Middle line,

mean; error bar, interquartile range; black dots, individual data points. **f:** n = 8 neurons for the control group; n = 8 neurons for the CytoTape group. **g:** n = 8 neurons for the control group; n = 7 neurons for the CytoTape group. Single traces of calcium dynamics of cultured neurons (**h**) with or (**i**) without CytoTape expression. Scatter plots of calcium imaging–based network activity metrics in control (no gene delivery; n = 201 neurons from 9 fields of view from 3 cultures) and CytoTape (expressing *UbC*-CytoTape for 11 days; n = 163 neurons from 8 fields of view from 3 cultures) groups, including (**j**) amplitude of calcium spikes, (**k**) frequency of calcium spikes, (**l**) synchronous firing rate, and (**m**) coefficient of variation. Middle line, median; error bars, interquartile range; dots, individual neurons or fields of view. Throughout this figure: n.s., not significant; two-sided Mann–Whitney U test.

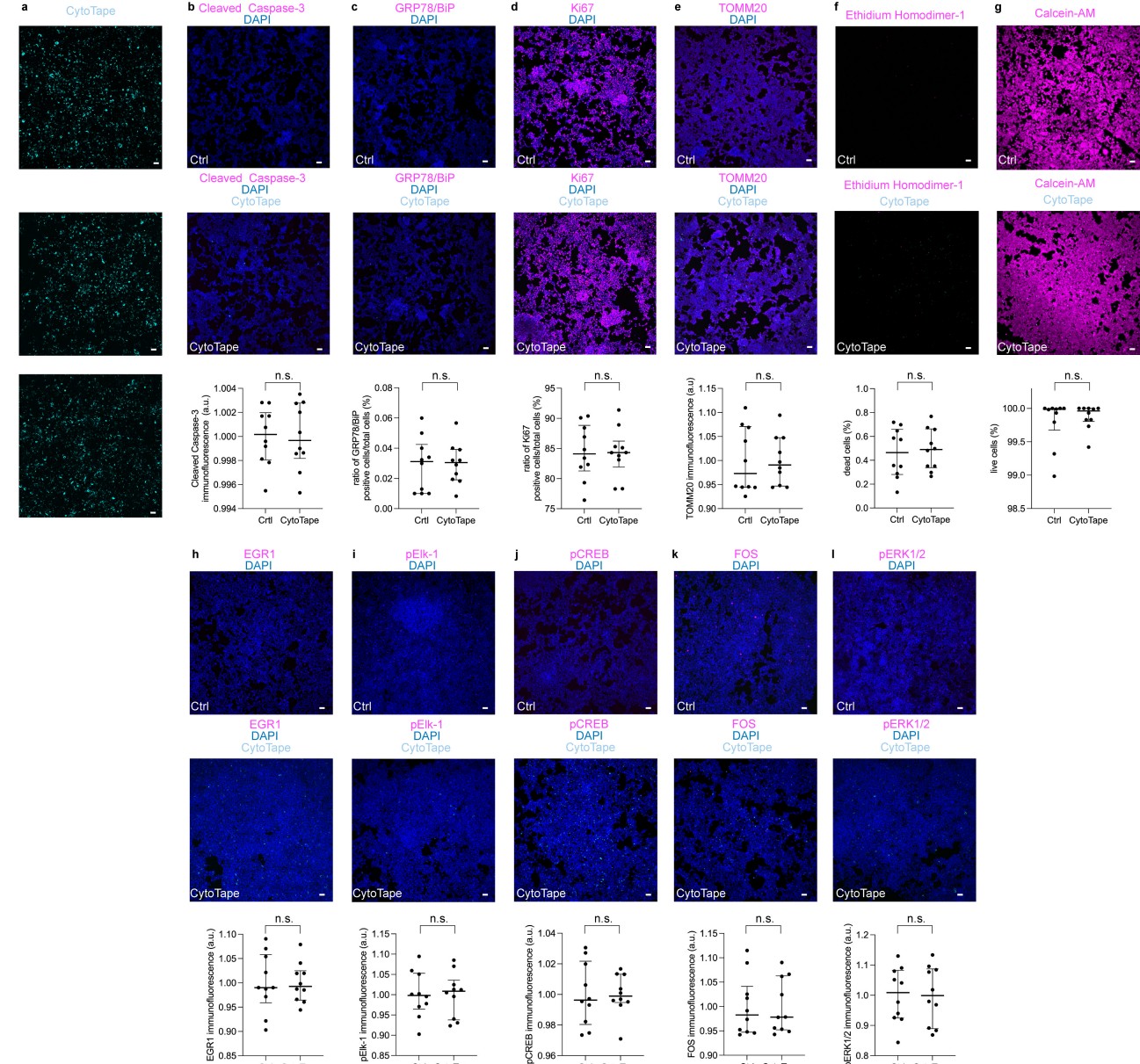

**Extended Data Fig. 3 | CytoTape does not alter cellular, physiological, transcriptional, or signalling markers in HEK cells.** (**a**) Representative confocal images of HEK cells expressing CyoTape for 3 days. Majority of HEK cells express CyoTape fibers. HEK cells were stained either by antibodies against cellular or physiological markers (**b-e**), fluorescent chemical dyes (**f,g**), or antibodies against transcriptional and signalling activity markers (**h-l**). (**b**,**e**,**h-l**) Mean fluorescence intensities were quantified in ImageJ as the averaged signal of the fluorescent secondary antibodies over imaged fields of view. Scale bars, 50 μm. (**b-l**, top) Representative confocal images of HEK cells transfected with pUC19 (Ctrl) or expressing CytoTape, stained with the indicated antibodies or imaged with the indicated fluorescent dyes together

with DAPI. Scale bars, 50 μm. (**b-l**, bottom) Scatter plots of the quantifications for each marker between control and CytoTape-expressing groups (n = 10 fields of view (FOVs) from 2 cultures). Middle line, median; whiskers, 95% CI; black dots, individual data points. n.s., not significant; two-sided Mann–Whitney U test. (**b**) Cleaved Caspase-3 (apoptotic marker). (**c**) GRP78/BiP (endoplasmic reticulum stress marker). (**d**) Ki-67 (cell proliferation marker). (**e**) TOMM20 (mitochondrial integrity marker). (**f**) Ethidium homodimer-1 (membrane permeability and DNA damage marker). (**g**) Calcein-AM (live cell marker). (**h**) EGR1 (EGR1 expression marker). (**i**) pElk-1 (Elk-1 activity marker). (**j**) pCREB (CREB activity marker). (**k**) FOS (FOS expression marker). (**l**) pERK1/2 (ERK activity marker).

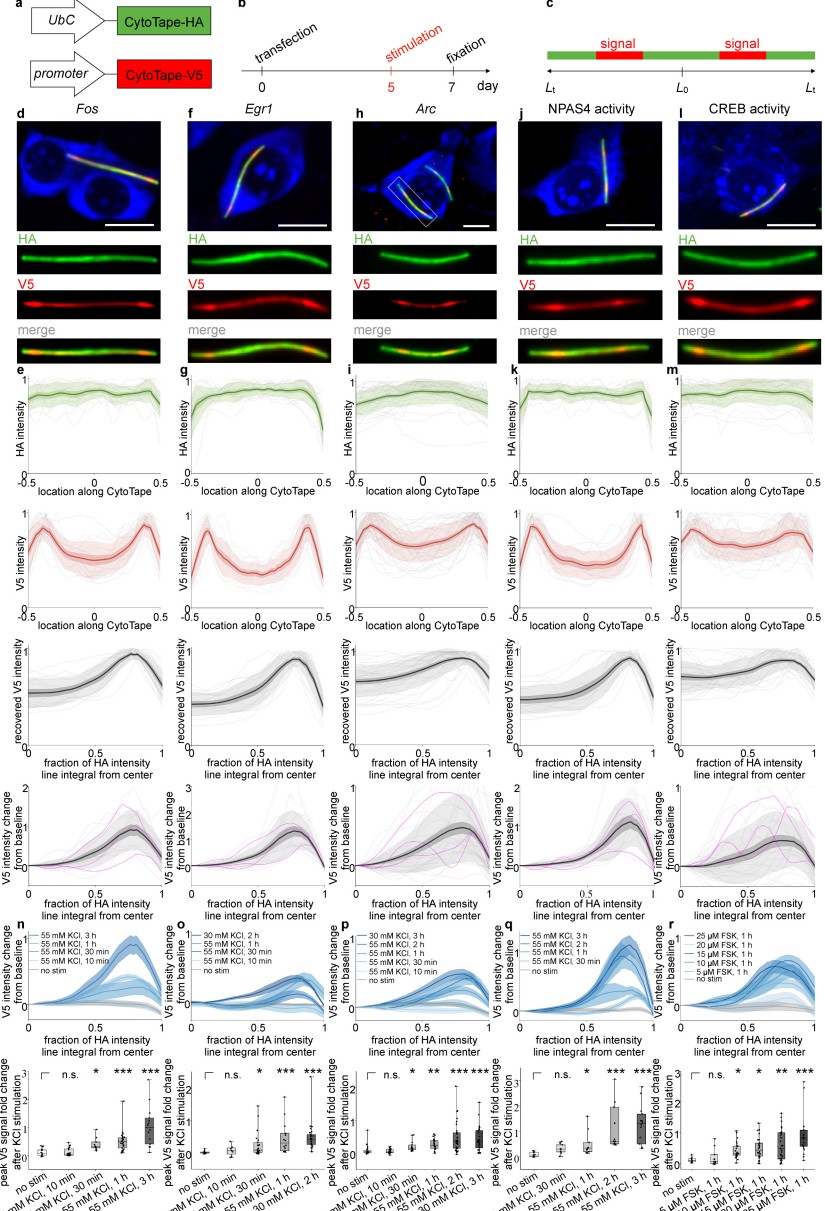

**Extended Data Fig. 4 | CytoTape enables modular design of transcriptional recorders for tracking gene regulation dynamics.** (**a**) Schematic of the constructs transfected into cultured neurons and (**b**) the experimental timeline. (**c**) Expected HA and V5 tags distributions along the CytoTape. (**d**,**f**,**h**,**j**,**l**) Representative confocal images of cultured neurons expressing constructs shown in **a**. Neurons were stimulated with 55 mM KCl for 3 h for *Fos*, 30 mM KCl for 2 h for *Egr1*, 55 mM KCl for 3 h for *Arc*, 55 mM KCl for 3 h for NPAS4 activity, and 10 µM FSK for 1 h for CREB activity. Enlarged views of the CytoTape in the top-row panels are shown in the three rows of rectangular panels below. Scale bars, 10 µm. (**e**,**g**,**i**,**k**,**m**) Profiles of HA and V5 signal intensities along CytoTape, based on the experiment in **b**. First row, HA intensity profile; second row, V5 intensity profile; third row, recovered V5 signal; fourth row, V5 signal relative to baseline plotted as a function of the fraction of the HA intensity line integral. *Fos*: n = 13 CytoTapes from 13 neurons, 2 cultures; *Egr1*: n = 29 CytoTapes from 29 neurons, 2 cultures; *Arc*: n = 26 CytoTapes from 26 neurons, 3 cultures; NPAS4 activity: n = 14 CytoTapes from 14 neurons, 2 cultures; CREB activity: n = 11 CytoTapes from 11 neurons, 2 cultures. Thick centerline, mean; darker boundary, s.e.m.; lighter boundary, s.d.; thin light lines, intensity profiles from all individual CytoTapes; thin magenta lines, three

representative intensity profiles for each group. (**n**,**o**,**p**,**q**,**r**) Top, V5 signal relative to baseline with different KCl or FSK stimulations. Centerline, mean; shaded boundary, s.e.m. Sample size (n) is listed below as (X, Y, Z), which denotes n = X protein assemblies from Y neurons from Z cultures. (**n**) *Fos*: "55 mM KCl, 3 h", (13, 13, 2); "55 mM KCl, 1 h", (8, 8, 2), "55 mM KCl, 30 min", (26, 26, 4), "55 mM KCl, 10 min", (20, 20, 3), "No Stim": (13, 13, 2). (**o**) *Egr1*: "55 mM KCl, 3 h", (29, 29, 2); "55 mM KCl, 1 h", (14, 14, 3); "55 mM KCl, 30 min", (23, 23, 2); "55 mM KCl, 10 min", (8, 8, 2); "No Stim", (12, 12, 2). (**p**) *Arc*: "30 mM KCl, 3 h", (26, 26, 3); "30 mM KCl, 2 h", (36, 36, 2); "55 mM KCl, 1 h", (20, 20, 2); "55 mM KCl, 30 min", (15, 15, 2); "55 mM KCl, 10 min", (27, 27, 3); "No Stim", (24, 24, 2). (**q**) NPAS4 activity: "55 mM KCl, 3 h", (14, 14, 2); "55 mM KCl, 2 h", (11, 11, 2); "55 mM KCl, 1 h", (10, 10, 2); "55 mM KCl, 30 min", (10, 10, 2); "No Stim", (11, 11, 2). (**r**) CREB activity: "25 µM FSK, 1 h", (21, 21, 3); "20 µM FSK, 1 h", (27, 27, 2); "15 µM FSK, 1 h", (26, 26, 3); "10 µM FSK, 1 h", (26, 26, 3); "5 µM FSK, 1 h", (11, 11, 2); "No Stim", (9, 9, 2). Bottom, box plots comparing (top) peak relative change in V5 signal. *, $P < 0.05$; **, $P < 0.01$; ***, $P < 0.001$; n.s., not significant; Kruskal–Wallis analysis with Dunn's post hoc tests. Box plot: middle line, median; box boundary, interquartile range; whiskers, minimum and maximum; black dots, individual data points.

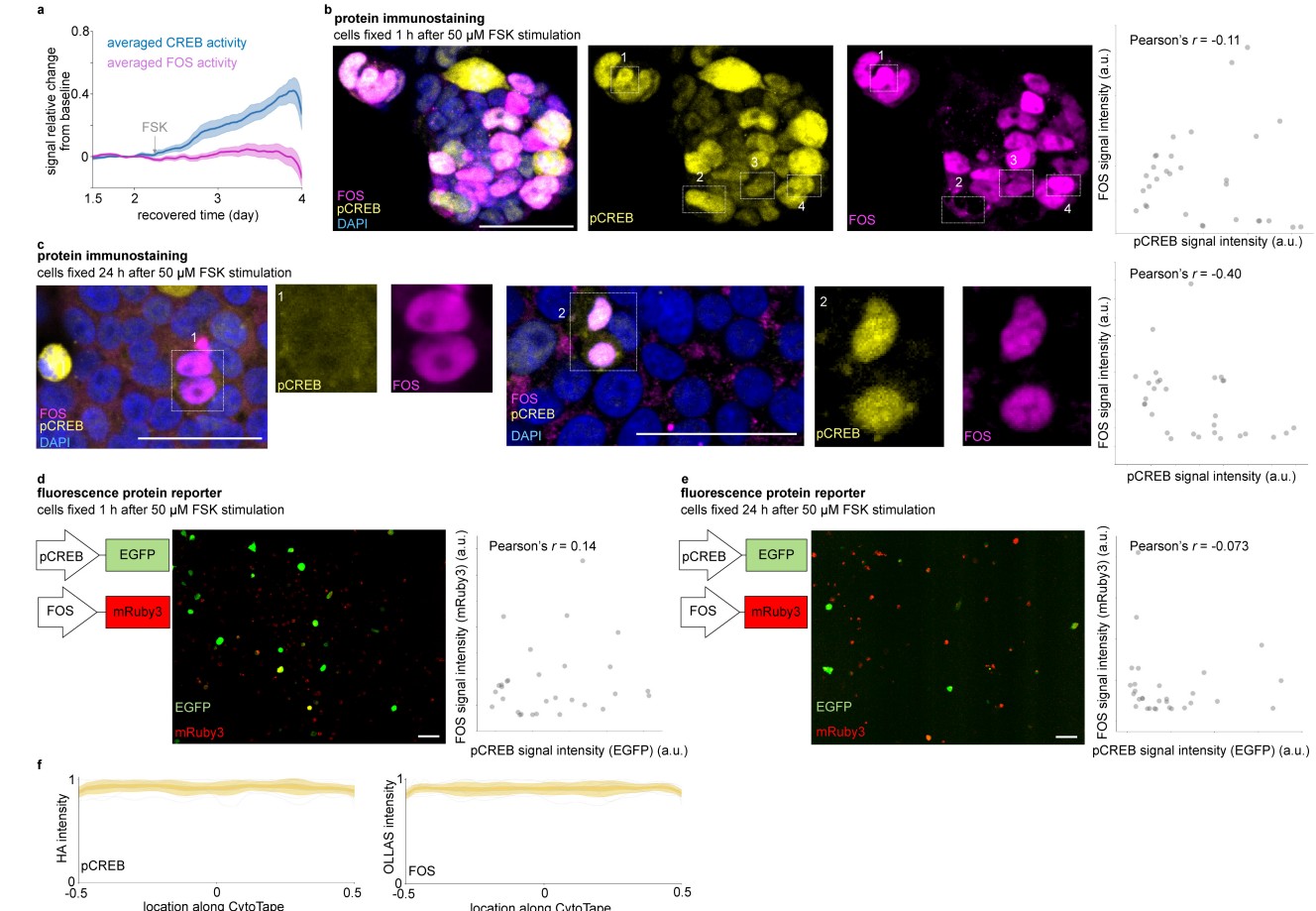

**Extended Data Fig. 5 | CREB and FOS exhibit complex, temporally decoupled dynamics following CREB activation, as validated by protein immunostaining and fluorescence protein reporter assays in HEK cells. (a)** Averaged pCREB and FOS signals (relative change from baseline) recorded by CytoTape, plotted against recovered time post-transfection. Thick centerline, mean; darker boundary, s.e.m. Confocal images of pCREB and FOS protein immunostaining in cells fixed at (**b**) 1 h and (**c**) 24 h after the completion of a 1-hour stimulation with 50 μM FSK. Confocal images of fluorescent reporters, pCREB-eGFP and FOS-mRuby3, in cells fixed at (**d**) 1 hour and (**e**) 24 h after the completion of a 1-hour stimulation with 50 μM FSK. Scatter plots in (**b**-**e**) were quantified in n = 30 cells from 2 cultures. Each dot represents a single HEK cell. Scale bar, 50 μm. (**f**) Profiles of pCREB and FOS signals along CytoTape in the control group (no stimulation). Thick centerline, mean; boundary, s.e.m; thin lines, single traces. pCREB and FOS signals were quantified in n = 8 cells from 2 cultures.

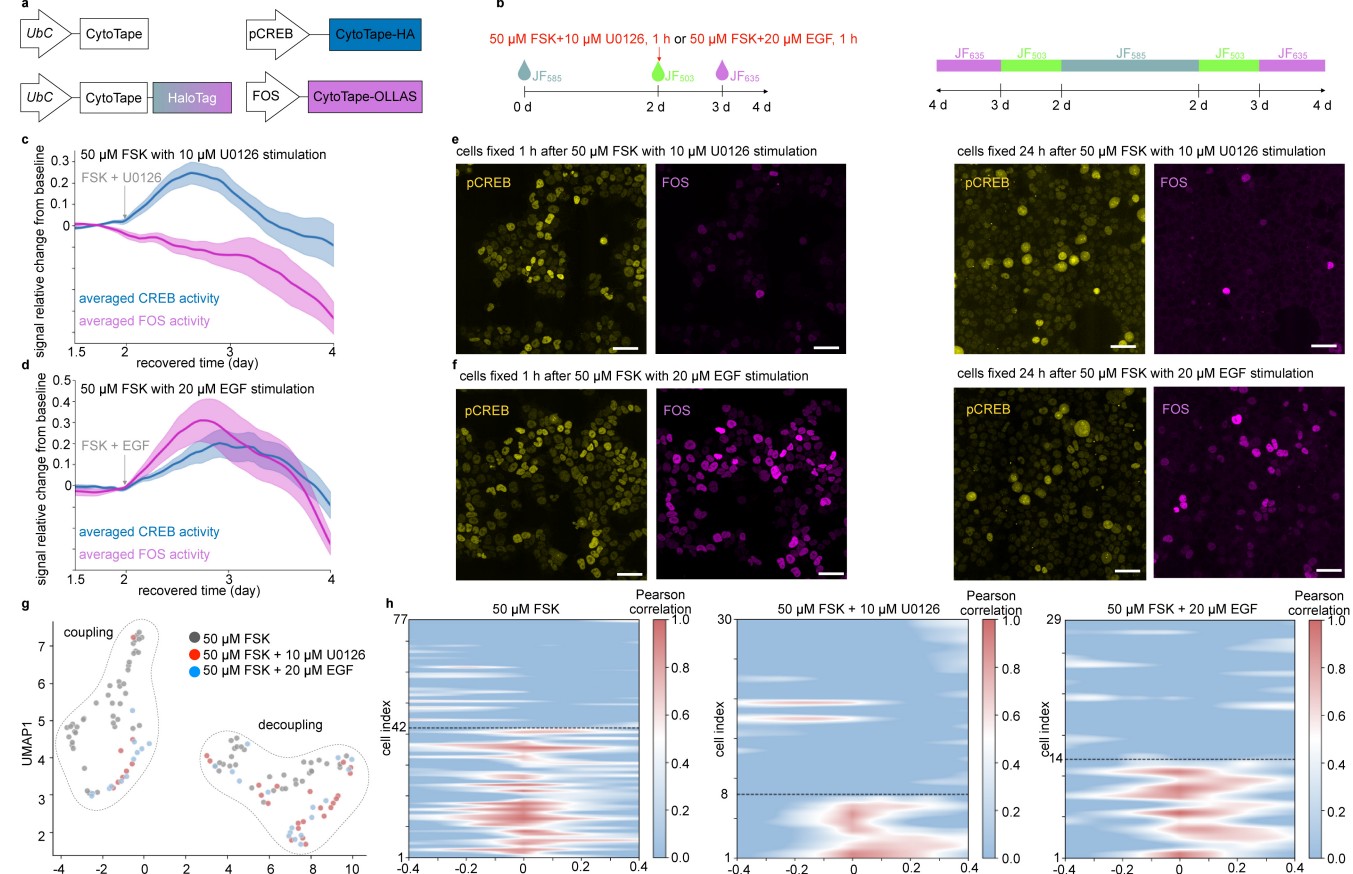

**Extended Data Fig. 6 | CytoTape resolves CREB–FOS decoupling under pathway perturbations in single cells.** (**a**) Schematics of the constructs transfected into HEK cells. (**b**) Experimental timelines (left) and expected JF dyes distributions along CytoTape (right). Averaged curve of pCREB and FOS signal relative change from baseline plotted against recovered time under (**c**) 50 μM FSK + 10 μM U0126 and (**d**) 50 μM FSK + 20 μM EGF stimulation. Thick centerline, mean; boundary, s.e.m. Confocal images of pCREB and FOS protein immunostaining in cells fixed at 1 h (left) and 24 h (right) after the completion of a 1-hour stimulation with (**e**) 50 μM FSK + 10 μM U0126 stimulation and (**f**) 50 μM FSK + 20 μM EGF. Scale bar, 50 μm. (**g**) UMAP plot of time-lagged

correlations between pCREB and FOS signals across single cells under 50 μM FSK (n = 77 CytoTapes from 77 cells, 5 cultures), 50 μM FSK + 10 μM U0126 (n = 30 CytoTapes from 30 cells, 2 cultures), and 50 μM FSK + 20 μM EGF (n = 29 CytoTapes from 29 cells, 2 cultures). (**h**) Heatmaps showing time-lagged correlations between pCREB and FOS signals for decoupled (upper) and coupled (lower) HEK cells under 50 μM FSK (left), 50 μM FSK + 10 μM U0126, and 50 μM FSK + 20 μM EGF stimulation. x-axis represents a time-lagged shift window of ±0.4 days. y-axis represents HEK cell numbers. Data of the "50 μM FSK" group is from Fig. 4.

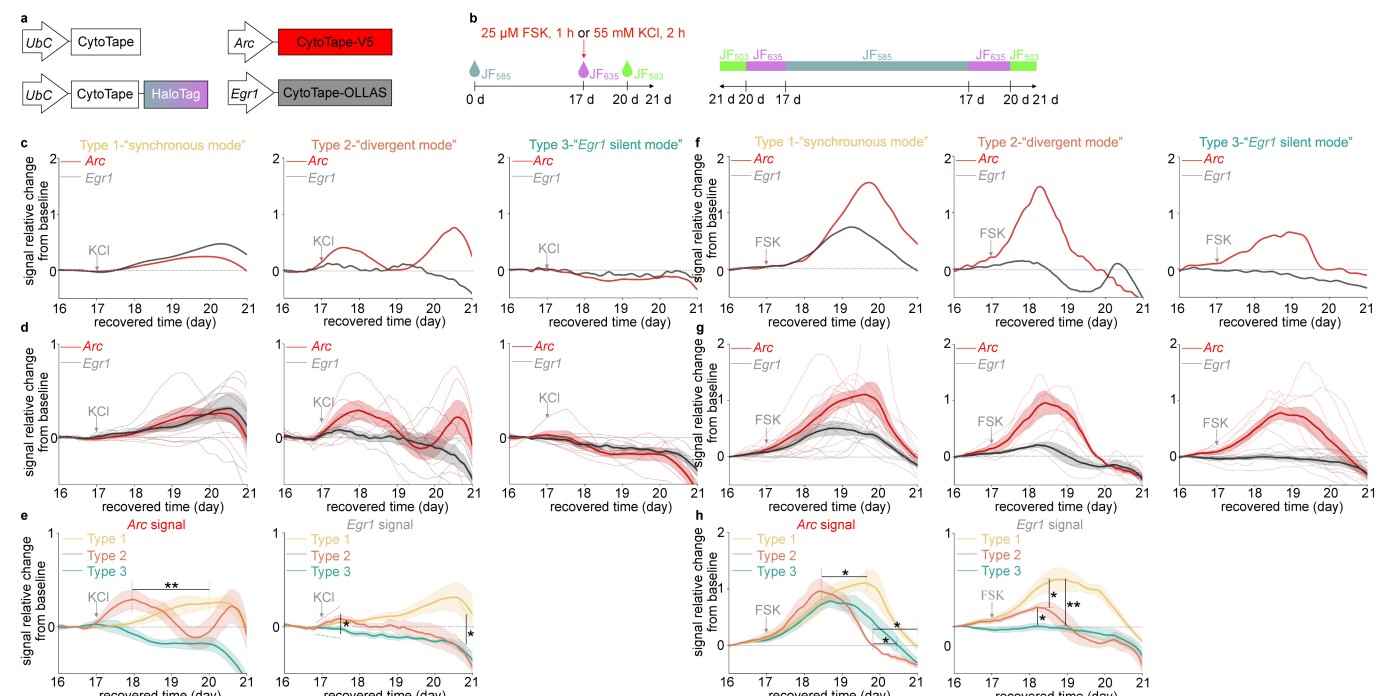

**Extended Data Fig. 7 | CytoTape reveals nonlinear waveform correlations between *Arc*- and *Egr1*-promoter activities in single neurons.** (**a**) Schematic of the constructs transfected into cultured neurons, (**b**) the experimental timeline (left), and the expected dye distributions along the CytoTape (right). (**c**) Representative single traces of *Arc* and *Egr1* signals relative change from baseline, plotted against recovered time after calcium-phosphate transfection in three responsive types under KCl stimulation. (**d**) Averaged *Arc* and *Egr1* signals relative change from baseline plotted against recovered time after calcium-phosphate transfection (n = 16 CytoTapes from 16 neurons, 3 cultures). (**e**) Comparison of *Arc* (left) and *Egr1* (right) signals across three responsive cell types. (**f**) Representative single traces of *Arc* and *Egr1* signals relative change from baseline plotted against recovered time after calcium-phosphate transfection in three responsive types under FSK stimulation. (**g**) Averaged *Arc* and *Egr1* signals relative change from baseline plotted against recovered time after transfection (n = 21 CytoTapes from 21 neurons, 3 cultures). (**h**) Comparison of *Arc* (left) and *Egr1* (right) signals across three responsive cell types. In **d**,**e**,**g**,**h**: thick centerline, mean; boundary, s.e.m.; thin light lines, individual CytoTape data; *, $P < 0.05$; **, $P < 0.01$; Kruskal–Wallis analysis with Dunn's post hoc tests.

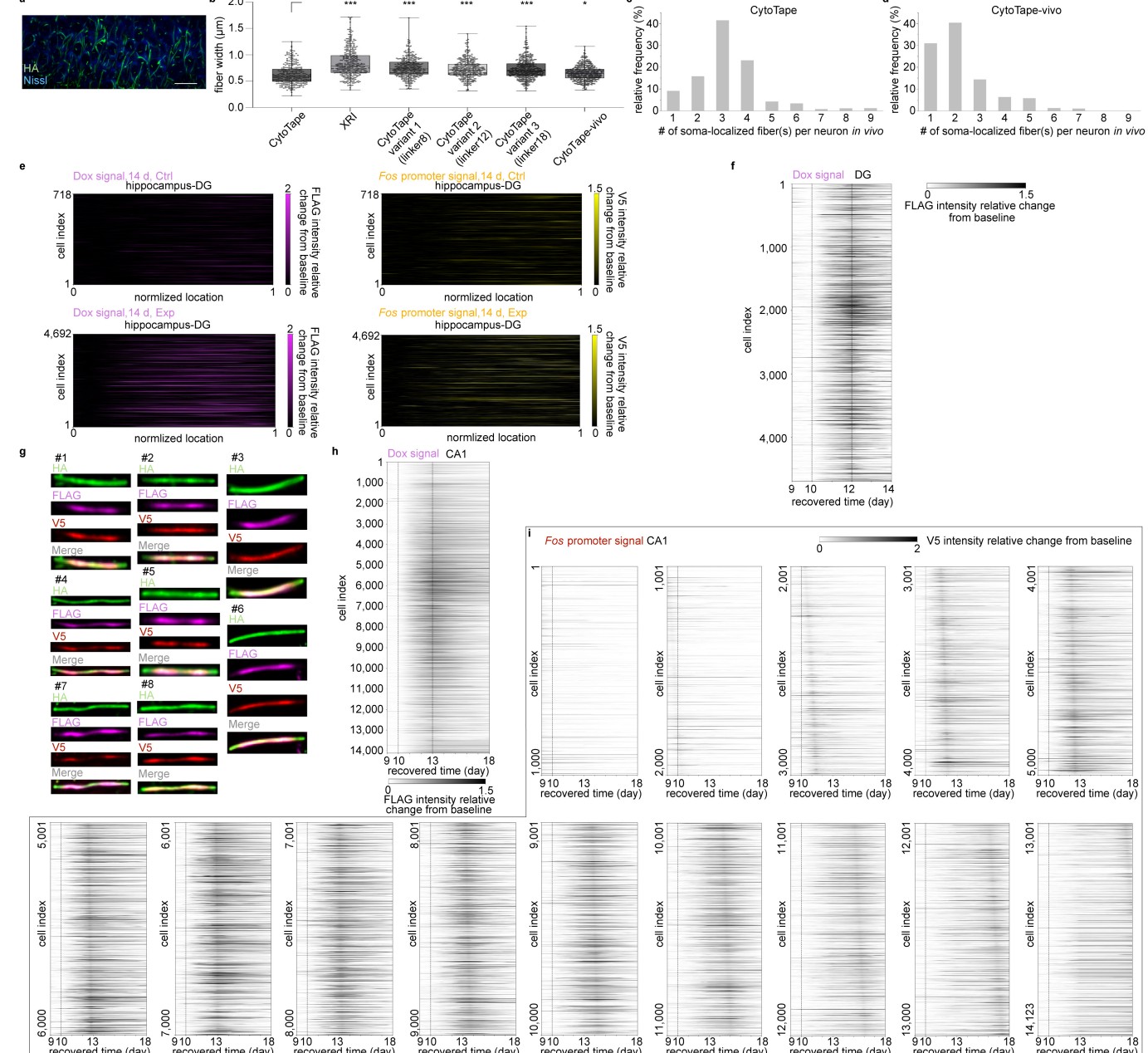

**Extended Data Fig. 8 | CytoTape-vivo screening, characterization, and weeks-long recording across large neuron populations in vivo.** (**a**) A confocal image of CytoTape fibers in DG neurons of mouse brain 14 days after AAV injection. Scale bar, 10 μm (**b**) Fiber width of CytoTape, XRI, and CytoTape-vivo variants in the soma of DG neurons in mouse brains 14 days after AAV injection. n = 312 CytoTape fibers, 358 XRI fibers, 364 CytoTape-vivo variant 1 (linker8) fibers, 374 CytoTape-vivo variant 2 (linker12) fibers, 460 CytoTape-vivo variant 3 (linker18) fibers, and 501 CytoTape-vivo fibers; each group is from 1 mouse. *, $P < 0.05$; ***, $P < 0.001$; Kruskal–Wallis analysis with Dunn's post hoc tests. Middle line, median; box boundary, interquartile range; whiskers, minimum and maximum; black dots, individual data points. Histogram of the number of soma-localized CytoTape (**c**) and CytoTape-vivo (**d**) fibers per neuron. n = 156

DG neurons from 1 mouse for CytoTape and 173 DG neurons from 1 mouse for CytoTape-vivo. For the experiment described in Fig. 5g–j: (**e**) Dox signals (left) and *Fos* promoter signals (right) in the control group (first row, "Ctrl"; no Dox, no KA; n = 718 DG neurons from 1 mouse) and the experimental group (second row, "Exp"; Dox and KA administered; n = 4,692 DG neurons from 1 mouse). (**f**) Recorded relative change in Dox signal monomer intensity from baseline in the experimental group (n = 4,692 neurons from one mouse). For the experiment described in Fig. 5k–m: (**g**) Representative confocal images of CytoTape-vivo. (**h**) Recorded relative change in Dox signal monomer intensity from baseline. (**i**) Recorded *Fos* signal monomer intensity relative change from baseline (enlarged views of Fig. 5l). n = 14,123 CA1 neurons from one mouse.

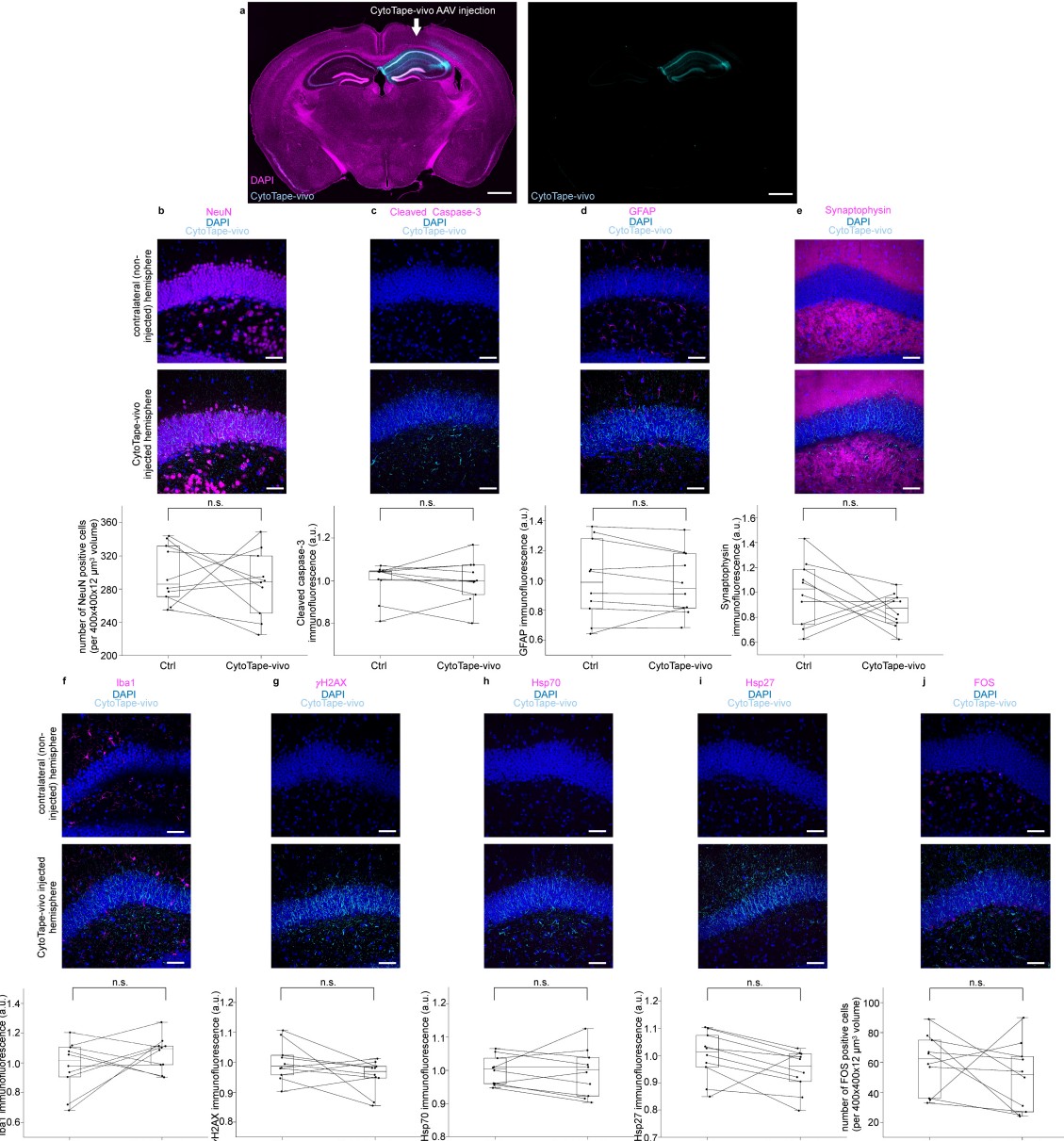

**Extended Data Fig. 9 | Immunohistochemical characterization of cellular and synaptic physiology markers in mouse brains expressing CytoTape-vivo.** (**a**) Representative confocal images of brain slices from adult mice expressing CytoTape-vivo in the hippocampus of the right cerebral hemisphere for 14 days following AAV injection. Scale bars, 500 μm. (**b-j**, top) Representative confocal images of the hippocampus region in the non-injected hemisphere (Ctrl) and CytoTape-vivo-injected hemisphere (CytoTape-vivo) in the brain slices stained with antibodies against each of the cellular and synaptic markers indicated, and DAPI. Scale bars, 50 μm. (**b-j**, bottom) Box plots of the quantifications for each of the cellular and synaptic markers between the

non-injected hemisphere (Ctrl) and CytoTape-vivo-injected hemisphere (CytoTape-vivo); for each marker in each hemisphere, n = 10 fields of view (FOVs) from 3 mice. Middle line in box plot, median; box boundary, interquartile range; whiskers, minimum and maximum; black dots, individual data points. n.s., not significant; two-sided Wilcoxon signed-rank test. (**b**) NeuN (a neuronal marker). (**c**) Cleaved Caspase-3 (an apoptotic marker). (**d**) GFAP (an astrocyte marker). (**e**) Synaptophysin (a synaptic protein marker). (**f**) Iba1 (a microglial marker). (**g**) γH2AX (a DNA damage marker). (**h,i**) Hsp70 and Hsp27 (cell physiological stress markers). (**j**) FOS protein (a cellular activity maker).

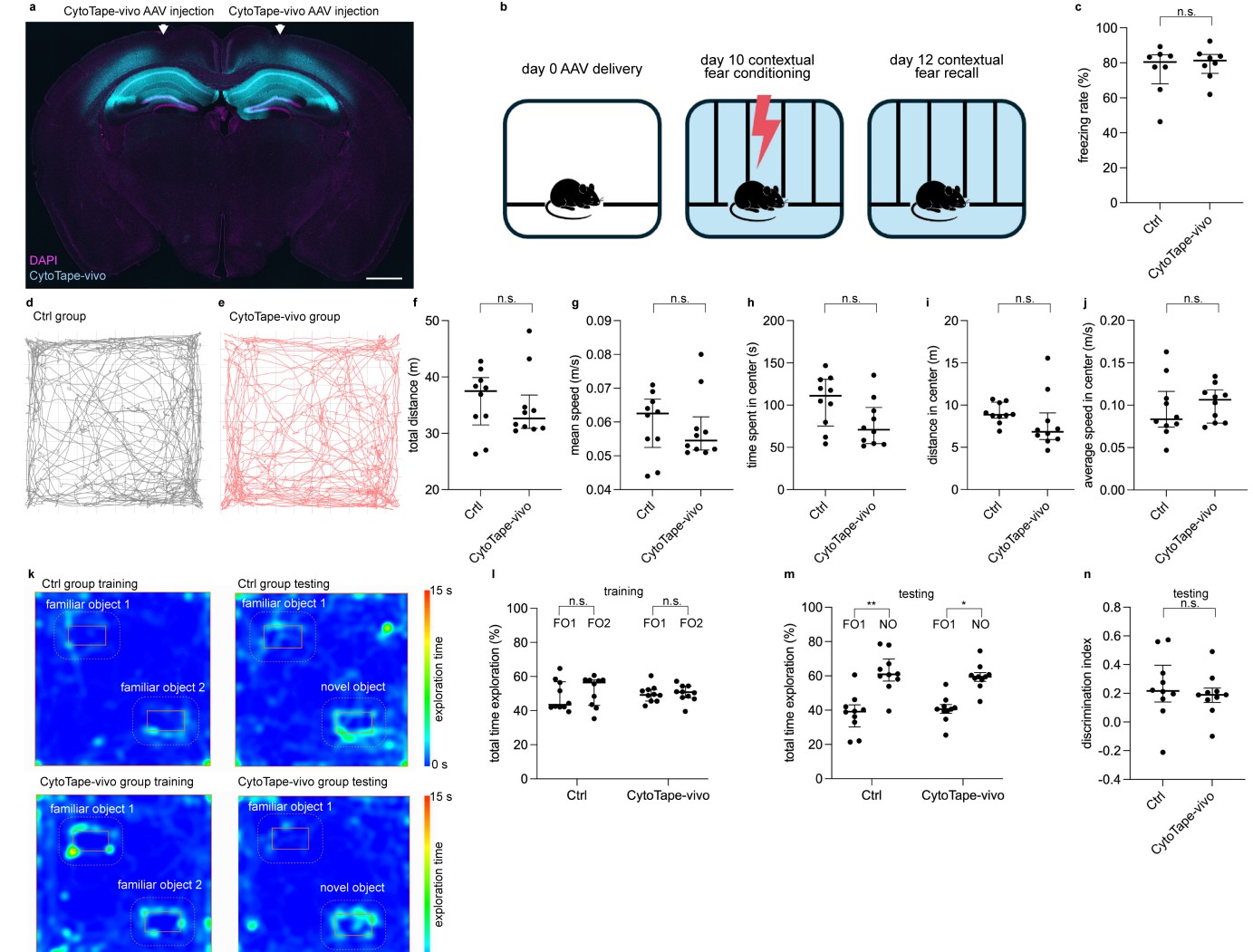

**Extended Data Fig. 10 | Behavioral characterizations of mice with and without CytoTape-vivo expression, under contextual fear conditioning, open field test, and novel object recognition.** (**a**) Representative confocal images of brain slices from adult mice expressing CytoTape-vivo over weeks in the hippocampi of both the left and the right cerebral hemispheres following AAV injection. Scale bars, 500 μm. (**b**) Experimental pipeline of contextual fear conditioning experiment for mice with or without CytoTape-vivo expression for 12 days. (**c**) Freezing rate during contextual fear recall (n = 8 mice per group). n.s., not significant; two-sided Mann–Whitney U test. Middle line, median; whiskers, interquartile range; black dots, data from individual mice. (**d,e**) Representative movement traces of mice with or without CytoTape-vivo expression for 10 days in the open-field arena. Quantification of locomotor parameters, including (**f**) total travel distance, (**g**) mean speed, (**h**) time spent in the center, (**i**) distance traveled in the center, and (**j**) average speed in the center, comparing the control and CytoTape-vivo groups (n = 10 mice per group). n.s., not significant; two-sided Mann–Whitney U test. Middle line, median; whiskers, interquartile range; black dots, data from individual mice. (**k**) Representative exploration heatmaps of mice with or without CytoTape-vivo expression for 17 days during the training (left) and testing (right) phases of the novel object recognition task. Dashed outlines indicate predefined object zones. Quantification of novel object recognition performance, including (**l**) percentage of total exploration time during familiarization of object 1 (FO1) and object 2 (FO2), (**m**) percentage of exploration time for familiar (FO1) and novel (NO) objects during the test phase, and (**n**) discrimination index comparing between the control and CytoTape-vivo groups (n = 10 mice per group). In **l**, n.s., not significant; two-sided Wilcoxon signed-rank test. In **m**, *, P < 0.05; **, P < 0.01; two-sided Wilcoxon signed-rank test. In **n**, n.s., not significant; two-sided Mann–Whitney U test. Middle line in bar plots, median; whiskers, interquartile range; black dots, data from individual mice.

# Reporting Summary

## Statistics

For all statistical analyses, confirm that the following items are present in the figure legend, table legend, main text, or Methods section.

| n/a | Confirmed | |
|---|---|---|
| ☐ | ☒ | The exact sample size (*n*) for each experimental group/condition, given as a discrete number and unit of measurement |
| ☐ | ☒ | A statement on whether measurements were taken from distinct samples or whether the same sample was measured repeatedly |
| ☐ | ☒ | The statistical test(s) used AND whether they are one- or two-sided<br>*Only common tests should be described solely by name; describe more complex techniques in the Methods section.* |
| ☒ | ☐ | A description of all covariates tested |
| ☐ | ☒ | A description of any assumptions or corrections, such as tests of normality and adjustment for multiple comparisons |
| ☐ | ☒ | A full description of the statistical parameters including central tendency (e.g. means) or other basic estimates (e.g. regression coefficient) AND variation (e.g. standard deviation) or associated estimates of uncertainty (e.g. confidence intervals) |
| ☐ | ☒ | For null hypothesis testing, the test statistic (e.g. *F*, *t*, *r*) with confidence intervals, effect sizes, degrees of freedom and *P* value noted<br>*Give P values as exact values whenever suitable.* |
| ☒ | ☐ | For Bayesian analysis, information on the choice of priors and Markov chain Monte Carlo settings |
| ☐ | ☒ | For hierarchical and complex designs, identification of the appropriate level for tests and full reporting of outcomes |
| ☒ | ☐ | Estimates of effect sizes (e.g. Cohen's *d*, Pearson's *r*), indicating how they were calculated |

*Our web collection on statistics for biologists contains articles on many of the points above.*

## Software and code

Policy information about availability of computer code

| | |
|---|---|
| Data collection | Nikon NIS-Elements software was used for image acquisition on fluorescence microscopes. |
| Data analysis | Statistical analysis was performed in Prism 10 (GraphPad). Image analysis was performed in ImageJ (National Institutes of Health), napari (napari contributors; doi:10.5281/zenodo.3555620), and Python. The code developed in this work for in vivo image analysis, Tape Reader v1.0, can be found at https://github.com/LinghuLab/TapeReader. The code developed in this work for signal extraction and analysis of protein tape recordings, Tape Analyzer v1.0, can be found at https://github.com/LinghuLab/TapeAnalyzer. The structure of the protein monomer for simulations was predicted by previously published AlphaFold3 (alphafoldserver.com). Molecular dynamics simulations were performed using previously published GROMACS 2021.1 packages (manual.gromacs.org/2021.1). The previously published ProtSSN model for protein mutation prediction can be found at https://github.com/tyang816/ProtSSN. The previously published CPDiffusion model for protein sequence generation prediction can be found at https://github.com/bzho3923/CPDiffusion. Video analysis of mouse behavior was performed by ezTrack v1.2 (https://github.com/denisecailab/ezTrack) and LabGym v2.9.0 (https://github.com/umyelab/LabGym). |

For manuscripts utilizing custom algorithms or software that are central to the research but not yet described in published literature, software must be made available to editors and reviewers. We strongly encourage code deposition in a community repository (e.g. GitHub). See the Nature Portfolio guidelines for submitting code & software for further information.

# Data

Policy information about availability of data

All manuscripts must include a data availability statement. This statement should provide the following information, where applicable:
- Accession codes, unique identifiers, or web links for publicly available datasets
- A description of any restrictions on data availability
- For clinical datasets or third party data, please ensure that the statement adheres to our policy

The plasmids and the corresponding sequence of working CytoTape constructs reported in this paper are available at Addgene (plasmid IDs 239423-239430, 239616, and 250670-250672). The mouse brain in vivo recording datasets generated and analyzed in this study are available at Zenodo (https://doi.org/10.5281/zenodo.18123891). Supplementary Information includes the development and discussion of the CytoTape toolkit, Supplementary Figs. 1-24, Supplementary Video 1, Supplementary Tables 1-11, and Supplementary References. There is no restriction on data availability.

# Research involving human participants, their data, or biological material

Policy information about studies with human participants or human data. See also policy information about sex, gender (identity/presentation), and sexual orientation and race, ethnicity and racism.

| | |
|---|---|
| Reporting on sex and gender | N/A |
| Reporting on race, ethnicity, or other socially relevant groupings | N/A |
| Population characteristics | N/A |
| Recruitment | N/A |
| Ethics oversight | N/A |

Note that full information on the approval of the study protocol must also be provided in the manuscript.

# Field-specific reporting

Please select the one below that is the best fit for your research. If you are not sure, read the appropriate sections before making your selection.

☒ Life sciences          ☐ Behavioural & social sciences          ☐ Ecological, evolutionary & environmental sciences

For a reference copy of the document with all sections, see nature.com/documents/nr-reporting-summary-flat.pdf

# Life sciences study design

All studies must disclose on these points even when the disclosure is negative.

| | |
|---|---|
| Sample size | For cell culture experiments and mouse experiments (excluding behavioral tests), sample sizes were chosen based on our previously published molecular technology development work (Nature Biotechnology 41, 640–651, 2023; Cell 183, 1682-1698, 2020) and established literature in molecular biosensor and recorder development. For mouse behavioral tests, sample sizes were determined based on the expected variance and effect sizes reported in relevant previous studies (Nature 531, 508-512, 2016; Cell 181, 410-423, 2020). No statistical method was used to predetermine sample sizes. This approach is consistent with the primary objective of this work, which is to develop, validate, and demonstrate a novel molecular technology. We found the sample sizes sufficient to yield reproducible results. |
| Data exclusions | We applied multiple quality control criteria to ensure the computationally segmented fiber structures from mouse brain tissue accurately reflected fiber morphology and anatomical positioning. First, segmented structures with length smaller than 8 µm were excluded. Next, to ensure morphological accuracy of segmentation, segmented structures with non-fiber-like morphology were removed through principal component analysis (PCA). Specifically, we calculated the explained variance of the first principal component of the skeleton of the segmented structure and excluded segmented structures with values below 0.8. Finally, fibers located outside soma regions detected by Nissl staining were excluded from further analysis. In 14-day and 18-day mouse brain in vivo recording experiments, if there is more than one fiber in a cell, the longest fiber was used for subsequent analysis. For electrophysiological characterization of synaptic transmission, recordings were excluded if the series resistance exceeded 20 MΩ to ensure high-quality voltage clamp. |
| Replication | Experiments were replicated at least once. All attempts at replication were successful. The detailed experimental protocols are provided to facilitate replication by others. |
| Randomization | All biological replicates were treated identically, and randomization was not relevant to this technology development work. |
| Blinding | Samples for all biological replicates were obtained under identical conditions. Experimenters were blinded to manipulation when analyzing the data, except for mouse behavior tests where experimenters were not blinded to group identity. |

# Reporting for specific materials, systems and methods

We require information from authors about some types of materials, experimental systems and methods used in many studies. Here, indicate whether each material, system or method listed is relevant to your study. If you are not sure if a list item applies to your research, read the appropriate section before selecting a response.

## Materials & experimental systems

| n/a | Involved in the study |
|---|---|
| ☐ | ☒ Antibodies |
| ☐ | ☒ Eukaryotic cell lines |
| ☒ | ☐ Palaeontology and archaeology |
| ☐ | ☒ Animals and other organisms |
| ☒ | ☐ Clinical data |
| ☒ | ☐ Dual use research of concern |
| ☒ | ☐ Plants |

## Methods

| n/a | Involved in the study |
|---|---|
| ☒ | ☐ ChIP-seq |
| ☒ | ☐ Flow cytometry |
| ☒ | ☐ MRI-based neuroimaging |

## Antibodies

| Antibodies used | Primary antibodies (1:500 for immunofluorescence of cultured cells and brain slices): |
|---|---|
| | HA Tag Monoclonal Antibody (clone name: C29F4), Rabbit IgG, Cell Signaling Technology Cat# 3724; RRID: AB_1549585 |
| | HA Tag Polyclonal Antibody, Chicken IgY, Invitrogen Cat# PA5-33243; RRID: AB_2550658 |
| | V5 Tag Monoclonal Antibody (clone name: SV5-Pk1), Mouse IgG2a, Invitrogen Cat# R960-25; RRID: AB_2556564 |
| | DYKDDDDK (FLAG) Tag Superclonal Antibody (clone names: 20H18L16, 20H1L23, 8H2L5, 8H8L17), Rabbit IgG, Invitrogen Cat# 740001;RRID: AB_2610628 |
| | OLLAS Tag Monoclonal Antibody (clone name: L2), Rat IgG1 kappa, Invitrogen Cat# MA5-16125; RRID: AB_11152481 |
| | E Tag Monoclonal Antibody (clone name: 11H12B3), Mouse IgG1, Invitrogen Cat# MA5-38276; RRID: AB_2898191 |
| | c-Fos Monoclonal Antibody (clone name: Ch108B5), Chicken IgY, Synaptic Systems Cat# 226 009; RRID: AB_2943525 |
| | Phospho-CREB (Ser133) Monoclonal Antibody (clone name: 87G3), Rabbit IgG, Cell Signaling Technology Cat# 9198; RRID: AB_2561044 |
| | Phospho-Elk-1 (Ser383) Monoclonal Antibody (clone name: 2B1), Mouse IgG1, Cell Signaling Technology Cat# 9186; RRID: AB_2277933 |
| | Egr1 Monoclonal Antibody (clone name: 15F7), Rabbit IgG, Cell Signaling Technology Cat# 4153; RRID: AB_2097038 |
| | Phospho-p44/42 MAPK (Erk1/2) (Thr202/Tyr204) Monoclonal Antibody (clone name: D13.14.4E), Rabbit IgG, Cell Signaling Technology Cat# 4370; RRID: AB_2315112 |
| | GFAP Monoclonal Antibody (clone name: D1F4Q), Rabbit IgG, Cell Signaling Technology Cat# 12389; RRID: AB_2631098 |
| | Cleaved Caspase-3 (Asp175) Monoclonal Antibody (clone name: 5A1E), Rabbit IgG, Cell Signaling Technology Cat# 9664; RRID: AB_2070042 |
| | HSP70 Polyclonal Antibody, Rabbit IgG, Cell Signaling Technology Cat# 4872; RRID: AB_2279841 |
| | HSP27 Monoclonal Antibody (clone name: G31), Mouse IgG1, Cell Signaling Technology Cat# 2402; RRID: AB_331761 |
| | Phospho-Histone H2A.X (γH2AX; Ser139) Antibody (clone name: JBW301), Mouse IgG1, Sigma Cat# 05-636; RRID: AB_309864 |
| | Synaptophysin Monoclonal Antibody (clone name: SVP-38), Mouse IgG1, Sigma Cat# S5768; RRID: AB_477523 |
| | NeuN Polyclonal Antibody, Guinea Pig IgG, Synaptic Systems Cat# 266 004; RRID: AB_2619988 |
| | Iba1 Polyclonal Antibody, Rabbit IgG, Wako Chemicals Cat# 019-19741; RRID: AB_839504 |
| | Ki67 Monoclonal Antibody (clone name: B56), Mouse IgG1, Abcam Cat# ab279653; RRID: AB_2934265 |
| | GRP78 BiP Polyclonal Antibody, Rabbit IgG, Abcam Cat# ab21685; RRID: AB_2119834 |
| | TOMM20 Monoclonal Antibody (clone name: EPR15581-54), Rabbit IgG, Abcam Cat# ab186735; RRID: AB_2889972 |
| | |
| | Fluorescent secondary antibodies (1:500 for immunofluorescence of cultured cells and brain slices): |
| | Goat anti-Mouse IgG1 (Gamma 1 chain) Pre-absorbed Secondary Antibody, ATTO 425, Rockland Cat# 610-151-040; RRID: AB_2614850 |
| | Goat anti-Mouse IgG2a Cross-Adsorbed Secondary Antibody, Alexa Fluor 488, Invitrogen Cat# A-21131; RRID: AB_2535771 |
| | Goat anti-Guinea Pig IgG (H+L) Highly Cross-Adsorbed Secondary Antibody, Alexa Fluor 488, Invitrogen Cat# A-11073; RRID: AB_2534117 |
| | Goat anti-Rabbit IgG (H+L) Cross-Adsorbed Secondary Antibody, Alexa Fluor 488, Invitrogen Cat# A-11008; RRID: AB_143165 |
| | Goat anti-Rabbit IgG (H+L) Highly Cross-Adsorbed Secondary Antibody, Alexa Fluor Plus 488, Invitrogen Cat# A-32731; RRID: AB_2633280 |
| | Goat anti-Mouse IgG2a Cross-Adsorbed Secondary Antibody, Alexa Fluor 546, Invitrogen Cat# A-21133; RRID: AB_2535772 |
| | Goat anti-Rat IgG (H+L) Cross-Adsorbed Secondary Antibody, Alexa Fluor 546, Invitrogen Cat# A-11081; RRID: AB_2534125 |
| | Goat anti-Rabbit IgG (H+L) Cross-Adsorbed Secondary Antibody, Alexa Fluor 594, Invitrogen Cat# A-11012; RRID: AB_2534079 |
| | Goat anti-Chicken IgY (H+L) Cross-Adsorbed Secondary Antibody, Alexa Fluor Plus 647, Invitrogen Cat# A-32933; RRID: AB_2762845 |
| | Goat anti-Mouse IgG1 Cross-Adsorbed Secondary Antibody, Alexa Fluor 647, Invitrogen Cat# A-21240; RRID: AB_2535809 |
| | |
| | Additional details of primary antibodies, secondary antibodies, dyes, and other reagents used in this study are listed in Supplementary Table S10. |
| Validation | Validation statements for use in immunohistochemistry and relevant citations of the primary antibodies used in this study are listed on the manufacturers' website listed below. Citations of the antibodies can also be searched at www.citeab.com |

anti-Etag (Invitrogen, Cat# MA5-38276), https://www.thermofisher.com/antibody/product/E-Tag-Antibody-clone-11H12B3-Monoclonal/MA5-38276
anti-V5 (Invitrogen, Cat# R960-25), https://www.thermofisher.com/antibody/product/V5-Tag-Antibody-clone-SV5-Pk1-Monoclonal/R960-25
anti-OLLAS (Invitrogen, Cat# MA5-16125), https://www.thermofisher.com/antibody/product/OLLAS-Tag-Antibody-clone-L2-Monoclonal/MA5-16125
anti-FLAG (Invitrogen, Cat# 740001), https://www.thermofisher.com/antibody/product/DYKDDDDK-Tag-Antibody-clone-20H18L16-20H1L23-8H2L5-8H8L17-Recombinant-Superclonal/740001
anti-HA (Cell Signaling Technology, Cat# 3724), https://www.cellsignal.com/products/primary-antibodies/ha-tag-c29f4-rabbit-monoclonal-antibody/3724
anti-HA (Invitrogen, Cat# PA5-33243), https://www.thermofisher.com/antibody/product/HA-Tag-Antibody-Polyclonal/PA5-33243
anti-c-Fos (Synaptic Systems, Cat# 226009), https://sysy.com/product/226009
anti-Phospho-CREB (Ser133) (Cell Signaling Technology, Cat# 9198), https://www.cellsignal.com/products/primary-antibodies/phospho-creb-ser133-87g3-rabbit-mab/9198
anti-GFAP (Cell Signaling Technology, Cat# 12389), https://www.cellsignal.com/products/primary-antibodies/gfap-d1f4q-xp-rabbit-mab/12389
anti-Cleaved Caspase-3 (Cell Signaling Technology, Cat# 9664), https://www.cellsignal.com/products/primary-antibodies/cleaved-caspase-3-asp175-5a1e-rabbit-mab/9664
anti-Hsp70 (Cell Signaling Technology, Cat# 4872), https://www.cellsignal.com/products/primary-antibodies/hsp70-antibody/4872
anti-Hsp27 (Cell Signaling Technology, Cat# 2402), https://www.cellsignal.com/products/primary-antibodies/hsp27-g31-mouse-mab/2402
anti-γH2AX (Millipore, Cat# 05-636), https://www.sigmaaldrich.com/US/en/product/mm/05636
anti-Synaptophysin (Sigma, Cat# S5768), https://www.sigmaaldrich.com/US/en/product/sigma/s5768
anti-NeuN (SYSY, Cat# 266004), https://sysy.com/product/266004
anti-Iba1 (Wako Chemicals, Cat# ab279653), https://fujifilmbiosciences.fujifilm.com/us/anti-iba1-goat.html
anti-Ki67 (Abcam, Cat# ab279653), https://www.abcam.com/en-us/products/primary-antibodies/ki67-antibody-b56-ab279653
anti-GRP78 BiP (Abcam, Cat# ab21685), https://www.abcam.com/en-us/products/primary-antibodies/grp78-bip-antibody-ab21685
anti-TOMM20 (Abcam, Cat# ab186735), https://www.abcam.com/en-us/products/primary-antibodies/tomm20-antibody-epr15581-54-mitochondrial-marker-ab186735
anti-Phospho-Elk-1 (Cell Signaling Technology, Cat# 9186), https://www.cellsignal.com/products/primary-antibodies/phospho-elk-1-ser383-2b1-mouse-mab/9186
anti-Egr1(Cell Signaling Technology, Cat# 4153), https://www.cellsignal.com/products/primary-antibodies/egr1-15f7-rabbit-mab/4153
anti-Phospho-p44/42 MAPK (Erk1/2) (Thr202/Tyr204) (D13.14.4E) (Cell Signaling Technology, Cat# 4370), https://www.cellsignal.com/products/primary-antibodies/phospho-p44-42-mapk-erk1-2-thr202-tyr204-d13-14-4e-xp-rabbit-mab/4370

# Eukaryotic cell lines

Policy information about cell lines and Sex and Gender in Research

| Cell line source(s) | HEK293T clone 17 (CRL-11268) and HeLa (CCL-2) cell lines from ATCC. |
|---|---|
| Authentication | The cell line was authenticated by the manufacturer via STR profiling. |
| Mycoplasma contamination | The cell lines were tested for mycoplasma contamination by the manufacturer to their standard levels of stringency (mycoplasma contamination was not detected). |
| Commonly misidentified lines (See ICLAC register) | No commonly misidentified cell lines were used in the study. |

# Animals and other research organisms

Policy information about studies involving animals; ARRIVE guidelines recommended for reporting animal research, and Sex and Gender in Research

| Laboratory animals | Male and female Swiss Webster mice at postnatal day 0 or 1 (Taconic), male C57BL/6J mice at 2-5 months of age (Jackson Laboratory). Mice were maintained on a 12-hour light/dark cycle (lights on at 06:00 EST) in a temperature-controlled environment at 22±1°C, with a relative humidity of 30-50%. Mice were group-housed, except for pregnant females (individually housed prior to delivery) and mice post-surgery (individually housed for the remainder of the experiment). |
|---|---|
| Wild animals | The study did not involve wild animals. |
| Reporting on sex | Male and female Swiss Webster neonatal mice were used randomly in this study, delivered by pregnant females. Male C57BL/6J mice were used in this study. |
| Field-collected samples | The study did not involve field-collected samples. |
| Ethics oversight | All animal procedures were conducted in accordance with the United States National Institutes of Health Guide for the Care and Use of Laboratory Animals and were approved by the Institutional Animal Care and Use Committee of the institution where each procedure was conducted (University of Michigan, Icahn School of Medicine at Mount Sinai, or Massachusetts Institute of Technology). |

Note that full information on the approval of the study protocol must also be provided in the manuscript.

# Plants

Seed stocks

*Report on the source of all seed stocks or other plant material used. If applicable, state the seed stock centre and catalogue number. If plant specimens were collected from the field, describe the collection location, date and sampling procedures.*

Novel plant genotypes

*Describe the methods by which all novel plant genotypes were produced. This includes those generated by transgenic approaches, gene editing, chemical/radiation-based mutagenesis and hybridization. For transgenic lines, describe the transformation method, the number of independent lines analyzed and the generation upon which experiments were performed. For gene-edited lines, describe the editor used, the endogenous sequence targeted for editing, the targeting guide RNA sequence (if applicable) and how the editor was applied.*

Authentication

*Describe any authentication procedures for each seed stock used or novel genotype generated. Describe any experiments used to assess the effect of a mutation and, where applicable, how potential secondary effects (e.g. second site T-DNA insertions, mosiacism, off-target gene editing) were examined.*

