## [Peer Review File · Nature]

Scalable and multiplexed recorders of gene regulation dynamics across weeks

Corresponding Author: Dr Changyang Linghu

Version 1:

Reviewer comments:

Referee #1

(Remarks to the Author)

This manuscript by Zheng et al. presents a technical step forward in temporal/analog molecular recording via expanding protein tapes. The technology is interesting and potentially useful, the experiments are high-quality, and the overall presentation of results is clear despite substantial complexity. The major advance here is additional engineering of the repeating unit that makes up the recording filament to increase flexibility, which allows for longer assemblies (and thus temporally longer recordings) without as much deformation of the cell. My major concerns are whether this engineering warrants reintroduction of the approach under a new name, rather than XRI v2, and whether the authors have demonstrated any uses that could not be achieved using older, established approaches. Other concerns are relatively minor.

Major Concerns:

1. The efforts to create a thinner, more flexible recording filament are interesting, and clearly yielded an improved technology. However, this is conceptually the same approach presented as XRI in 2023 by an overlapping set of authors, and mechanistically the components of the technology are the same, only changed by an impactful mutation in the 1POK and truncations to the linker and MBP tag. While these are important changes, I have concern for the progression of the field that presenting this work as a new technology with a new name (it is a better name) will create unnecessary confusion. A more accurate framing of this work would be as an extension and improvement of the XRI approach.
2. While the authors demonstrate the core technology in many ways, the examples chosen do not, in my opinion, demonstrate anything that really benefits from the longer recording capacity or frankly anything that would not have been possible using simple live imaging of fluorescent proteins driven by the same promoters. In terms of the longer recording capacity, prior work on the XRI showed multi-day recordings as well. Here the overall time of the recording appears longer, but on closer inspection what has been extended is the delay time from the start of the recording to the first stimulation. Unless I have missed something, the stimulation events that are being recorded always occur within 2-3 days of the fixation and imaging. Dye switching is done over a longer range of intervals, but not the stimulated recording. It would have been a much more convincing demonstration of the increased temporal window to stimulate on days 3 and 15 of a 21 day recording, for instance rather than days 18 and 20 as the authors have shown. Can stimulation at day 3 vs 5 be distinguished 20 days later? Those types of questions would be more useful to test the technological advance.

Additionally, the authors chose to show this technology in cultured cells. While that is fine for an initial proof-of-concept demonstration, the conceptual approach has already been demonstrated. Given that this builds on previous work, I would hope to see examples where biological insight is possible that could not be obtained using simpler technology. In figures 5 and 6, the authors show potential stratification of response types among cells to different stimulations. However, given the access and timeline, this insight could have as easily been gained by live imaging of fluorescent proteins driven by the same promoters used for the recording. It would critically help readers understand the potential of the technology if it were used for something that is otherwise impossible or impractical.

Minor Concerns:

- While most interpretations of the results are supported by the data, Lines 605-608 point to differences across cells that the authors interpret as a biological difference in transcriptional dynamics. I do not think that the authors can rule out that the differences are due to technical differences in the growth of the tape or reconstruction of the signal from imaging, so to claim biological insight is an over interpretation of the data without some other type of evidence supporting a functional difference in those cells. The authors should scale back these conclusions, acknowledging the potential technical contribution, and similarly be careful when referring to this finding elsewhere in the manuscript.

- While the authors should be commended on mostly clear writing, there are a few points where unnecessary jargon is introduced. Line 110-111 “spatiotemporally scalable recording” and line 472 “temporally scalable” are two examples. This seems to mean that it can be used for different durations and with events at different times. That is the entire conceptual utility of the approach, so adding these terms does not add meaning, just hype. Similarly, such ability to record different timings is not a “unique capability” (lines 472-475), but the assumed utility.

- The authors should strive for completeness when discussing other recording technologies. For instance, almost all nucleic acid recording technologies are cited, but CRISPR integrase-based recordings are omitted (Schmidt et al., Nature 2018; Bhattarai-Kline et al., Nature 2022) and Hao et al. bioRxiv 2024, while still a preprint, would be useful to discuss as it is nucleic acid based, but recovered by imaging.

- The initial explanation of lateral accumulation/width would be helped by the inclusion of a schematic of the molecule highlighting the dimensions and example images to accompany Fig 1d showing the quantification of width.

- Paragraph 773 discusses advantages of protein recording over nucleic acid-based recording technologies, which is reasonable, but the authors should take care not to be overly biased. For instance, the authors say that protein-based preserved the integrity of “cellular content.” It is not at all clear what that means. Additionally, there are nucleic acid-based recordings that have been recovered spatially, so to include the advantage of the spatial context requires a more nuanced statement. Also mentioned is that protein-based rather than nucleic acid-based eliminates the need for high-throughput sequencing or custom instrumentation. Sequencers are hardly a major barrier these days and there are multiple Illumina and nanopore sequencers that can be purchased at a fraction of the cost of the spinning disk confocal microscope that was used in this work. Moreover, sequencing can be outsourced as a service, while it is unlikely that one could easily get high quality images of these samples as a service from a commercial vendor.

- Paragraph 806 discussing limitations of the technique could be more detailed. For instance, the authors discuss scalability in many places throughout, but confocal imaging will inherently put a substantial limitation on the number of cells. Most data in this manuscript is based on 10s of cells per condition, while typical sequencing approaches are based on 10s-100s of thousands of cells.

(Remarks on code availability)

Referee #2

(Remarks to the Author)

This manuscript presents CytoTape, an intracellular recorder derived from the iPOK(E239Y)-based XRI system, enabling long-term, multiplexed analog tracking of gene regulation dynamics in live mammalian cells. The authors improved previous timestamp technology and developed thin, flexible assemblies that enable long-term assembly growth in cells for many days. CytoTape records promoter-driven transcriptional activity over weeks with single-cell resolution and up to five signals simultaneously. Compared to the previous XRI version, the CytoTape design achieves thinner, more flexible fibers with apparent improvements in signal detection. Protein engineering was partially successful. The data looks nice. However, CytoTape is essentially a modified and improved version of the XRI system from the author’s prior Nature Biotechnology paper (doi:10.1038/s41587-022-01586-7). The contribution of AI is marginal and overstated. There is no new significant biological discovery arising from this technology. Therefore, the conceptual novelty doesn’t meet Nature’s standard but could fit a specialized journal if the following issues are addressed.

Major Concerns

1. The fiber width readout shows little difference between B1 and C4 over time (7d, 14d, 18d). This suggests that removing the N-terminal four residues from MBP in C4 had no positive effect. The authors compare C4 to XRI (Line 229 to 300), which is inappropriate since C4 was derived from B1. To test whether MBP trimming reduces lateral growth, C4 should be directly compared to B1, not XRI, which contains many other differences.

2. From the description in Lines 301–305, the only difference between C4 and CytoTape is the removal of a 2-amino-acid linker; both carry the L349K mutation (retained after Line 259–260). Therefore, the final gain in morphology (thinner, bendable fiber) likely comes from linker removal alone. If L349K had minimal effects (as seen from A2), and all later designs retained it, then it is the linker, rather than other mutations, that contributes most to the CytoTape phenotype.

3. Following point #2, the authors should summarize the structures and derivation paths of all designs (XRI, A1–A3, B1, C1–C5, and CytoTape) in a table (e.g., in Figure 1). The narrative is difficult to follow, and some lines (e.g., 259–260) are prone to misinterpretation. This table should clarify which mutation or deletion each design includes and from which parent it was derived. The role of AI is overstated, CPDiffusion did not contribute to the final design, and the only AI-assisted mutation (L349K via ProtSSN) had limited effect. Therefore, the abstract’s claim of ‘AI-guided rational design’ (Line 38–40) is not justified.

4. The predicted structural model (Fig. 1f, 1g) requires further validation. Please provide the original AlphaFold3 output to examine confidence levels and support the hypothesis regarding L349K's position and its functional relevance.
5. In Figure 2b (left), both XRI and CytoTape show a similar elongation rate until day 14, after which CytoTape accelerates. This coincides with increased curvature seen at day 14 (Fig. 2a bottom). If curvature increases elongation speed, fiber growth becomes nonlinear. This could compromise the accuracy of using CytoTape to compare transcriptional activity before and after day 14. The authors should clarify whether the system can reliably record analog signals under non-linear growth conditions.
6. In Figure 2d (the V5 experiment), the baseline Egr1 expression (indicated by V5) is dramatically different between XRI and CytoTape groups. It seems that in the XRI condition, neurons are already activated at the time of HA and V5 transfection, triggering IEG expression, while in CytoTape neurons the baseline is nearly silent before KCl addition. If this quantitative difference is real, it implies that either a) XRI increases the firing probability of neurons at baseline, or b) CytoTape suppresses neuronal activity. Which hypothesis is correct? Furthermore, in Figure 4f, Egr1 expression over time looks normal, with a higher baseline from day 0 to day 5 than in Figure 2. This inconsistency adds confusion. In the XRI paper (doi:10.1038/s41587-022-01586-7), Figure 1 and Figure 4 showed better XRI overall data quality than this manuscript. Therefore, the experiment shown in Figure 2d should not be used to evaluate neuronal health after transfection. If the authors aim to support the hypothesis that XRI harms neurons while CytoTape does not, proper electrophysiological experiments should be conducted. These could include:
 - a) neuronal firing ability measured via patch clamp recordings,
 - b) synaptic transmission tested through EPSCs/IPSCs, or alternatively calcium imaging to assess activity patterns.
7. The CREB-Fos experiment is interesting. The unexpected decoupling of CREB and Fos activity in some cells could reflect unhealthy states or unidentified regulatory mechanisms. Could the decoupling reflect inhibition, delayed activation, or artifacts of labeling? Moreover, in coupled cells, can CytoTape resolve whether CREB precedes Fos expression? Given that the CREB signal represents expression level—not phosphorylation—the authors should discuss how they interpret CREB activity and whether it reflects pCREB or total CREB.
8. Building on point #7, what is the temporal resolution of CytoTape in practice? Can it resolve transcriptional events at hourly or sub-hourly resolution, or is it limited to multi-day differences? Time-course validation is needed to determine the minimum resolvable interval between transcription events. If CytoTape can distinguish transcription at sub-daily resolution, it would be a powerful tool for studying gene regulatory networks.

Minor Comments

1. In Figures 2a and 2d, Nissl staining fails to clearly show neuronal morphology. Without cell boundaries, it's hard to determine whether CytoTape deforms membranes or alters morphology. Please consider beta-III tubulin or surface markers to visualize cell outlines.
2. Typo corrections. Line 162: bioorthogonal should be 'bioorthogonal'. Line 42: 'to be readout at scale' should be 'to be read out at scale'
3. The authors state that they bleached JF dyes before immunostaining (Line 1198) to avoid spectral crosstalk. However, evidence for complete bleaching is not shown. Please provide supplementary data verifying complete dye removal. Additionally, with closely spaced channels (e.g., 561 and 594 nm), crosstalk is likely. How did the authors address this in multiplexed imaging? What is the maximum number of distinguishable signals the system can currently resolve?
4. After JF dye bleaching and antibody staining, how do the authors re-identify the same fibers for imaging? Did they use stage-coordinates, some markers, or software-based registration? Please describe this process in the Methods.

(Remarks on code availability)

Referee #3

(Remarks to the Author)

I co-reviewed this manuscript with one of the reviewers who provided the listed reports.

(Remarks on code availability)

Referee #4

(Remarks to the Author)

Zheng and coworkers report on a new method for "recording" multiple transcriptional events at a single-cell level over weeks. I am not an expert on the cell biology aspects that are central to the work, but as a non-expert, this sounds like a remarkable development. My expertise is in computational protein design, and the first part of the work relates to some protein engineering efforts on which I comment below.

As the rationale for the protein engineering effort, the authors write that a long and rigid assembly would either deform the cell or break up. What lengths did they reach, and how do those relate to the typical length of the cells they tested in this paper?

The first design attempt using CPDiffusion failed to produce long fibres, and they abandoned it. The authors nevertheless

state that the results show the ability of the method to design fibres and suggest that further work would fix it. I think that a balanced, facts-based way of reporting the results would state that the effort failed to produce the desired results and that future work to improve CPDiffusion may be necessary.

The end result of the computational design work is a single-point mutation Leu349Lys that induces higher solubility of the lateral surface of the fibre where growth was not desired. Given the very modest contribution of computational design to the work, I recommend toning down the remarks on AI-guided protein design. Solubility design is an extremely mature field, and introducing a single-point mutation -- though it had an important effect here -- is hardly innovative. The paper feels rather long, and I recommend focusing on the innovative aspects in the main results. Because there was no technical innovation in protein design, I would briefly explain the rationale for the solubility mutation without describing the method and remove the CPDiffusion section entirely -- it didn't contribute to the work. I would also remove the reference to AI-based design from the abstract (and other sections) because the effort here relied much more on run-of-the-mill protein engineering methods.

If I understand correctly, the AlphaFold3 modelling shown in Figure 1 is only illustrative. Yet, it's not clear to me that the AF3 reliability scores for the interaction between the fused domains are high enough for a model to be presented. The authors could check the PAE, iPTM and pLDDT scores at the interface to verify this. If the scores are not high, I think that the AF3 model should not be presented, and the authors should then only use their schematic representation.

As with other sections, the Discussion is very long. The last paragraph is mostly speculation on how future developments in protein design could improve the constructs, but I don't think that this goes beyond trivia. I think a much more focused discussion would be best.

The methods section on molecular dynamics simulations contained fewer technical details than I'm used to seeing in such sections. It would be good to verify with an expert that this section is complete.

GoogleScholar shows that CPDiffusion and ProtSSN have been published previously. If so, extended figure 1 on which describes the methods, isn't necessary.

(Remarks on code availability)

Version 2:

Reviewer comments:

Referee #1

(Remarks to the Author)

This substantially revised manuscript is an impressive work that now explicitly demonstrates the advantages of the technology that were hinted at in the original submission. Whether an improvement in an existing approach is appropriate for this particular journal is an editorial decision. Either way, I do think that this technology has clear value to the field.

I applaud the new experiments. I may be biased since it was performed at my suggestion, but I think the new results reported in Figure 5f-I are the best demonstration of the power of the approach. The recordings sharply show temporally precise recordings over an impressive span of time. I would only ask that the authors provide some example images to accompany these experiments in a supplemental figure given that I strongly think this will be a critical result that is discussed in labs around the world when evaluating the use of the approach, and while interpreting results if they adopt the technology. I also appreciate the more direct demonstration of advantages of standard timelapse FP imaging, highlighting the utility of a record the moves with the cell, as well as the in vivo data.

My last remaining concern is semantic, but it is something that I think is very important for the integrity of the literature. This technology is an improved version of a previously developed approach called XRI. Yes, the fibers are more controlled and that yields much better recording and potential for multiplexing and time-stamping. However, the technology is presented by the authors as though it is a fully new approach and given a new name, distinct from XRI. I takes an unnecessary amount of effort to determine how related the two approaches are since one was developed from the other, and that is bad for the health of the field of technology development and a user's ability to clearly evaluate which tools they should be using. I don't even mind that there is a new name. I think CytoTape is, much like the technology, better than the original. However, there is no reason to disguise the relationship. I would like to see one statement in the abstract and one in the introduction that clearly states something like "We improved upon a previous protein recording technology called XRI to create more temporally precise and multiplexable recording approach, which we now call CytoTape."

(Remarks on code availability)

Referee #2

(Remarks to the Author)

The revised manuscript and the authors' responses are carefully reviewed. Unfortunately, this reviewer is not convinced that the revised manuscript represents a significant advance in either technical or scientific dimensions. Although the authors demonstrate that CytoTape outperforms XRI, its core concept remains essentially unchanged; CytoTape appears to be an improved version of the XRI system rather than a new technology.

From a scientific standpoint, the reported decoupling of pCREB and c-Fos in some neurons is unexpected, but its biology mechanism is addressed. Without a clearly posed question, a testable hypothesis, and experiments designed to verify it, presenting only a descriptive observation is premature for publication and does not represent a significant scientific breakthrough. Importantly, there is also no evidence that this phenomenon is not an artifact of the in-vitro system. The newly added in-vivo data in Fig. 7 closely mirror the strategy and presentation of Fig. 4e-f in the published XRI Nature Biotechnology paper, not offering new advances.

Comments on specifics:

- Figure R13 shows only a marginal difference between C-4 and B1; three outliers in B1 drive a small shift, which is unconvincing.
- The revised summary is clearer.
- Figure R3 does not demonstrate recording beyond day 14 or long-term test.
- The authors claim CytoTape grows only horizontally, while XRI expands in all directions. Yet, in their published Nature Biotechnology paper (Fig. 1), they showed that XRI also grows horizontally, whereas iPOK(293Y) grows in all directions. The rationale behind these versions is essentially the same, and the progress is incremental: iPOK(293Y) is version 1 (First paper, Dr. Levy group, <https://www.nature.com/articles/nature23320>), XRI is version 2 (the authors' Nat. Biotechnology), and CytoTape is version 3 (or maybe 2.5). These similar iterations with different names make it difficult for readers to trace the previously published work.
- The decoupling of pCREB and c-Fos remains unexpected. More studies are needed to determine whether it is naturally occurring or induced by the artificial system, and question/hypothesis driven biology.
- Only systems with long-term, sub-daily resolution, and no functional influence on neurons and circuits are suitable for behavioral studies such as memory tracing. Functional validation in neurons/circuitry is needed before publishing.

(Remarks on code availability)

Referee #3

(Remarks to the Author)

I co-reviewed this manuscript with one of the reviewers who provided the listed reports.

(Remarks on code availability)

I co-reviewed this manuscript with one of the reviewers who provided the listed reports.

Referee #4

(Remarks to the Author)

The authors addressed my comments satisfactorily. Reading the other reviewers' remarks about precedents, I agree with them that the manuscript gives the sense of a new method, e.g., in the abstract. It's impressive work, even as an "upgrade" of an existing method, and it may be better to explain the gap the previous method left to be filled here.

(Remarks on code availability)

Version 3:

Reviewer comments:

Referee #1

(Remarks to the Author)

My concern about the clarity of this technology versus prior named technologies is satisfied in this newly revised version, and I appreciate the inclusion of images to support the new recordings with longer gaps between events.

Cross-reviewing for Reviewer 2, I find that the revised pCREB-Fos decoupling section is sufficiently explored and validated with an orthogonal approach, so that it is unlikely to be a simple artifact. For a demonstration of a technology, rather than a work of fundamental biology, the finding is detailed enough to show a potential use case and spur further investigation. Reviewer 2's other technical concern related to outliers driving an effect is appropriately addressed.

(Remarks on code availability)

Referee #3

(Remarks to the Author)

I co-reviewed this manuscript with one of the reviewers who provided the listed reports.

(Remarks on code availability)

Authors' Response:

We are grateful to the referees and the editor for the constructive comments and for the opportunity to submit our revised paper in response to the comments. Over the past few months, we have performed demonstration of the recording capability of CytoTape where previous protein ticker tape systems (XRI and iPAK4) and existing fluorescent protein imaging approaches fail, *in vivo* validation of CytoTape for weeks-long recording in living mouse brains, validation of biological insights identified by CytoTape on the cell-to-cell transcriptional heterogeneity and waveform coupling between CREB and Fos activities, demonstration of down to minutes-scale time precision and the ability to differentiate transcriptional events 12 h apart, three-week continuous recordings of multiple transcriptional events, quantitative analysis of the contributions of computationally identified mutations, and confidence analysis for AlphaFold3 protein structural modeling. Given the many new experiments and analyses in the paper, we are first presenting here a general overview of how the new paper differs from the original manuscript, before addressing the specific comments from the referees (highlighted in blue below) in full. In the revised manuscript, all the modifications are highlighted in red.

1. Benchmarking against XRI across multiple cell types: New data demonstrate that the prior XRI system fails to form stable, thin fibers in dividing cells (HEK, HeLa) and cannot record transcriptional events or preserve temporal information over multi-week durations in neurons due to lateral growth. In contrast, CytoTape forms thin and flexible fibers across multiple cell types and maintains long-term recording capability in both dividing and post-mitotic cells.

2. Benchmarking against iPAK4: iPAK4 fibers significantly perturb cell morphology and affect cell mitosis, whereas CytoTape fibers do not impact either process due to their flexibility.

3. *In vivo* validation: We performed *in vivo* validation of the CytoTape and the *in vivo*-optimized CytoTape-*vivo* systems in living mouse brains over 14 days of expression, showing significant reductions of fiber thickness compared to XRI, a key metric we established in earlier parts of this work, indicating minimized lateral growth and enhanced long-term recording fidelity. We further performed immunohistochemical characterization of mouse brains expressing CytoTape and found CytoTape expression in cell populations *in vivo* does not alter cellular and synaptic state markers, including NeuN as a neuronal marker, cleaved Caspase-3 as an apoptotic marker, GFAP as an astrocyte marker, Synaptophysin as a synaptic protein marker, Iba1 as a microglial marker, γ H2AX as a DNA damage marker, Hsp70 and Hsp27 as cell physiological stress markers, and *c-fos* protein as a cellular activity marker.

4. *In vivo* demonstration: We performed *in vivo* demonstration of CytoTape-*vivo* recording in living mice, achieving multiplexed recording of *c-fos* and Doxycycline (Dox)-induced transcriptional dynamics across large neuron populations (8,639 neurons in a single mouse; 13,101 neurons across 3 mice in total, read out via standard confocal microscopy within 2 hours of imaging time) in both the hippocampus and cortex in the same brain over 11, 14, and 18 days of CytoTape-*vivo* expression. This allows us to use the Dox ON and OFF signals controlled by drinking water as non-invasively introduced timestamps to achieve large-scale, multi-brain-region, spatiotemporally-resolved *c-fos* recording *in vivo* over 14 days (4,541 neurons were

analyzed from a single mouse brain) and 18 days (14,123 neurons were analyzed from a single mouse brain) of CytoTape-*vivo* expression. In addition, we developed a robust, user-friendly computational platform for high-throughput segmentation and analysis of fibers from large cell populations across brain regions *in vivo*.

5. Direct comparison with timelapse GFP imaging: CytoTape was benchmarked against timelapse GFP imaging for transcriptional dynamics. CytoTape can robustly distinguish transcriptional activation events separated 12 hours apart, while GFP cannot and instead report one slowly and continuously rising waveform, which may cause experimenter's incorrect interpretation of the result. Due to accumulation of GFP molecules over time in cells, the GFP reporter assay suffers from significant artifact from baseline drifting effect, showing baseline increases of GFP signal even in the absence of transcriptional activation. CytoTape does not have these artifacts, as well as other well-documented issues in live FP imaging including photobleaching effect and large spectral overlaps when multiplexing is needed, due to CytoTape's unique recording mechanism. Furthermore, *in vivo* GFP timelapse imaging suffers from limited field of view sizes from the *in vivo* imaging objectives (due to the intrinsic tradeoff between resolution and scale in *in vivo* imaging systems), limited imaging access to deep tissue due to tissue absorption and scattering of light, and photobleaching issues over time. As a result, most biologists can only image *in vivo* one or two color(s) of FPs in less than thousands of single cells that are located at most hundreds of microns deep in tissue, for a few hours continuously before the FP being photobleached. In contrast, the CytoTape toolkit enables large-3D-scale, long-term recording of multiple cellular transcriptional events in multiple deep and shallow brain regions *in vivo*.

6. Biological insight enabled by CytoTape and CytoTape-*vivo*: Using CytoTape, we discovered unexpected decoupling between CREB and Fos transcriptional dynamics in single cells following stimulation—findings further confirmed by direct immunostaining readout of snapshots of CREB and Fos activities in time. Using CytoTape-*vivo*, we identified heterogeneity of *c-fos* activity under seizure behavior at single-cell resolution across the hippocampus and cortex in living mice.

7. Three-week continuous recordings of multiple transcriptional events: We tested a stimulation condition—15 μ M forskolin (FSK) for 1 h on both day 3 and day 5, representing closely spaced dual stimulations occurring early in the recording window—during 21-day recording with CytoTape in cultured neurons. CytoTape successfully recorded both events and their continuous temporal waveforms. We also tested CytoTape with widely spaced dual stimulations—15 μ M FSK for 1 h on day 3 and 1.5 h on day 15. As expected, CytoTape successfully recorded both events and their continuous waveform features, such as the transcriptional activation durations and amplitudes that correlated with the increasing stimulation strengths. These results, together with the 21-day multiplexed recording results in the original manuscript, demonstrate that CytoTape enables continuous analog recording of multiple cellular events over many days and weeks.

8. Temporal resolution enhancements: New timestamping experiments demonstrate that CytoTape achieves minutes-scale resolution using sequential HaloTag labeling resolved by

standard confocal microscopy. We also demonstrated transcriptional events separated by 12 hours, and their continuous waveforms are clearly and robustly resolved, validating the temporal precision of CytoTape for continuous waveform recording.

9. Single-cell recording across large spatial scale: We added large-field confocal imaging of thousands of HEK cells to demonstrate recording scalability within 10 mins. We also validated that CytoTape can record distinct events across multiple regions in the living mouse brain at single cell resolution. These results validate CytoTape's ability to combine single-cell-resolution, multiplexed recording with population-scale data acquisition both in cell culture and *in vivo*.

10. Clarified structural contributions of key mutations: New comparative analyses clarified the respective roles of the computational identified L349K mutation and the linker truncations in improving CytoTape morphology. Mutants lacking L349K formed thicker fibers, confirming its contribution to lateral growth suppression.

11. Structural model confidence analysis: We performed AlphaFold3 model confidence evaluation (PAE, pLDDT, iPTM) to further support structural interpretations related to monomer design and domain interaction.

Referee #1's comments and authors' responses, Pages 5-27

Referees #2 and #3's comments and authors' responses, Pages 28-47

Referee #4's comments and authors' responses, Pages 48-54

Referees' comments:

Referee #1 (Remarks to the Author):

This manuscript by Zheng et al. presents a technical step forward in temporal/analog molecular recording via expanding protein tapes. The technology is interesting and potentially useful, the experiments are high-quality, and the overall presentation of results is clear despite substantial complexity. The major advance here is additional engineering of the repeating unit that makes up the recording filament to increase flexibility, which allows for longer assemblies (and thus temporally longer recordings) without as much deformation of the cell. My major concerns are whether this engineering warrants reintroduction of the approach under a new name, rather than XRI v2, and whether the authors have demonstrated any uses that could not be achieved using older, established approaches. Other concerns are relatively minor.

Reply: We thank the referee for recognizing the technical significance, high experimental quality, and clarity of the presentation of our work. We have addressed the major concerns below.

Major Concerns:

1. The efforts to create a thinner, more flexible recording filament are interesting, and clearly yielded an improved technology. However, this is conceptually the same approach presented as XRI in 2023 by an overlapping set of authors, and mechanistically the components of the technology are the same, only changed by an impactful mutation in the 1POK and truncations to the linker and MBP tag. While these are important changes, I have concern for the progression of the field that presenting this work as a new technology with a new name (it is a better name) will create unnecessary confusion. A more accurate framing of this work would be as an extension and improvement of the XRI approach.

Reply 1: While CytoTape is engineered from the XRI system, the thread-like flexible tape concept, the multiplexing concept, and new kinds of experiments it enables (via its unique tech specs including minutes-scale precision, 21-day recording duration, 8 colors encoding time and signals on one fiber, and working in dividing cells) are uniquely novel and biologically impactful, which are not achievable by the XRI system. We respectfully highlight several key advances that, together, we believe warrant introduction of CytoTape as a distinct and unique platform:

CytoTape functions robustly in dividing cells where XRI fails: In HEK cells, XRI forms intracellular puncta and thick fibers, while in HeLa cells it produces disordered, interwoven bundles (Fig. 2f). Due to these disordered XRI structures, the order of temporal events is no longer preserved (**Fig. R1**, left; HEK cells). These results demonstrate XRI's incompatibility with proliferative cell types. In contrast, CytoTape consistently forms thin and flexible fibers in both HEK and HeLa cells without perturbing cellular morphology (Fig. 2 and 3, Extended Fig. 3), enabling stable recording in cellular contexts that were previously inaccessible by XRI.

CytoTape enables continuous, multi-week analog recording where XRI fails: Due to its lateral binding over time, XRI cannot achieve weeks-long recording (Fig. 2d, upper panel). In contrast, CytoTape grows linearly and continuously over multiple weeks (Fig. 2d, lower panel), while preserving normal cell architecture via its flexibility and achieving continuous, waveform feature-resolved signal recording over three weeks (**Fig. R2**). Thus, the flexible CytpTape fibers accommodate intracellular constraints and permit continuous analog recording far beyond XRI's recording timescale.

CytoTape incorporates a HaloTag-based timestamping mechanism, enabling precise temporal reconstruction of transcriptional activity at single-cell resolution where XRI cannot achieve: XRI lacked a method for temporally annotating fiber segments at the single-cell level, resulting in a ~1 day scale time precision. In contrast, CytoTape uses sequential HaloTag dye labeling to precisely timestamp fibers within individual cells (Fig. 3, Extended Data Fig. 3), with precision down to minutes-scale (**Fig. R3**). This enables direct and precise reconstruction of the time axis at single-cell resolution, which is essential for resolving gene regulation dynamics in individual cells, and these capabilities are not accessible with XRI.

CytoTape supports multiplexed transcriptional recording of five independent signals at single-cell resolution: While XRI records one transcriptional event at a time, CytoTape supports simultaneous recording from five distinct transcriptional reporters simultaneously in single cells (Fig. 5o-r), each spatially encoded within the same fiber (with three distinct colors for timestamps; totally 8-color multiplexing on one fiber to encode both the signal and time information). This enables multi-dimensional reconstruction of single-cell gene regulation dynamics in cell populations.

In summary, CytoTape introduces both structural innovation (flexible, non-disruptive assemblies) and functional innovation (weeks-long recording duration, precise timestamping, five-signal multiplexing, broad compatibility across cell types (HEK, HeLa, neuron, and glial cells)) that unlock new biologically significant use cases not achievable with XRI (Fig. 6, Extended Data Fig. 10). We also summarized the performance of XRI and CytoTape below for a clearer side-by-side comparison (**Table R1**). We have retained the names CytoTape (and CytoTape-vivo for *in vivo* use) in the current manuscript to reflect their distinct engineering features and functional capabilities. That being said, we respectfully defer to the editor's recommendation regarding nomenclature.

Fig. R1 XRI loses temporal information in HEK for 3 days and cultured neurons for 15 days. Left panel: JF₅₈₅ and JF₆₃₅ were added to HEK cell cultures on day 0 and day 2, respectively, and cells were fixed on day 3 after transfection. Scale bar, 50 μ m. Right panel: JF₅₈₅ and JF₆₃₅ were added to neuronal cultures on day 5 and day 10, respectively, and neurons were fixed on day 15 after transfection.

Fig. R2 Three-week-long continuous analog recording of *c-fos*-promoter-driven expression by CytoTape in cultured neurons. (a) Left panel: *c-fos* signal relative change from baseline plotted against recovered time after calcium phosphate transfection. Right panel: statistical analysis of *c-fos* signal relative change from baseline plotted against recovered time after calcium phosphate transfection. 15 μ M FSK was added to the culture on day 3 and day 5, with an incubation time of 1 h. Data are from $n = 16$ CytoTapes from 16 neurons from five cultures. Thick centerline, mean; lighter boundary, s.d.; thin gray lines, individual CytoTape data. (b) Statistical analysis of the FWHM (full width at half maximum; from (a)) between the sequential V5 signals. n.s., not significant; Wilcoxon signed-rank test. Middle line in box plot, median; box boundary, interquartile range; whiskers, minimum and maximum, not indicated in the box plot; black dots, individual data points. (c) Left panel: *c-fos* signal relative change from baseline plotted against recovered time after calcium phosphate transfection. Right panel: statistical analysis of *c-fos* signal relative change from baseline plotted against recovered time after calcium phosphate transfection. 15 μ M FSK was added to the culture on day 3 and day 15, with an incubation time of 1 h for the first stimulation and 1.5 h for the second stimulation. Data are from $n = 10$ CytoTapes from 10 neurons from five cultures. Thick centerline, mean; lighter boundary, s.d.; thin gray lines, individual CytoTape data. (d) Statistical analysis of the FWHM (from (c)) between the sequential V5 signals. **, $P < 0.01$; Wilcoxon signed-rank test. Middle line in box plot, median; box boundary, interquartile range; whiskers, minimum and maximum, not indicated in the box plot; black dots, individual data points.

Fig. R3 CytoTape shows minutes-scale temporal resolution. (a) Schematic of the constructs transfected into cells. (b) Time points of JF₅₈₅, JF₆₃₅, and JF₅₀₃ addition and cell fixation. (c) Confocal images of timestamped fibers labeled with different dye-switching intervals. The dashed lines indicate the transient dye switching. The confocal images were taken using a spinning disk confocal microscope with a 60X objective.

Table R1: Comparison of XRI and CytoTape in multiple cell types in cell culture and *in vivo*

Tech Specs	XRI	CytoTape (this work)
Morphology in cultured neurons	fiber	fiber (functional)
Morphology in HEK cells	puncta and fiber (not functional)	fiber (functional)
Morphology in HeLa cells	puncta and fiber (not functional)	fiber (functional)
Morphology in cultured glial cells	not tested	fiber (functional)

Time recovery method in cell culture	Time calibration via tamoxifen-induced Cre/FLEX system	Timestamps via HaloTag and dye switches
Time recovery method in in vivo	not tested	Dox-inducible system (Tet-On)
Timestamp precision in cell culture (hour)	24	0.5
Longest recording duration reported in cultured neurons (day)	7	21
Longest recording duration reported in HEK cells (day)	not tested	5
Longest recording duration reported in HeLa cells (day)	not tested	4
Number of developed recorders for distinct cell physiological activities	1 (c-fos transcriptional activity)	7 (c-fos , Arc , Egr1 , and HSPA1A transcriptional activities; CREB, Npas4, and Fos protein activities)
Reported number of simultaneously recorded distinct cell physiological activities	1	5

2. While the authors demonstrate the core technology in many ways, the examples chosen do not, in my opinion, demonstrate anything that really benefits from the longer recording capacity or frankly anything that would not have been possible using simple live imaging of fluorescent proteins driven by the same promoters. In terms of the longer recording capacity, prior work on the XRI showed multi-day recordings as well. Here the overall time of the recording appears longer, but on closer inspection what has been extended is the delay time from the start of the recording to the first stimulation. Unless I have missed something, the stimulation events that are being recorded always occur within 2-3 days of the fixation and imaging. Dye switching is done over a longer range of intervals, but not the stimulated recording. It would have been a much more convincing demonstration of the increased temporal window to stimulate on days 3 and 15 of a 21 day recording, for instance rather than days 18 and 20 as the authors have

shown. Can stimulation at day 3 vs 5 be distinguished 20 days later? Those types of questions would be more useful to test the technological advance.

Reply 2: We respectfully clarify the key advantages of CytoTape over traditional live imaging approaches using fluorescent proteins (e.g., GFP) as follows. **Spatial scalability recording across cell population and *in vivo*:** CytoTape offers superior spatial scalability compared to live imaging or time-lapse GFP imaging for tracking gene expression across large cell populations over extended time. While GFP imaging requires continuous microscopy and is constrained by the intrinsic trade-off between spatial scale and spatial resolution, photobleaching effect, and motion artifacts in dividing/migrating cells and *in vivo* (*Histochemistry and Cell Biology* 2022, 158, 301-323; *Mol. Cell* 58, 644-659; *Chem. Soc. Rev.* 2009, 38, 2887-2921), CytoTape addresses these issues by *in situ* recording of transcriptional dynamics within intracellular protein assemblies to be read out post hoc. This decoupling from real-time imaging enables simultaneous analysis of large numbers of cells across broad spatial regions using standard post-fixation imaging techniques, making CytoTape particularly well-suited for high-throughput, single-cell-resolved studies over large tissue volumes (**Fig. R4** and **R5**, and Extended Data Fig. 12 and 13 in the revised main text). We further performed immunohistochemical characterization of mouse brains expressing CytoTape-vivo and found CytoTape-vivo expression in cell populations *in vivo* does not alter cellular and synaptic state markers, including NeuN as a neuronal marker, cleaved Caspase-3 as an apoptotic marker, GFAP as an astrocyte marker, Synaptophysin as a synaptic protein marker, Iba1 as a microglial marker, γ H2AX as a DNA damage marker, Hsp70 and Hsp27 as cell physiological stress markers, and *c-fos* protein as a cellular activity marker (**Fig. R6**). **Long-term recording without continuous imaging:** Live-cell GFP imaging requires sustained access to advanced microscopy systems and constant environmental control, often introducing phototoxicity and photobleaching over time (*IUBMB life* 2009, 61, 1029-1042; *Chem. Soc. Rev.* 2009, 38, 2887-2921). CytoTape, by contrast, continuously and passively encodes gene expression dynamics intracellularly over days to weeks, without requiring real-time monitoring or extended imaging sessions. **Multiplexing fidelity:** Different fluorescent proteins require different amounts of time to begin exhibiting fluorescence (e.g., ~1-2 h for EGFP; ~3-4 h or more for red-shifted FPs) (*Nat. Methods* 2018, 20, 47-51), leading to delayed readouts for multiplexing. These discrepancies complicate multiplexed analysis. CytoTape has minutes-scale temporal resolution (**Fig. R3**) and can encode transcriptional activity into monomers (same structure) with unique epitope tags. This enables precise reconstruction of asynchronous events, independent of fluorophore kinetics, exceeding the practical spectral limit of live-cell FP imaging, which is typically limited to 2-3 non-overlapping channels due to significant crosstalk from FP's wide emission spectral windows (*Nat. Methods* 2005, 2, 905-909).

To further demonstrate that CytoTape can faithfully reveal hours- and days-scale transcriptional events that GFP live imaging cannot, we directly compared both methods in tracking CREB expression following forskolin stimulation. As shown in **Fig. R4**, we found that it becomes increasingly difficult to monitor single-cell GFP dynamics over time due to cell division in HEK cells, which disrupts lineage tracking. In contrast, CytoTape reliably recorded CREB transcriptional dynamics at single-cell resolution across thousands of cells with a single confocal image. Thus, compared to GFP, CytoTape enables large-scale, single-cell-resolved readout of gene expression kinetics over extended timescales. Furthermore, we tested the performance of

the Fos-GFP live imaging system under unstimulated conditions (**Fig. R7**). We observed that GFP fluorescence intensity gradually increased over time, even in the absence of stimulation (**Fig. R7**, upper panel). However, immunofluorescence staining showed that the actual Fos protein levels in cells were not detectably changed under the same conditions (**Fig. R7**, lower panel). This suggests that the GFP system has a significant “baseline drift” artifact likely due to the accumulation of GFP molecules in cells over time from the steady baseline expression (*Frontiers in Behavioral Neuroscience* 2024, 18, 1500794), which can interfere with interpretations of the actual gene regulation dynamics. In contrast, CytoTape exhibits a flat baseline in the absence of stimulation (Extended Data Fig. 4 and 7), minimizing background noise and improving the reliability of stimulus-induced signal recording. We also tested whether live GFP imaging can distinguish two stimulation events separated by a 12-hour interval. To minimize the impact of cell density on single-cell GFP signal tracing during live imaging, we seeded HEK cells at very low density. As shown in **Fig. R8**, CytoTape clearly resolves two distinct peaks, whereas GFP imaging does not. These results demonstrate that CytoTape offers superior temporal resolution compared to live GFP imaging. Thus, the above results demonstrate that CytoTape is far superior to live GFP imaging and indicate that the results shown in Fig. 5-6 and Extended Data Fig. 10 cannot be obtained by live GFP imaging.

We also respectfully clarify the key advantages of CytoTape over XRI by providing new results. In dividing cells, we previously showed that XRI forms puncta and thick or intertwined fibers in HEK and HeLa cells, (Fig. 2f) respectively, indicating that it is incompatible with stable recording in proliferative cell types. To further assess this limitation, we tested whether XRI can preserve temporal information in HEK cells. As shown in **Fig. R1** (left panel), XRI fails to form elongated fibers, indicating that it cannot support temporal recording in these cells. In neurons, we previously showed that XRI is unable to reliably record physiological signals over multiple weeks due to lateral binding that compromises spatial encoding (Fig. 2d). To further evaluate this, we applied XRI for recording long-term temporal information. As shown in **Fig. R1** (right panel), XRI failed to preserve temporal information. Collectively, these results demonstrate that, compared to XRI, CytoTape enables robust long-term transcriptional recording in both neurons and dividing cells.

Following the referee’s suggestion, we stimulated neurons on day 3 and day 5, or on day 3 and day 15, over a 21-day recording. As shown in **Fig. R2**, CytoTape readout 21 days later clearly resolved the two distinct transcriptional events corresponding to the two stimulations. This result demonstrates that CytoTape can reliably support long-term transcriptional recording. Fig. R1-R8, Table R1, and related discussions have been added into the revised main text and Supplementary Information.

Fig. R4 Comparison of CREB activity following forskolin stimulation recorded by time-lapse GFP imaging and CytoTape. (a) Schematic of the constructs transfected into HEK cells (left panel) and time points of stimulation (30 μ M FSK for 1 h) (right panel). (b) Confocal images of live imaging of GFP driven by the pCREB promoter. Scale bar, 100 μ m. (c) Schematic of the constructs transfected into HEK cells (left panel) and time points of JF₅₈₅ and JF₆₃₅ addition and fixation and stimulation (30 μ M FSK for 1 h) (right panel). The JF₅₈₅ and JF₆₃₅ dyes were used for labeling time within the CytoTape. (d) Low magnification images of CytoTape in HEK cells,

which is taken after fixation on day 3, and immunostaining against the V5 tag. The objective lenses used for the left panel, middle panel, and right panel were 10X, 20X, and 40X, respectively. Some fibers in these low-magnification images do not display clear timestamp transitions or CREB signals because the optimal image contrast to visualize these fluorescence features in each color channel varies across individual cells. Scale bar, 50 μm .

Fig. R5 CytoTape-vivo records simultaneously Dox-dependent and *c-fos* signal monomer expression histories across multiple brain regions *in vivo*. (a) Schematic of the CytoTape-vivo system for multiplexed and scalable recording of cellular physiological histories in living mouse brains. CytoTape-vivo is formed by structural monomers that create the intracellular protein assembly. Dox-dependent monomers, which encode both temporal information and chemically inducible gene expression information, are sparsely incorporated along this assembly to embed the time axis along the spatial dimension. Signal monomers encode cellular physiological activities and are also incorporated along the same assembly to achieve recording. (b) Upper left panel, the experimental pipeline for CytoTape-vivo *in vivo* application. AAVs were injected into the dorsal CA1 region of 3-month-old mouse brains on day 0, followed by seizure induction via kainic acid (KA) injection and subsequent fixation via 4% paraformaldehyde perfusion. Dox is administered via drinking water to induce Dox-dependent monomer expression (Dox ON). Replacing the Dox-containing water with plain water terminates the expression (Dox OFF). The preserved brains were sectioned coronally at 40 μm thickness and stained with anti-HA, anti-FLAG, anti-V5, and Nissl stain. Upper right panel, a series of low-

magnification coronal-view confocal images from a mouse brain expressing the CytoTape-vivo system. Lower panel, examples of computational segmentation of neuronal soma and CytoTape-vivo fibers from high-magnification volumetric images in DG. Each color represents an individual neuron or fiber. Details of the segmentation method are described in Methods. See **Supplementary Video 1** for full FOV. (c) The experimental timeline for multiplexing *in vivo*. Dox-containing drinking water was administered on day 10, seizure induction was performed on day 10, and the brain was fixed on day 11. (d) Left panel, a low-magnification multi-channel composite image of a representative brain section. Scale bar, 500 μm . Middle panel, maximum intensity projection of a 3.2- μm -thick volume in the region of interest in DG. Some CytoTape-vivo fibers extend beyond this volume in the Z (depth) dimension and are therefore not fully visible in these two-dimensional projections. Scale bars, 50 μm . Right panel, representative confocal images showing *c-fos*-promoter driven gene expression and Dox-dependent signal monomer expression in DG neurons. (e) Dox signals and *c-fos* signals in the cortex (left panel), CA1 (middle panel), and DG (right panel). $n = 604$ fibers from 600 layer V neurons from posterior parietal cortex; $n = 1,835$ fibers from 1,697 neurons from hippocampal CA1; $n = 11,205$ fibers from 6,342 neurons from hippocampal DG. Data collected from one mouse. (f) Amplitude of *c-fos* signals obtained from e. ***, $P < 0.001$; n.s., not significant; Kruskal–Wallis analysis of variance followed by Dunn’s post hoc tests. Middle line in box plot, median; box boundary, interquartile range; whiskers, 10-90 percentile; black dots, individual data points. See **Supplementary Table 2** for details of statistical analysis. (g) The experimental timeline for *c-fos* signal monomer recording *in vivo*. Dox-containing drinking water was administered on day 10, seizure induction was performed on day 10, Dox-containing drinking water was replaced with regular water on day 12, and the brain was fixed on day 14. (h) Left panel, Dox signal relative change from baseline over the 14-day experiment in the control and experimental group. Centerline, mean; shaded area, interquartile range (IQR). Right panel, amplitudes of Dox signals relative change from baseline in control (Ctrl; no Dox, no KA) and experimental (Exp; described in g) mice. ***, $P < 0.001$; n.s., not significant; Kruskal–Wallis analysis of variance followed by Dunn’s post hoc tests. Middle line in box plot, median; box boundary, interquartile range; whiskers, 10-90 percentile; black dots, individual data points. See **Supplementary Table 2** for details of statistical analysis. (i) Averaged *c-fos* signals relative change from baseline in control (Ctrl; no Dox, no KA) and experimental (Exp; described in g) mice. ***, $P < 0.001$; n.s., not significant; Kruskal–Wallis analysis of variance followed by Dunn’s post hoc tests. Middle line in box plot, median; box boundary, interquartile range; whiskers, 10-90 percentile; black dots, individual data points. See **Supplementary Table 2** for details of statistical analysis. For the control group, $n = 717$ neurons from hippocampal DG from one mouse. For the experimental group, $n = 4,514$ neurons from hippocampal DG from one mouse. (j) Recorded *c-fos* signal monomer intensity relative change from baseline in DG neurons under KA-induced seizure (the experiment is described in g). The time axis was reconstructed for each fiber using Dox ON (onset of Dox signal rise), Dox OFF (onset of Dox signal decay), and fixation times shown in g as timestamps. $n = 4,514$ neurons from one mouse. Dashed vertical line, time of seizure induction. (k) The experimental timeline for *c-fos* signal monomer recording *in vivo*. Dox-containing drinking water was administered on day 10, seizure induction was performed on day 10, Dox-containing drinking water was replaced with regular water on day 13, and the brain was fixed on day 18. (l) Recorded *c-fos* signal monomer intensity relative change from baseline in

CA1 neurons under KA-induced seizure (the experiment is described in **k**). The time axis was reconstructed for each fiber using Dox ON (onset of Dox signal rise), Dox OFF (onset of Dox signal decay), and fixation times shown in **k** as timestamps. $n = 14,123$ neurons from one mouse. Dashed vertical line, time of seizure induction.

Fig. R6 Immunohistochemical characterization of cellular and synaptic state markers in mouse brains expressing CytoTape-vivo. (a) Representative confocal images of brain slices from adult mice expressing CytoTape-vivo in the hippocampus in the right cerebral hemisphere following AAV injection. 3-month-old mice were injected with AAV9-CytoTape-vivo-HA at the

CA1 region in the hippocampus of the right cerebral hemisphere. When the mice reached 14 days after AAV injection, they were perfused with 4% PFA, and brains were sliced coronally at 40 μm in 1X PBS, and stained with antibodies against one of the cellular and synaptic markers below (see **b-j**) and against HA tag to label CytoTape-vivo fibers, together with DAPI to label nuclei. Staining intensities of cellular and synaptic markers in the hippocampus were imaged volumetrically using a 40x objective on a spinning disk confocal microscope, with identical imaging conditions, measured in ImageJ as the averaged fluorescent intensities of the fluorescent secondary antibodies over imaged fields of view (400 μm \times 400 μm \times 40 μm for each fields of view), and compared between the left hemisphere and the right hemisphere. Scale bars, 500 μm . (**b-j**, top) Representative confocal images of the CA1 region in the non-injected hemisphere (Ctrl) and CytoTape-vivo-injected hemisphere (CytoTape-vivo) in the brain slices stained with antibodies against each of the cellular and synaptic markers indicated, and DAPI. Scale bars, 50 μm . (**b-j**, bottom) Box plots of the quantifications for each of the cellular and synaptic markers between the non-injected hemisphere (Ctrl) and CytoTape-vivo-injected hemisphere (CytoTape-vivo); for each marker in each hemisphere, $n = 10$ fields of view (FOVs) from 3 mice. Middle line in box plot, median; box boundary, interquartile range; whiskers, minimum and maximum; black dots, individual data points. See **Supplementary Table 2** for details of statistical analysis. n.s., not significant; two-sided Wilcoxon rank sum test. (**b**) NeuN (a neuronal marker). (**c**) Cleaved Caspase-3 (an apoptotic marker). (**d**) GFAP (an astrocyte marker). (**e**) Synaptophysin (a synaptic protein marker). (**f**) Iba1 (a microglial marker). (**g**) γH2AX (a DNA damage marker). (**h,i**) Hsp70 and Hsp27 (cell physiological stress markers). (**j**) *c-fos* protein (a cellular activity maker).

Fig. R7 Spontaneous Fos signals increase in GFP recording without stimulation. (a) Confocal images of live imaging of GFP driven by the F-RAM promoter. Scale bar, 100 μm . (b) Confocal images of Fos protein immunostaining in cells fixed at 24 hours (left panel) and 48 hours (right panel) after plasmid transfection. Scale bar, 50 μm .

Fig. R8 CytoTape captures dual CREB stimulation undetectable by time-lapse GFP imaging under low dose forskolin treatments. 10 μ M FSK was added to the culture at 50 and 62 hours after plasmid transfection, with each stimulation lasting 1 hour. GFP time-lapse images were captured every 2 hours. GFP imaging was performed in 31 HEK cells, and CytoTape recording in 60 HEK cells.

Additionally, the authors chose to show this technology in cultured cells. While that is fine for an initial proof-of-concept demonstration, the conceptual approach has already been demonstrated. Given that this builds on previous work, I would hope to see examples where biological insight is possible that could not be obtained using simpler technology. In figures 5 and 6, the authors show potential stratification of response types among cells to different stimulations. However, given the access and timeline, this insight could have as easily been gained by live imaging of fluorescent proteins driven by the same promoters used for the recording. It would critically help readers understand the potential of the technology if it were used for something that is otherwise impossible or impractical.

Reply 3: To further demonstrate that our design strategy generates thin and flexible protein fibers, we injected CytoTape and XRI into the mouse brain and quantified fiber thickness (**Fig. R9**). CytoTape forms significantly thinner fibers compared to XRI. This result demonstrates that our design strategy enables the formation of structurally optimized fibers not only in cell culture but also *in vivo*.

To address the concern that “the biological insights presented in Fig. 5 and 6 might also be attainable using fluorescent protein (FP)-based live imaging”, we have included new data and clarified the unique capabilities of CytoTape that extend beyond the reach of conventional methods. As shown in **Fig. R4**, tracking single-cell GFP expression becomes challenging in dense cell populations due to cell movement and division, which may introduce optical artifact signals from cellular motions and changes in cell volumes over time. **Fig. R7** demonstrates that baseline GFP expression in the absence of stimulation compromises the fidelity of stimulus-induced signal detection, showing a large and always increasing baseline drift artifact in single cells likely due to accumulations of GFP molecules over time after construct transfection. This is in agreement with our previous observations (*Nat. Biotech.* 2023, 41, 640-651, **Fig. R10**) using AAV-mediated construct delivery of GFP under *c-fos* promoter, showing a large baseline drift artifact over a week. Such a large baseline drift artifact may mask out signals with smaller amplitudes and introduce bias and uncertainty to signals with comparable amplitudes that may

or may not be actually increasing over time. As shown in **Fig. R10**, on day 7, the amplitude of the GFP baseline drift artifact signal in non-stimulated neurons is reaching the GFP signal amplitude in neurons under strong *c-fos* induction by 3 hours of KCl treatment. CytoTape instead provides a flat, stable baseline because CytoTape monomers are stably stored and embedded on the fiber, unlike FP molecules that remain free-floating in cells and the readout signal is proportional to the concentration of FP free-floating molecules that is sensitive to long-term molecular accumulation, degradation, and photobleaching in cells. While established cell lines with stable FP expression via genome integration of the FP gene may provide a more stable baseline FP signal, it would be impractical to generate a stable FP cell line for every new combination of a cell type and a transcription signal to record (not to mention when one hopes to perform multiplexed recording of a set of customly selected transcriptional signals). CytoTape provides flexibility and multiplexability for such recordings in a modular, programmable manner. Furthermore, **Fig. R8** shows that the GFP-based system cannot distinguish two stimulation events separated by a 12-hour interval and instead reports a slowly and continuously increasing trend, whereas CytoTape clearly resolves the two transcriptional events, reporting two distinct peaks and their continuous waveform features, such as peak height and peak width that correlate with increasing stimulation strength. These results collectively demonstrate that the biological insights enabled by CytoTape cannot be readily achieved through live imaging of GFP. To further validate the observed decoupling between CREB and Fos shown in Fig. 6, we performed immunostaining for CREB and Fos proteins at 1 hour and 24 hours after completion of forskolin (FSK) stimulation. As shown in **Fig. R11**, CREB and Fos snapshot signals were decoupled in a substantial number of cells as time went on, supporting the decoupling revealed by CytoTape, despite that the snapshot immunostaining method cannot provide continuous signal waveform over time in single cells, a key feature in CREB/Fos decoupling revealed by CytoTape. We have added Fig. R9 and R11 in the revised main text and related discussion in the revised Supplementary Information.

We further optimized CytoTape to CytoTape-vivo for *in vivo* application, enabling scalable, spatiotemporally resolved, single-cell analog recording of transcriptional dynamics across multiple brain regions (**Fig. R5**). Conventional *in vivo* fluorescent protein imaging techniques rely on establishing and maintaining imaging access to tissue and are limited by photobleaching time, spectral crosstalk of fluorescent proteins, restricted field of view sizes under high magnification objectives to resolve single cells, and limited imaging depth due to tissue absorption and scattering of light (*Histochemistry and Cell Biology* 2022, 158, 301-323; *Mol. Cell* 58, 644-659; *Chem. Soc. Rev.* 2009, 38, 2887-2921; *Nat. Methods* 2018, 20, 47-51; *Nat. Methods* 2005, 2, 905-909). As a result, most biologists can only image less than thousands of single cells that are located at most a few hundreds of micrometers deep in tissue, for a few hours continuously before the FP being photobleached (*Methods Cell Biol.* 2014, 123, 77-94). In contrast, CytoTape-vivo passively encodes gene expression into stable intracellular protein fibers, allowing retrospective and multiplexed readout without real-time imaging. By leveraging non-invasive doxycycline (Dox) administration via drinking water as a temporal marker (Dox ON and OFF), CytoTape-vivo enables time-resolved recording without requiring cranial access or dye injections. It faithfully captures asynchronous transcriptional events, such as kainic acid-induced *c-fos* activation (*Nature* 2012, 484, 381-385; *eLife* 2016, 5, e13918), and reveals spatially heterogeneous gene expression dynamics across the hippocampus and cortex.

Operating across large spatial (multi-brain-region) scales, CytoTape-vivo provides a powerful and broadly accessible platform for long-term, high-resolution recording of gene expression in the intact brain.

Fig. R9 Analysis of XRI and CytoTape fiber width in the hippocampus of the mouse brain *in vivo*. The brain was fixed 14 days after AAV injection. A total of 358 XRI fibers and 312 CytoTape fibers were analyzed for statistical analysis. ***, $P < 0.001$; Mann–Whitney U test. Middle line in box plot, median; box boundary, interquartile range; whiskers, minimum and maximum; black dots, individual data points.

[Figure Redacted]

Fig. R10 KCl stimulation of cultured neurons with c-fos promoter-driven expression of GFP. (a) Construct schematic of GFP under c-fos promoter and representative confocal images of live cultured mouse hippocampal neurons in the GFP channel 1-7 days (1d-7d) after AAV transduction, without (upper row) and with (lower row) 55 mM KCl stimulation for 3 hours on 5d. All images were captured under the same imaging condition. Scale bar, 10 μ m. (b) GFP fluorescence at soma (normalized by the average GFP fluorescence at soma over days 1-5) versus time (n = 11 neurons from 2 cultures for 'No Stim' group; n = 12 neurons from 2 cultures for 'KCl Stim' group). n.s., not significant; **, $P = 0.0038$; two-sided Wilcoxon rank sum tests with Holm-Sidak correction between 'No Stim' and 'KCl Stim' on day 6 or day 7 after AAV transduction. Centerline, mean; shaded boundary, standard deviation. (The figure is reproduced from Extended Data Fig. 7a,b in Linghu et al., *Nature Biotechnology*, 2023 <https://doi.org/10.1038/s41587-022-01586-7>).

a
Cells fixed 1 h after 50 uM FSK stimulation

b
Cells fixed 24 h after 50 uM FSK stimulation

Fig. R11 CREB and Fos activities exhibit complex cell-to-cell heterogeneity and become decoupled in a substantial fraction of cells following FSK stimulation, as indicated by activity snapshots from immunostaining against pCREB and Fos across HEK cells.

Confocal images of immunostaining fluorescence signals against pCREB and Fos in HEK cells fixed at (a) 1 hour and (b) 24 hours after the completion of a 1-hour stimulation with 50 μ M FSK. pCREB and Fos immunofluorescence intensities from the same cells were quantified pair-wise in scatter plots, where each dot represents a single cell ($n = 30$ cells from 2 cultures for each fixation time). Scale bar, 50 μ m.

Minor Concerns:

- While most interpretations of the results are supported by the data, Lines 605-608 point to differences across cells that the authors interpret as a biological difference in transcriptional dynamics. I do not think that the authors can rule out that the differences are due to technical differences in the growth of the tape or reconstruction of the signal from imaging, so to claim biological insight is an over interpretation of the data without some other type of evidence supporting a functional difference in those cells. The authors should scale back these conclusions, acknowledging the potential technical contribution, and similarly be careful when referring to this finding elsewhere in the manuscript.

Reply 4: We thank the referee for acknowledging that most interpretations of the results are supported by the data and for pointing out technical vs biological differences. In the original main text, we analyzed signal profiles across multiple CytoTape assemblies within the same neuron when more than one assembly was present (Extended Data Fig. 4n). These independent assemblies reported highly consistent transcriptional waveforms. This suggests that CytoTape robustly encodes transcriptional activity and that local fiber morphology does not substantially distort the recorded signal. In addition, time-lapse GFP imaging in cultured neurons (Extended Data Fig. 5) also shows heterogeneity following forskolin or KCl stimulation. Furthermore, in **Fig. R11** above we directly measured CREB and Fos activity snapshots 1 h and 24 h after the completion of FSK stimulation across individual HEK cells via immunostaining, which also shows large cell-to-cell heterogeneity of these transcription factor activities.

Given these validation results, we interpret the cell-to-cell variability recorded by CytoTape as reflecting underlying biological heterogeneity in transcriptional responses. This is also consistent with prior reports indicating that even genetically similar neurons can exhibit divergent gene expression dynamics in response to the same stimulus, particularly in pathways involving immediate early genes (*Current Opinion in Neurobiology* 2022, 75, 102568; *PLoS Biol.* 2016, 14, e1002585; *Nat. Commun.* 2018, 9, 3084). Nonetheless, we acknowledge the referee's comment that technical factors cannot be entirely ruled out, since currently we don't have a precise alternative methodology for continuously analog recording of gene regulation dynamics ground truth over time—that's why we are devoted to enable such measurement by developing CytoTape to address this critical technology gap in this field. Following the referee's suggestion, we have now added the above discussion in the revised main text to address this point.

- While the authors should be commended on mostly clear writing, there are a few points where unnecessary jargon is introduced. Line 110-111 "spatiotemporally scalable recording" and line 472 "temporally scalable" are two examples. This seems to mean that it can be used for

different durations and with events at different times. That is the entire conceptual utility of the approach, so adding these terms does not add meaning, just hype. Similarly, such ability to record different timings is not a “unique capability” (lines 472-475), but the assumed utility.

Reply 5: Following the referee’s suggestion, we have removed the terms “spatiotemporally scalable recording,” “temporally scalable,” and “unique capability” to improve clarity and avoid unnecessary jargon.

- The authors should strive for completeness when discussing other recording technologies. For instance, almost all nucleic acid recording technologies are cited, but CRISPR integrase-based recordings are omitted (Schmidt et al., Nature 2018; Bhattarai-Kline et al., Nature 2022) and Hao et al. bioRxiv 2024, while still a preprint, would be useful to discuss as it is nucleic acid based, but recovered by imaging.

Reply 6: We thank the referee for this helpful suggestion. In the revised manuscript, we have now included discussion of CRISPR integrase-based recording systems (Schmidt et al., Nature 2018; Bhattarai-Kline et al., Nature 2022) and the imaging-recoverable nucleic acid recorder described by Hao et al. (bioRxiv 2024). The revised discussion reads:

“Recent advances in nucleic acid-based molecular recording have enabled storage of physiological histories within the genome. CRISPR integrase-based systems such as Record-seq (Schmidt et al., 2018) and Retro-Cascorder (Bhattarai-Kline et al., 2022) achieve transcriptome-wide RNA-to-DNA conversion and barcode editing. Hao et al. (bioRxiv 2024) further preserve spatial and cellular contexts via in situ readout of nucleic acid barcodes, providing a single-signal readout per cell. This system establishes a scalable platform for genetic recording and in situ reconstruction of signaling histories in single cells, potentially advancing the quantitative analysis of cell–cell heterogeneity and communication during development and disease.

However, these approaches typically do not offer continuous time recording and are primarily optimized for use in prokaryotic cells or mammalian cell cultures. By contrast, CytoTape enables non-destructive, single-cell continuous recording of transcriptional dynamics with temporal resolution down to minutes-scale, recording duration for up to three weeks, and multiplexing capability to allow three color time encoding and five color transcriptional dynamics encoding, all on a single CytoTape fiber. CytoTape functions in both post-mitotic and dividing cells and does not require genome editing or sequencing-based recovery. Furthermore, CytoTape-vivo enables recording multiple events across multiple regions in the living mouse brain at single-cell resolution. While nucleic acid-based methods are powerful for transcriptome profiling and lineage tracing, CytoTape complements these toolkits by enabling use cases where continuous tracking of one or multiple transcriptional dynamics is needed in mammalian cells *in vitro* and *in vivo*.”

- The initial explanation of lateral accumulation/width would be helped by the inclusion of a schematic of the molecule highlighting the dimensions and example images to accompany Fig 1d showing the quantification of width.

Reply 7: Following the referee’s suggestion, we added the schematic of lateral binding (Fig. R12) and the quantification of fiber width (Fig. R13) in the revised Supplementary Information.

Fig. R12 Schematic of structural monomer binding kinetics in XRI and CytoTape during weeks-long recording. Dark green and light green rectangles represent previously incorporated and newly expressed structural monomers, respectively.

$$Thickness = \frac{\sum_{i=1}^n D_i}{n}$$

Fig. R13 Quantification of protein fiber width. The fiber was segmented and skeletonized using the algorithms described in the Methods section. The gray, tubular-shaped structure represents the fiber mask in 3D space, with the red line in the center indicating the extracted skeleton. Red dots mark representative points that are evenly distributed along the skeleton at intervals of 1.5 voxels. D_i denotes the minimum distance from each skeleton point (red dot) to the nearest boundary of the segmented fiber, illustrated by a dark gray arrowed line.

-Paragraph 773 discusses advantages of protein recording over nucleic acid-based recording technologies, which is reasonable, but the authors should take care not to be overly biased. For instance, the authors say that protein-based preserved the integrity of “cellular content.” It is not at all clear what that means. Additionally, there are nucleic acid-based recordings that have been recovered spatially, so to include the advantage of the spatial context requires a more nuanced statement. Also mentioned is that protein-based rather than nucleic acid-based

eliminates the need for high-throughput sequencing or custom instrumentation. Sequencers are hardly a major barrier these days and there are multiple Illumina and nanopore sequencers that can be purchased at a fraction of the cost of the spinning disk confocal microscope that was used in this work. Moreover, sequencing can be outsourced as a service, while it is unlikely that one could easily get high quality images of these samples as a service from a commercial vendor.

Reply 8: We thank the referee for their acknowledgement that our rationale on the advantages of protein ticker tape recording over nucleic acid-based recording is reasonable and for pointing out more specific and/or nuanced statements should be used for describing the advantages in preserving cellular content, preserving spatial context, and instrumentation. Before we address these comments below, we hope to take this opportunity to respectfully clarify that our intention is to develop a technology that complements the use cases of sequencing-based recorders and generates datasets that are likewise complementary to those obtained from sequencing-based recorders, thereby broadening the scope and impact of molecular recorder-based cell biology studies.

In the original manuscript, our reference to “preserving the integrity of cellular content” was meant to highlight that CytoTape does not require lysis, fixation-incompatible reagents, or nucleic acid extraction steps (*Nat. Biotechnol.* 2023, 41, 640–651). Instead, it maintains native cell morphology and intracellular architecture, enabling high-resolution spatial readout. This feature is especially important for future studies that requires pairing between the recording with other spatial features in cells and in tissue, such as the visualization cell type markers, correlation with synaptic plasticity, proximity of the recorded signals to local vasculature and local release of messengers, to name a few. This preservation of native cell content and spatial context is particularly critical for interpreting spatially resolved biological phenomena in heterogeneous tissues such as the brain (*Nature* 2023, 624, 343–354; *Cell* 2016, 166, 1028 - 1040; *Nat. Methods* 2021, 18, 997–1012). More specific and nuanced statements about DNA ticker tape and CytoTape are described in Reply 6 and updated in the revised manuscript.

We also revised the manuscript to present a more nuanced view on instrumentation. While sequencing is widely accessible and cost-effective, CytoTape’s compatibility with conventional microscopy provides a parallel, low-barrier alternative for contexts where preserving cell morphology and performing parallel phenotypic analyses are essential. We choose to use the spinning disk microscopy because it is readily available in our lab. We have also tested a laser scanning microscope available in a neighboring lab, another popular type of confocal microscope, and a routine wide-field epifluorescence microscope we have, and both can resolve the fibers well despite the routine wide-field epifluorescence microscope having less spatial resolution which may reduce the readout precision. Since many biology labs have confocal microscopes or have access to them via departmental shared equipment or institutional imaging core facilities, we do not believe the use of confocal microscopes or imaging techniques in general would be a major technical barrier to apply our technology. Our vision is that imaging can be scalable. In fact, a large portion of modern sequencers are imagers (e.g., almost all Illumina sequencers are imaging-based). We envision future efforts could combine protein recorders with high-throughput imaging techniques, perhaps drawing inspiration from imaging-based sequencers, to achieve large-scale readout of protein recorders at the throughput of

sequencers. That being said, we have now adjusted the relevant statements in the main text accordingly to maintain a balanced and objective tone.

-Paragraph 806 discussing limitations of the technique could be more detailed. For instance, the authors discuss scalability in many places throughout, but confocal imaging will inherently put a substantial limitation on the number of cells. Most data in this manuscript is based on 10s of cells per condition, while typical sequencing approaches are based on 10s-100s of thousands of cells.

Reply 9: To evaluate the scalability of the CytoTape toolkit under confocal imaging, we have now included *in vivo* data in the revised main text. As shown in **Fig. R5**, CytoTape-vivo recorded from 13,000+ cells across multiple regions in mouse brains while maintaining single-cell resolution and spatial context in tissue, all being read out within 2 hours of volumetric imaging time under the 40x objective in the Nikon CSU-W1 spinning disk confocal microscope, which was first commercially released in 2012. Although our vision is that imaging can be scalable (see rationale in Reply 8), our intention here is to develop a technology that complements the use cases of sequencing-based recorders and generates datasets that are likewise complementary to those obtained from sequencing-based recorders, thereby broadening the scope and impact of molecular recorder-based cell biology studies. We envision future efforts may even combine CytoTape with sequencing-based recorders—perhaps through imaging-based *in situ* sequencing (*Nucleic Acids Res.* 2020, 48, e112) plus CytoTape immunofluorescence—to generate synergistic datasets that leverage the strengths of both kinds of technologies. The above discussion and Fig. R5 have been added to the revised main text.

Referee #2 (Remarks to the Author):

This manuscript presents CytoTape, an intracellular recorder derived from the iPOK(E239Y)-based XRI system, enabling long-term, multiplexed analog tracking of gene regulation dynamics in live mammalian cells. The authors improved previous timestamp technology and developed thin, flexible assemblies that enable long-term assembly growth in cells for many days.

CytoTape records promoter-driven transcriptional activity over weeks with single-cell resolution and up to five signals simultaneously. Compared to the previous XRI version, the CytoTape design achieves thinner, more flexible fibers with apparent improvements in signal detection. Protein engineering was partially successful. The data looks nice. However, CytoTape is essentially a modified and improved version of the XRI system from the author's prior Nature Biotechnology paper (doi:10.1038/s41587-022-01586-7). The contribution of AI is marginal and overstated. There is no new significant biological discovery arising from this technology. Therefore, the conceptual novelty doesn't meet Nature's standard but could fit a specialized journal if the following issues are addressed.

Reply: We thank the referee for highlighting the significance of our work, commenting the data looks nice, and comparing CytoTape with the XRI system. While CytoTape is engineered from the XRI system, the thread-like flexible tape concept, the multiplexing concept, and new kinds of experiments it enables (via its unique tech specs including minutes-scale precision, 21-day recording duration, 8 colors encoding time and signals on one fiber, working in dividing cells, and multiplexed, multi-brain-region *in vivo* single-cell recording) are uniquely novel and biologically impactful, which are not achievable by the XRI system. While questions of novelty and significance are inherently subjective, we respectfully present arguments below for why this work constitutes a major advance in intracellular recording and is of broad interest to Nature's readers.

Technological innovation: CytoTape introduces a fundamentally new recording platform that overcomes key limitations of the original XRI system. While XRI was restricted by bulky, thick assemblies and cannot function in dividing cells, CytoTape features thin, highly flexible protein assemblies that remain stable and functional across multiple cell types (HEK, HeLa, neuron, and glial cells). This compatibility with proliferating cells represents a critical functional expansion, enabling applications in diverse biological systems where cell division is inherent—such as stem cells, cancer models, and developmental contexts. In addition, CytoTape supports multi-week, multiplexed, and continuous recording of transcriptional activities at single-cell resolution, with technical specifications that include down to minutes-scale temporal precision, up to 21-day recording duration, encoding of eight distinct temporal and transcriptional signals on a single fiber, and large-scale *in vivo* transcriptional recording. These capabilities go well beyond those of XRI and establish CytoTape as a versatile platform for long-term, high-resolution gene expression tracking across a wide range of biological systems.

Biological significance: The improved performance of CytoTape enables novel types of biological inquiry that were previously inaccessible, such as XRI, GFP and protein immunostaining. We demonstrate its ability to resolve the timing and co-expression patterns of multiple transcriptional programs over days in single cells, including in actively dividing populations. In Fig. 6, we used CytoTape to examine the temporal relationship between CREB and Fos, two canonical activity-regulated transcription factors. While CREB is often considered upstream of Fos in stimulus-responsive gene expression cascades, our recordings reveal that

this relationship is not always tightly coupled across cells or over time. Specifically, we observed that many cells exhibit strong CREB activity in the absence of a corresponding Fos response. This decoupling—difficult to capture in real-time due to variability and timing constraints—suggests that additional regulatory layers (e.g., inhibition, stress, epigenetic state) modulate the functional output of CREB signaling (*Science* 2001, 82, *pe1*; *PNAS* 2013, 110, 16645-16650; *Cell Rep.* 2018, 25, 2716-2728; *PNAS* 2020, 117, 23270-23279). This insight into divergent CREB-Fos coupling would have been difficult to uncover without continuous and multiplexed temporal recording across hundreds of single cells. In Extended Data Fig. 10, we simultaneously recorded *Arc* and *Egr1*, two immediate early genes (IEGs) critical for synaptic plasticity, learning, and memory consolidation. While both genes are known to be activity-dependent, the extent to which their transcriptional dynamics diverge or synchronize across individual neurons over multi-day stimulation has remained unclear. CytoTape revealed that *Arc* and *Egr1* are often co-expressed and can exhibit temporal or amplitude coupling in response to identical stimulation. The observed heterogeneity and coupling in their transcriptional response profiles across neurons—particularly during the late phases following repeated stimulation—suggest that *Arc* and *Egr1* may interact in encoding neuronal activity history and driving circuit reconfiguration (*Cell* 2020, 181, 410-423; *PNAS* 2020, 117, 23304-23310; *Neuron* 2018, 100, 330-348).

***In vivo* demonstration.** We further optimized CytoTape into CytoTape-*vivo* for *in vivo* applications, enabling scalable and multiplexed (Dox-dependent and *c-fos*-promoter-dependent transcriptional dynamics), spatiotemporally resolved, single-cell analog recording of transcriptional dynamics across multiple brain regions (**Fig. R5**). Conventional *in vivo* fluorescent protein imaging techniques rely on establishing and maintaining imaging access to tissue and are limited by photobleaching time, spectral crosstalk of fluorescent proteins, restricted field of view sizes under high magnification objectives to resolve single cells, and limited imaging depth due to tissue absorption and scattering of light. As a result, most biologists can only image *in vivo* one or two color(s) of fluorescent reporter (due to their wide spectrum windows) in less than thousands of single cells that are located at most hundreds of microns deep in tissue, for a few hours continuously before the fluorescent reports being photobleached. In contrast, CytoTape-*vivo* encodes cellular events into stable intracellular protein fibers, allowing retrospective and multiplexed readout without real-time *in vivo* imaging. By leveraging non-invasive doxycycline (Dox) administration via drinking water as a temporal marker (Dox ON and OFF), CytoTape-*vivo* enables spatially and temporally resolved multiplexed single-cell recording. Operating across large spatial scales, CytoTape-*vivo* (and the associated high-throughput, user-friendly image analysis algorithm) provides a powerful and broadly accessible platform for long-term, high-resolution recording of gene expression histories *in vivo*.

We further performed immunohistochemical characterization of mouse brains expressing CytoTape-*vivo* and found CytoTape-*vivo* expression in cell populations *in vivo* does not alter cellular and synaptic state markers, including NeuN as a neuronal marker, cleaved Caspase-3 as an apoptotic marker, GFAP as an astrocyte marker, Synaptophysin as a synaptic protein marker, Iba1 as a microglial marker, γ H2AX as a DNA damage marker, Hsp70 and Hsp27 as cell physiological stress markers, and *c-fos* protein as a cellular activity marker (**Fig. R6**).

Fig. R5 CytoTape-vivo records simultaneously Dox-dependent and *c-fos* signal monomer expression histories across multiple brain regions *in vivo*. (a) Schematic of the CytoTape-vivo system for multiplexed and scalable recording of cellular physiological histories in living mouse brains. CytoTape-vivo is formed by structural monomers that create the intracellular protein assembly. Dox-dependent monomers, which encode both temporal information and chemically inducible gene expression information, are sparsely incorporated along this assembly to embed the time axis along the spatial dimension. Signal monomers encode cellular physiological activities and are also incorporated along the same assembly to achieve recording. (b) Upper left panel, the experimental pipeline for CytoTape-vivo *in vivo* application. AAVs were injected into the dorsal CA1 region of 3-month-old mouse brains on day 0, followed by seizure induction via kainic acid (KA) injection and subsequent fixation via 4% paraformaldehyde perfusion. Dox is administered via drinking water to induce Dox-dependent monomer expression (Dox ON). Replacing the Dox-containing water with plain water terminates the expression (Dox OFF). The preserved brains were sectioned coronally at 40 μ m thickness and stained with anti-HA, anti-FLAG, anti-V5, and Nissl stain. Upper right panel, a series of low-magnification coronal-view confocal images from a mouse brain expressing the CytoTape-vivo system. Lower panel, examples of computational segmentation of neuronal soma and CytoTape-vivo fibers from high-magnification volumetric images in DG. Each color represents an individual neuron or fiber. Details of the segmentation method are described in Methods. See **Supplementary Video 1** for full FOV. (c) The experimental timeline for multiplexing *in vivo*. Dox-containing drinking water was administered on day 10, seizure induction was performed on

day 10, and the brain was fixed on day 11. **(d)** Left panel, a low-magnification multi-channel composite image of a representative brain section. Scale bar, 500 μm . Middle panel, maximum intensity projection of a 3.2- μm -thick volume in the region of interest in DG. Some CytoTape-*vivo* fibers extend beyond this volume in the Z (depth) dimension and are therefore not fully visible in these two-dimensional projections. Scale bars, 50 μm . Right panel, representative confocal images showing *c-fos*-promoter driven gene expression and Dox-dependent signal monomer expression in DG neurons. **(e)** Dox signals and *c-fos* signals in the cortex (left panel), CA1 (middle panel), and DG (right panel). $n = 604$ fibers from 600 layer V neurons from posterior parietal cortex; $n = 1,835$ fibers from 1,697 neurons from hippocampal CA1; $n = 11,205$ fibers from 6,342 neurons from hippocampal DG. Data collected from one mouse. **(f)** Amplitude of *c-fos* signals obtained from **e**. ***, $P < 0.001$; n.s., not significant; Kruskal–Wallis analysis of variance followed by Dunn’s post hoc tests. Middle line in box plot, median; box boundary, interquartile range; whiskers, 10-90 percentile; black dots, individual data points. See **Supplementary Table 2** for details of statistical analysis. **(g)** The experimental timeline for *c-fos* signal monomer recording *in vivo*. Dox-containing drinking water was administered on day 10, seizure induction was performed on day 10, Dox-containing drinking water was replaced with regular water on day 12, and the brain was fixed on day 14. **(h)** Left panel, Dox signal relative change from baseline over the 14-day experiment in the control and experimental group. Centerline, mean; shaded area, interquartile range (IQR). Right panel, amplitudes of Dox signals relative change from baseline in control (Ctrl; no Dox, no KA) and experimental (Exp; described in **g**) mice. ***, $P < 0.001$; n.s., not significant; Kruskal–Wallis analysis of variance followed by Dunn’s post hoc tests. Middle line in box plot, median; box boundary, interquartile range; whiskers, 10-90 percentile; black dots, individual data points. See **Supplementary Table 2** for details of statistical analysis. **(i)** Averaged *c-fos* signals relative change from baseline in control (Ctrl; no Dox, no KA) and experimental (Exp; described in **g**) mice. ***, $P < 0.001$; n.s., not significant; Kruskal–Wallis analysis of variance followed by Dunn’s post hoc tests. Middle line in box plot, median; box boundary, interquartile range; whiskers, 10-90 percentile; black dots, individual data points. See **Supplementary Table 2** for details of statistical analysis. For the control group, $n = 717$ neurons from hippocampal DG from one mouse. For the experimental group, $n = 4,514$ neurons from hippocampal DG from one mouse. **(j)** Recorded *c-fos* signal monomer intensity relative change from baseline in DG neurons under KA-induced seizure (the experiment is described in **g**). The time axis was reconstructed for each fiber using Dox ON (onset of Dox signal rise), Dox OFF (onset of Dox signal decay), and fixation times shown in **g** as timestamps. $n = 4,514$ neurons from one mouse. Dashed vertical line, time of seizure induction. **(k)** The experimental timeline for *c-fos* signal monomer recording *in vivo*. Dox-containing drinking water was administered on day 10, seizure induction was performed on day 10, Dox-containing drinking water was replaced with regular water on day 13, and the brain was fixed on day 18. **(l)** Recorded *c-fos* signal monomer intensity relative change from baseline in CA1 neurons under KA-induced seizure (the experiment is described in **k**). The time axis was reconstructed for each fiber using Dox ON (onset of Dox signal rise), Dox OFF (onset of Dox signal decay), and fixation times shown in **k** as timestamps. $n = 14,123$ neurons from one mouse. Dashed vertical line, time of seizure induction.

Fig. R6 Immunohistochemical characterization of cellular and synaptic state markers in mouse brains expressing CytoTape-vivo. (a) Representative confocal images of brain slices from adult mice expressing CytoTape-vivo in the hippocampus in the right cerebral hemisphere following AAV injection. 3-month-old mice were injected with AAV9-CytoTape-vivo-HA at the CA1 region in the hippocampus of the right cerebral hemisphere. When the mice reached 14 days after AAV injection, they were perfused with 4% PFA, and brains were sliced coronally at 40 μm in 1X PBS, and stained with antibodies against one of the cellular and synaptic markers below (see b-j) and against HA tag to label CytoTape-vivo fibers, together with DAPI to label nuclei. Staining intensities of cellular and synaptic markers in the hippocampus were imaged

volumetrically using a 40x objective on a spinning disk confocal microscope, with identical imaging conditions, measured in ImageJ as the averaged fluorescent intensities of the fluorescent secondary antibodies over imaged fields of view ($400\ \mu\text{m} \times 400\ \mu\text{m} \times 40\ \mu\text{m}$ for each fields of view), and compared between the left hemisphere and the right hemisphere. Scale bars, $500\ \mu\text{m}$. **(b-j, top)** Representative confocal images of the CA1 region in the non-injected hemisphere (Ctrl) and CytoTape-vivo-injected hemisphere (CytoTape-vivo) in the brain slices stained with antibodies against each of the cellular and synaptic markers indicated, and DAPI. Scale bars, $50\ \mu\text{m}$. **(b-j, bottom)** Box plots of the quantifications for each of the cellular and synaptic markers between the non-injected hemisphere (Ctrl) and CytoTape-vivo-injected hemisphere (CytoTape-vivo); for each marker in each hemisphere, $n = 10$ fields of view (FOVs) from 3 mice. Middle line in box plot, median; box boundary, interquartile range; whiskers, minimum and maximum; black dots, individual data points. See **Supplementary Table 2** for details of statistical analysis. n.s., not significant; two-sided Wilcoxon rank sum test. **(b)** NeuN (a neuronal marker). **(c)** Cleaved Caspase-3 (an apoptotic marker). **(d)** GFAP (an astrocyte marker). **(e)** Synaptophysin (a synaptic protein marker). **(f)** Iba1 (a microglial marker). **(g)** γH2AX (a DNA damage marker). **(h,i)** Hsp70 and Hsp27 (cell physiological stress markers). **(j)** *c-fos* protein (a cellular activity maker).

Major Concerns

1. The fiber width readout shows little difference between B1 and C4 over time (7d, 14d, 18d). This suggests that removing the N-terminal four residues from MBP in C4 had no positive effect. The authors compare C4 to XRI (Line 229 to 300), which is inappropriate since C4 was derived from B1. To test whether MBP trimming reduces lateral growth, C4 should be directly compared to B1, not XRI, which contains many other differences.

Reply 1: Following the referee's suggestion, we compared the thickness of B1 (1POK(E239Y)-linker2-MBP-HA) and C4 (1POK(E239Y)-linker2-dMBP-HA). As shown in **Fig. R14**, C4 are thinner than B1. Therefore, we incorporated dMBP into CytoTape. Fig. R14 has been included as Extended Data Figure 1c in the revised main text and detailed statistical analysis has been included in the Supplementary Table 2.

Fig. R14 Statistical analysis of protein assembly width for B1 and C4 in cultured neurons at day 7. A total of 31 fibers from B1 and 29 fibers from C4 were analyzed.

2. From the description in Lines 301–305, the only difference between C4 and CytoTape is the removal of a 2-amino-acid linker; both carry the L349K mutation (retained after Line 259–260). Therefore, the final gain in morphology (thinner, bendable fiber) likely comes from linker removal alone. If L349K had minimal effects (as seen from A2), and all later designs retained it, then it is the linker, rather than other mutations, that contributes most to the CytoTape phenotype.

Reply 2: The differences between C4 and CytoTape involve two factors: removal of a 2-amino-acid linker and introduction of the L349K mutation. Among all the designs tested in Fig. 1c, only A2 and CytoTape contain the L349K mutation; all later designs do not include this mutation. To further evaluate the contribution of L349K to CytoTape thickness, we compared CytoTape (1POK(E239Y,L349K)-dMBP-HA) with a design lacking the L349K mutation (C6, 1POK(E239Y)-dMBP-HA). As shown in **Fig. R15**, the C6 formed thicker fibers than CytoTape, indicating that L349K contributes to reducing fiber thickness. This result is also consistent with when comparing XRI and A2-2, which indicates L349K can decrease the thickness of protein fiber. Fig. R15 has been included as Extended Data Figure 1c in the revised main text and detailed statistical analysis has been included in the Supplementary Table 2. The related discussion has been added into the revised Supplementary Information.

Fig. R15 Statistical analysis of protein assembly width for CytoTape and CytoTape without L349K mutation in cultured neurons at day 7. A total of 20 fibers from C6 and 16 fibers from CytoTape were analyzed.

3. Following point #2, the authors should summarize the structures and derivation paths of all designs (XRI, A1–A3, B1, C1–C5, and CytoTape) in a table (e.g., in Figure 1). The narrative is difficult to follow, and some lines (e.g., 259–260) are prone to misinterpretation. This table should clarify which mutation or deletion each design includes and from which parent it was derived. The role of AI is overstated, CPDiffusion did not contribute to the final design, and the

only AI-assisted mutation (L349K via ProtSSN) had limited effect. Therefore, the abstract's claim of 'AI-guided rational design' (Line 38–40) is not justified.

Reply 3: We provided all construct designs (Fig. 1c) in Supplementary Table S2 of the original Supplementary Information. Following the referee's suggestion, we have now moved this table to Extended Data Table 1 in the revised main text. Furthermore, we have also shortened descriptions about AI (CPDiffusion and ProtSSN) in the Abstract, Results, and Discussion sections. To further demonstrate that our design strategy generates thin and flexible protein fibers *in vivo*, we injected CytoTape and XRI into the mouse brain and quantified fiber thickness. As shown in **Fig. R9**, CytoTape forms significantly thinner fibers compared to XRI. This result indicates that our design strategy enables the formation of structurally optimized fibers not only in cell culture but also in the mouse brain. We have added Fig. R9 and Table R2 into the revised main text.

Fig. R9 Analysis of XRI and CytoTape fiber width in the hippocampus of the mouse brain *in vivo*. The brain was fixed 14 days after AAV injection. A total of 358 XRI fibers and 312 CytoTape fibers were analyzed for statistical analysis. ***, $P < 0.001$; Mann–Whitney U test. Middle line in box plot, median; box boundary, interquartile range; whiskers, minimum and maximum; black dots, individual data points.

Table R2 Constructs of structural monomer tested in cultured neurons in this study

Name	Construct (promoters are underlined)	Resulted pattern of protein self-assembly (in the cytosol unless noted otherwise)
A1-1	UbC -ATT1-Liner25-HA-Linker3-MBP	Short fibers
A1-2	UbC -ATT2-Liner25-HA-Linker3-MBP	Short fibers

A1-3	UbC -ATT3-Liner25-HA-Linker3-MBP	Short fibers
A1-4	UbC -ATT4-Liner25-HA-Linker3-MBP	Short fibers
A1-5	UbC -ATT5-Liner25-HA-Linker3-MBP	Short fibers
A2-1	UbC -1POK(E239Y, E149I)-Linker25-HA-Linker3-MBP	Fibers
A2-2	UbC -1POK(E239Y, L349K)-Linker25-HA-Linker3-MBP	Fibers
A3-1	UbC -mhYFP-Linker5-B. subtilis MreB(mrebb5)	Puncta
A3-2	UbC -mhYFP-Linker5-E. coli(mrebc)	Intertwined fibers
A3-3	UbC -mGFP-Linker5-EcMReB	Intertwined fibers
A3-4	UbC -PiiAN-Linker3-HA-Linker3-PiiAC	Puncta
A3-5	UbC -AvECN-Linker3-HA	Puncta
B-1	UbC -1POK(E239Y)-Linker2-MBP-HA	Fibers
C-1	UbC -1POK(E239Y)-Liner25-HA-Linker3-KpMBP	Fibers
C-2	UbC -1POK(E239Y)-Liner25-HA-Linker3-PsMBP	Puncta
C-3	UbC -1POK-Liner25-HA-Linker3-MBP-Linker3-MBP	Puncta
C-4	UbC -1POK(E239Y)-Linker2-dMBP-HA	Fibers
C-5	UbC - mGFP-P2A-1POK(E239Y)-Liner25-HA-Linker3-MBP	Fibers
C-6	UbC -1POK(E239Y)-dMBP-HA	Fibers
CytoTape	UbC -1POK(E239Y, L349K)-dMBP-HA	Fibers

4. The predicted structural model (Fig. 1f, 1g) requires further validation. Please provide the original AlphaFold3 output to examine confidence levels and support the hypothesis regarding L349K's position and its functional relevance.

Reply 4: The XRI and CytoTape monomer structures shown in Fig. 1e in the revised main text were predicted using AF3. The octameric assembly shown in Fig. 1f in the revised main text is based on the crystal structure of 1POK (PDB ID: 1POK). Both 1POK and MBP have been independently resolved by X-ray crystallography.

Following the referee's suggestion, we used AF3 to analyze the PAE, iTM, and pLDDT values of the CytoTape and XRI monomers. As shown in **Fig. R16**, both 1POK and MBP regions exhibit high-confidence predictions, including L349K, which can be attributed to their presence in the AlphaFold training set as experimentally determined crystal structures. In the CytoTape monomer, the interaction region between 1POK and MBP also shows high-confidence predictions. This region corresponds to the C-terminus of 1POK and the N-terminus of MBP, both of which are resolved by X-ray crystallography. By contrast, the corresponding interaction region in the XRI monomer shows low-confidence predictions, due to the presence of flexible elements such as linker25 (25 AAs), the HA epitope tag (7 AAs), and linker3 (3 AAs). These regions form unstructured coils (resolved by X-ray crystallography) and lack defined secondary structure (α -helices or β -sheets), contributing to the reduced prediction confidence. Although the linker region in XRI shows low confidence due to its intrinsic flexibility, the overall

structure exhibits high confidence and provides a reliable model of the monomer. This structural model aids in understanding how the protein monomer can be engineered and serves as a basis for molecular dynamics simulations. Therefore, we have retained the AF3 model in the manuscript. To clarify the confidence levels of the AF3-predicted structures of CytoTape monomer and XRI monomer, we have added Fig. R16 and related discussion to the revised Supplementary Information.

Fig. R16 Confidence analysis of AlphaFold3-predicted models for XRI and CytoTape monomers. The dashed black rectangle highlights the interaction (amino acid structure) between the two domains. Lower panels represent the PAE heatmap of XRI monomer (left panel) and CytoTape monomer (right panel). The linker25, HA epitope tag, and linker3 are

located at positions 391 to 427 in the XRI monomer. Overall iPTM of XRI monomer and CytoTape monomer is 0.55 and 0.7, respectively. All the panels are visualized by the AF3 online system at alphafoldserver.com.

5. In Figure 2b (left), both XRI and CytoTape show a similar elongation rate until day 14, after which CytoTape accelerates. This coincides with increased curvature seen at day 14 (Fig. 2a bottom). If curvature increases elongation speed, fiber growth becomes nonlinear. This could compromise the accuracy of using CytoTape to compare transcriptional activity before and after day 14. The authors should clarify whether the system can reliably record analog signals under non-linear growth conditions.

Reply 5: We thank the referee for the comment. As noted, intracellular protein fiber growth is not strictly linear and can vary depending on cellular context. We have also observed differences in growth rates across cell types and time points (Extended Data Fig. 2d, e in the revised main text). To address this, CytoTape incorporates a HaloTag-based timestamping strategy, as shown in Fig. 3 and Extended Data Fig. 3 in the revised main text. By labeling fibers at defined time points within individual cells, this approach enables reconstruction of temporal information independently of fiber length or curvature, thereby compensating for non-linear growth. To further evaluate temporal resolution under non-linear conditions, we performed dye-switching experiments using HaloTag ligands with defined labeling intervals. As shown in **Fig. R3**, CytoTape achieves minutes-scale temporal resolution. This result demonstrates that CytoTape can reliably record analog transcriptional signals with high temporal precision, despite non-linear fiber growth. We have added Fig. R3 in Fig. 3 in the revised main text.

Fig. R3 CytoTape shows minutes-scale temporal resolution. (a) Schematic of the constructs transfected into HEK cells. (b) Time points of JF₅₈₅, JF₆₃₅, and JF₅₀₃ addition and cell fixation. (c) Confocal images of timestamped fibers labeled with different dye-switching intervals. The dashed lines indicate the transient dye switching. The confocal images were taken using a spinning disk confocal microscope with a 60X objective.

6. In Figure 2d (the V5 experiment), the baseline Egr1 expression (indicated by V5) is dramatically different between XRI and CytoTape groups. It seems that in the XRI condition, neurons are already activated at the time of HA and V5 transfection, triggering IEG expression, while in CytoTape neurons the baseline is nearly silent before KCl addition. If this quantitative difference is real, it implies that either a) XRI increases the firing probability of neurons at baseline, or b) CytoTape suppresses neuronal activity. Which hypothesis is correct?

Furthermore, in Figure 4f, Egr1 expression over time looks normal, with a higher baseline from day 0 to day 5 than in Figure 2. This inconsistency adds confusion. In the XRI paper (doi:10.1038/s41587-022-01586-7), Figure 1 and Figure 4 showed better XRI overall data quality than this manuscript. Therefore, the experiment shown in Figure 2d should not be used to evaluate neuronal health after transfection. If the authors aim to support the hypothesis that XRI harms neurons while CytoTape does not, proper electrophysiological experiments should be conducted. These could include:

- a) neuronal firing ability measured via patch clamp recordings,
- b) synaptic transmission tested through EPSCs/IPSCs, or alternatively calcium imaging to assess activity patterns.

Reply 6: Compared with the *Egr1* baseline shown in Fig. 4f, as well as the baselines in Figs. 1-4 of the XRI paper, the elevated *Egr1* baseline observed in the XRI signal monomer in Fig. 2d is due to the lateral binding of XRI monomers during extended recording. We found that XRI recording for two weeks can no longer record transcriptional signals after neuronal stimulation, resulting a flat signal waveform along fiber. Under the same neuronal stimulation, the transcription activation signals can clearly be detected by CytoTape recording for two weeks (as we demonstrated in this work) and by XRI for a week (as demonstrated in the XRI paper). We speculate that, after a certain period, XRI fiber growth exceeds its thermodynamic equilibrium (ΔG) of protein monomer binding, leading to lateral rather than axial monomer incorporation. As a result, signal monomers bind radially around the fiber instead of along its axis, generating a high *Egr1* baseline across the entire XRI fiber (**Fig. R17**). This phenomenon explains why XRI is unsuitable for long-term recording and informs our design criteria for selecting CytoTape. This observation is consistent with the continuous increase in XRI fiber thickness shown in Fig. 2b. We have added Fig. R17 in the revised Supplementary information.

Fig. R17 Schematic of signal monomer binding kinetics of XRI and CytoTape during weeks-long recording. Green and red rectangles represent structural monomers and signal monomers, respectively.

7. The CREB-Fos experiment is interesting. The unexpected decoupling of CREB and Fos activity in some cells could reflect unhealthy states or unidentified regulatory mechanisms. Could the decoupling reflect inhibition, delayed activation, or artifacts of labeling? Moreover, in coupled cells, can CytoTape resolve whether CREB precedes Fos expression? Given that the CREB signal represents expression level—not phosphorylation—the authors should discuss how they interpret CREB activity and whether it reflects pCREB or total CREB.

Reply 7: We thank the referee for highlighting the significance of our findings. The CREB activity detected by CytoTape is driven by the 6xCRE_CMVmin promoter, which responds to phosphorylated CREB (pCREB) activity (*Biosensors* 2023, 13, 48). We have revised the main text to clarify this point. To validate the observed decoupling between CREB and Fos, we performed immunostaining against pCREB and Fos proteins at 1 hour and 24 hours after forskolin (FSK) stimulation. As shown in **Fig. R18b** and **c**, CREB and Fos signals were not

strongly coupled, supporting the notion that the decoupling may arise from alternative regulatory mechanisms. Upon integrating all detected signals across cells (**Fig. R18a**), we found that, on average, CREB activation precedes Fos expression, consistent with the expected upstream role of CREB in the transcriptional cascade. We have added Fig. R18 in Extended Data Fig. 9 in the revised main text.

Fig. R18 CREB and Fos exhibit complex expression kinetics following FSK stimulation, as revealed by protein immunostaining across HEK cells. (a) Averaged curve of CREB and Fos signals relative change from baseline plotted against recovered time after calcium phosphate transfection. Thick centerline, mean; darker boundary near the centerline, s.e.m. Confocal images of CREB and Fos protein immunostaining in cells fixed at (b) 1 hour and (c) 24 hours after the completion of a 1-hour stimulation with 50 μ M FSK. CREB and Fos intensities were quantified in 30 cells from 2 wells. Each dot represents a single HEK cell. Scale bar, 50 μ m.

8. Building on point #7, what is the temporal resolution of CytoTape in practice? Can it resolve transcriptional events at hourly or sub-hourly resolution, or is it limited to multi-day differences? Time-course validation is needed to determine the minimum resolvable interval between transcription events. If CytoTape can distinguish transcription at sub-daily resolution, it would be a powerful tool for studying gene regulatory networks.

Reply 9: As shown in **Fig. R3** fast dye switching experiments, the temporal resolution of CytoTape is down to minutes-scale. Following the referee's suggestion, we tested whether CytoTape can resolve transcriptional events at sub-daily resolution. As shown in **Fig. R19**, CytoTape successfully recorded two transcriptional events separated by a 12-hour interval, showing two peak-like temporal features at hours-scale. These results suggest that CytoTape can resolve gene regulatory networks on sub-daily timescales at least. We have added Fig. R19 and related discussion in the revised main text and Supplementary Information.

Fig. R19 CytoTape distinguishes CREB levels following two stimulations spaced 12 hours apart. 10 μ M FSK was added to the culture at 50 and 62 hours, with each stimulation lasting 1 hour. CytoTape was analyzed on 60 HEK cells.

Minor Comments

1. In Figures 2a and 2d, Nissl staining fails to clearly show neuronal morphology. Without cell boundaries, it's hard to determine whether CytoTape deforms membranes or alters morphology. Please consider beta-III tubulin or surface markers to visualize cell outlines.

Reply 9: Following the referee's suggestion, we used a cell surface marker, the CellMask plasma membrane stain (Invitrogen), to visualize the cell morphology and whether CytoTape can alter the cell morphology. As shown in **Fig. R20** and **Fig. R21b**, CytoTape does not change the neuron morphology and HEK morphology. We also evaluated whether iPAK4 fibers disrupt cell morphology. As shown in **Fig. R21a**, iPAK4 induces substantial morphological changes. We have added Fig. R20-R21 to the revised Supplementary Information.

Fig. R20 CytoTape does not deform cell membranes or alter cell morphology. CytoTape was expressed in neurons (left panel) and HEK cells (right panel) for 14 days and 3 days, respectively. CytoTape was visualized by HA tag immunostaining (green), and CellMask staining (magenta) was used to label cell morphology.

Fig. R21 iPAK4 deforms cell membranes and alters cell morphology, whereas CytoTape does not. (a) iPAK4 (cyan) and (b) CytoTape (cyan) were expressed for 3 days in HEK cells. Nissl (blue) and CellMask (red) staining were used to label cell morphology. The right panel shows an enlarged view of the dashed rectangle in the left panel. iPAK4 was visualized using the JF₅₈₅ dye, and CytoTape was visualized using HA tag antibody immunostaining. Scale bar, 50 μ m.

2. Typo corrections. Line 162: bioothogonal should be 'bioorthogonal'. Line 42: 'to be readout at scale' should be 'to be read out at scale'

Reply 10: The typos have been fixed.

3. The authors state that they bleached JF dyes before immunostaining (Line 1198) to avoid spectral crosstalk. However, evidence for complete bleaching is not shown. Please provide supplementary data verifying complete dye removal. Additionally, with closely spaced channels

(e.g., 561 and 594 nm), crosstalk is likely. How did the authors address this in multiplexed imaging? What is the maximum number of distinguishable signals the system can currently resolve?

Reply 11: Following the referee's suggestion, we have added the confocal images of JF dyes before and after bleaching (**Fig. R22**) in the revised Supplementary Information. In our study, Alexa Fluor™ 546 was imaged using the 561 nm channel, and Alexa Fluor™ 594 using the 594 nm channel. To assess potential crosstalk, we tested whether Alexa Fluor™ 546 produces signal in the 594 nm channel and whether Alexa Fluor™ 594 produces signal in the 561 nm channel. As shown in **Fig. R23**, Alexa Fluor™ 546 shows only weak signal in the 594 nm channel, and Alexa Fluor™ 594 shows only weak signal in the 561 nm channel. These results indicate that crosstalk between the two channels is minimal. We have added Fig. R22-R23 to the revised Supplementary Information. Currently, CytoTape can distinguish three JF dyes for timestamps and five signals (through epitope tag immunostaining) by standard confocal microscopy. We envision future efforts could greatly enhance the maximum number of distinguishable signals on single fibers via multi-round antibody staining techniques, e.g., those based on antibody striping (*Nat. Commun.* 2024, 15, 9722) or antibody barcoding (*Nat. Protocols* 2021, 16, 3802-3835), to reuse the same optical channels in different rounds to read out distinct signals from tags.

Fig. R22 Confocal images of JF₅₈₅ and JF₆₃₅ on CytoTapes in HEK cells before and after photobleaching. The same field of view was imaged under identical conditions before (left) and after (right) photobleaching of the JF dyes by white LED light for 50 minutes. The image contrast is identical between the two images. Scale bar, 50 μ m.

Fig. R23 No optical crosstalk was observed between the Alexa Fluor 546 dye in the 561 nm channel and the Alexa Fluor 594 dye in the 594 nm channel under the imaging conditions used in this work. Two fields of view (left and right) were imaged under confocal microscopy. The imaging conditions and image contrast for each channel are identical between the two fields of view.

4. After JF dye bleaching and antibody staining, how do the authors re-identify the same fibers for imaging? Did they use stage-coordinates, some markers, or software-based registration? Please describe this process in the Methods.

Reply 12: To re-identify the same fibers after JF dye photobleaching, we used stage coordinates recorded during the initial timestamp imaging to return to the same locations. Additionally, we acquired brightfield images of the same cell to capture its specific morphological context. After antibody immunostaining, we relocated the previously imaged fibers by combining stage-coordinate guidance with brightfield-based visual matching of cell morphology and the spatial arrangement of nearby cells. We have now added these procedure details in the Methods section under “Photobleaching of Janelia Fluor dyes” in the revised main text.

Referee #3 (Remarks to the Author):

I co-reviewed this manuscript with one of the reviewers who provided the listed reports.

Referee #4 (Remarks to the Author):

Zheng and coworkers report on a new method for "recording" multiple transcriptional events at a single-cell level over weeks. I am not an expert on the cell biology aspects that are central to the work, but as a non-expert, this sounds like a remarkable development. My expertise is in computational protein design, and the first part of the work relates to some protein engineering efforts on which I comment below.

Reply: We thank the referee for highlighting the significance of our work and for providing valuable comments on the protein engineering efforts, which is the core aspect of this work that enabled all downstream applications of the engineered proteins we presented in later parts of this paper.

As the rationale for the protein engineering effort, the authors write that a long and rigid assembly would either deform the cell or break up. What lengths did they reach, and how do those relate to the typical length of the cells they tested in this paper?

Reply 1: We observed that CytoTape fibers reached lengths of ~20-30 μm after 3 days of continuous expression in HEK cells (**Fig. R21b**). This exceeds the diameter of HEK cells, which is ~15-20 μm , indicating that the assemblies grew beyond the spatial confines of the cell body. Despite this, CytoTape fibers remained thin and flexible, exhibiting a thread-like morphology that curled within the cytoplasm without causing observable deformation of the plasma membrane or cellular architecture (**Fig. R21b**, right panel). By contrast, iPAK4—which forms rigid crystalline fibers—distorted cell morphology (**Fig. R21a**), reinforcing our hypothesis that long, rigid assemblies are poorly tolerated in live cells (**Fig. R21a**, right panel). These observations validate our engineering rationale: flexible assemblies, such as those formed by CytoTape, can accommodate intracellular spatial constraints and achieve elongation over multiple days or weeks without perturbing cell structure. This behavior was consistent across a range of cell sizes, with CytoTape preserving cell morphology due to its flexibility, whereas iPAK4 induced shape distortion as a result of its rigidity. We have added Fig. R21 and related discussion in the revised Supplementary Information.

Fig. R21 iPAK4 deforms cell membranes and alters cell morphology, whereas CytoTape does not. (a) iPAK4 (cyan) and (b) CytoTape (cyan) were expressed for 3 days. Nissl (blue) and CellMask (red) staining were used to label cell morphology. The right panel shows an enlarged view of the dashed rectangle in the left panel. iPAK4 was visualized using the JF₅₈₅ dye, and CytoTape was visualized using HA tag antibody immunostaining. Scale bar, 50 μm .

The first design attempt using CPDiffusion failed to produce long fibres, and they abandoned it. The authors nevertheless state that the results show the ability of the method to design fibres and suggest that further work would fix it. I think that a balanced, facts-based way of reporting the results would state that the effort failed to produce the desired results and that future work to improve CPDiffusion may be necessary.

Reply 2: We agree and have now revised the corresponding texts in Results following the referee's suggestion. The revised sentences read:

“In our initial attempt to apply CPDiffusion to the design of protein fibers, five variants (A1-1 to A1-5) were generated that incorporated 80 to 120 amino acid substitutions (70–80% sequence identity to the original 1POK(E293Y) scaffold). These designs formed fiber-like assemblies in cultured neurons after 7 days, with thickness comparable to that of XRI (Mann–Whitney U tests with Bonferroni correction, $P = 0.855, 0.585, 1.000, 0.952, 0.585$ for A1-1 to A1-5, respectively; significance threshold = 0.01; Fig. 1c, Extended Data Fig. 1). However, the majority of observed structures exhibited low aspect ratios (< 3) and limited spatial continuity (i.e., they did not form long and continuous fibers), which rendered them unsuitable for our protein tape recording application.”

The end result of the computational design work is a single-point mutation Leu349Lys that induces higher solubility of the lateral surface of the fibre where growth was not desired. Given the very modest contribution of computational design to the work, I recommend toning down the remarks on AI-guided protein design. Solubility design is an extremely mature field, and introducing a single-point mutation -- though it had an important effect here -- is hardly innovative. The paper feels rather long, and I recommend focusing on the innovative aspects in the main results. Because there was no technical innovation in protein design, I would briefly explain the rationale for the solubility mutation without describing the method and remove the CPDiffusion section entirely -- it didn't contribute to the work. I would also remove the reference to AI-based design from the abstract (and other sections) because the effort here relied much more on run-of-the-mill protein engineering methods.

Reply 3: We thank the referee for acknowledging the important roles of the AI-identified single-site mutation and rational design in our protein engineering efforts. Following the referee's suggestion, we have moved methods for CPDiffusion and ProtSSN to the revised Supplementary Information and shortened its description in the Abstract, Results, and Discussion sections. We have also toned down the remarks on AI-guided protein design. The revised sentences read:

Abstract: “Gene expression is constantly regulated by gene regulatory networks that consist of multiple regulatory components to mediate cellular functions. An ideal tool for analyzing gene regulation processes would provide simultaneous measurements of the dynamics of many components in the gene regulatory network, but existing methodologies fall short of simultaneously tracking the dynamics of components over long periods of time. Here, we present CytoTape—a genetically encoded, modular, continuous recorder for multiplexed and scalable *in situ* molecular recording of gene regulation dynamics over up to three weeks, with single-cell resolution and down to minutes-scale temporal precision. CytoTape consists of a flexible, thread-like, elongating intracellular protein self-assembly engineered via computationally assisted rational design, readable by standard microscopy at scale. We demonstrated the utility of CytoTape in mammalian embryonic kidney cells, cancer cells, glial cells, and neurons, achieving simultaneous recording of five cell plasticity-associated transcription factor activities and immediate early gene (IEG) expression levels within single cells in a spatiotemporally scalable manner. CytoTape revealed divergent transcriptional trajectories of genetically identical cells correlate to prior signaling histories, enabling investigations of how transcriptional memory and intrinsic signaling states shape transcriptional logics. CytoTape also captured diverse, multi-peak waveforms of IEG activities following single

stimuli and complex temporal couplings between distinct IEGs in single neurons. We further optimized CytoTape into CytoTape-vivo for spatiotemporally resolved and scalable single-cell recording in the living brain, and demonstrated its utility by simultaneous single-cell recording of doxycycline-dependent and IEG promoter-dependent gene expression histories across up to 8,639 neurons in the hippocampus and cortex of single mice. Together, the CytoTape toolkit opens new avenues for investigating how gene regulatory networks function and govern fundamental physiological processes in cell cultures and *in vivo*.”

Results: “We deployed *ProtSSN* to score and rank single-site saturation mutations of the 1POK(E293Y) scaffold and selected the top 2 mutations, L349K and E149I, for further experimental evaluation. We found that the L349K mutant (design A2-2) resulted in thinner protein fibers compared to the XRI fibers at 7 days of expression (Fig. 1c) (MWU test, $P = 0.020$). Lateral growth for this design was still observed between day 7 and day 14, but not between day 14 and day 18, indicating an improved suppression of long-term lateral growth compared to the XRI design. In contrast, the E149I mutant (design A2-1) did not lead to noticeable changes in fiber thickness compared to the XRI design (Fig. 1c) (MWU test, $P = 0.751$). Based on these observations, we retained the L349K mutation for subsequent design iterations.”

Discussion: We have removed the discussion on future developments in AI-guided protein ticker tape design.

If I understand correctly, the AlphaFold3 modelling shown in Figure 1 is only illustrative. Yet, it's not clear to me that the AF3 reliability scores for the interaction between the fused domains are high enough for a model to be presented. The authors could check the PAE, iPTM and pLDDT scores at the interface to verify this. If the scores are not high, I think that the AF3 model should not be presented, and the authors should then only use their schematic representation.

Reply 4: The XRI and CytoTape monomer structures shown in Fig. 1b, e, and h were predicted using AF3. The octameric assembly shown in Fig. 1f is based on the crystal structure of 1POK (PDB ID: 1POK). Both 1POK and MBP have been independently resolved by X-ray crystallography.

Following the referee's suggestion, we used AF3 to analyze the PAE, iPTM, and pLDDT values of the CytoTape and XRI monomers. As shown in **Fig. R16**, both 1POK(E293Y), MBP, and dMBP regions exhibit high-confidence predictions, which may be attributed to the presence of their crystal structures (and/or those from their related variants) in the AlphaFold training dataset. In the CytoTape monomer, the interaction region between 1POK and MBP also shows high-confidence predictions. This region corresponds to the C-terminus of 1POK and the N-terminus of MBP, both of which are resolved by X-ray crystallography. By contrast, the corresponding interaction region in the XRI monomer shows low-confidence predictions, due to the presence of flexible elements such as linker25 (25 AAs), the HA epitope tag (7 AAs), and linker3 (3 AAs). These regions form unstructured coils (resolved by X-ray crystallography) and lack defined secondary structure (α -helices or β -sheets), contributing to the reduced prediction confidence. Although the linker region in XRI shows low confidence due to its intrinsic flexibility, the overall structure exhibits high confidence and provides a reliable model of the monomer. This structural model aids in understanding how the protein monomer can be engineered and serves as a basis for molecular dynamics simulations. Therefore, we have retained the AF3

model in the manuscript. To clarify the confidence levels of the AF3-predicted structures of CytoTape monomer and XRI monomer, we have added Fig. R16 and related discussion and details to the revised Supplementary Information.

Fig. R16 Confidence analysis of AlphaFold3-predicted models for XRI and CytoTape monomers. The dashed black rectangle highlights the interaction (amino acid structure) between the two domains. Lower panels represent the PAE heatmap of XRI monomer (left panel) and CytoTape monomer (right panel). The linker25, HA epitope tag, and linker3 are located at positions 391 to 427 in the XRI monomer. Overall iPTM of XRI monomer and CytoTape monomer is 0.55 and 0.7, respectively. All the panels are visualized by the AF3 online system at alphafoldserver.com.

As with other sections, the Discussion is very long. The last paragraph is mostly speculation on how future developments in protein design could improve the constructs, but I don't think that this goes beyond trivia. I think a much more focused discussion would be best.

Reply 5: We agree the Discussion is very long and we have removed the texts in the Discussion on future developments in AI-guided protein ticker tape design. We respectfully defer to the editor's recommendation regarding whether the current Discussion is still over the word limit and whether removing additional texts there are necessary.

The methods section on molecular dynamics simulations contained fewer technical details than I'm used to seeing in such sections. It would be good to verify with an expert that this section is complete.

Reply 6: Following the referee's suggestion, we have revised the molecular dynamics (MD) simulation section of Methods and expanded it to include additional technical details, including the force field used, system preparation steps, equilibration protocols, simulation length, temperature and pressure controls, and analysis methods. The revised MD method reads:

"The structure of the protein monomer for simulations was predicted by AlphaFold3 (*Nature* 2024, 630, 493-500) (Fig. 1h and i). The simulations were replicated three times. A cubic box was filled with one protein monomer. Water molecules were randomly inserted into the box to reach the protein that is fully covered by water molecules in the simulation. Chlorine counter ions were added to keep the system neutral in charge. The CHARMM27 force field was used for the complex and the CHARMM-modified TIP3P model was chosen for water. The simulations were carried out at 310 K (37 °C). After the 5000-step energy-minimization procedure, the systems were heated and equilibrated for 100 ps in the NVT ensemble and 500 ps in the NPT ensemble. 500 ns production simulations were conducted with a trajectory saving frequency of 10 ps. The final 300 ns (30,000 frames) were extracted for subsequent analysis. The integration step was set to 2 fs, and only the covalent bonds with hydrogen atoms were constrained by the LINCS algorithm. Lennard-Jones interactions were truncated at 12 Å with a force-switching function from 10 to 12 Å. The electrostatic interactions were calculated using the particle mesh Ewald method with a cutoff of 12 Å on a 1 Å grid with a fourth-order spline. The temperature and pressure of the system are controlled by the velocity rescaling thermostat and the Parrinello-Rahman algorithm, respectively. All molecular dynamics simulations were performed using GROMACS 2021.1 packages.

The free energy landscape (FEL) (*Nat. Commun.* 2025, 16, 6211; *Chem. Sci.* 2024, 15, 5612-5626; *Nucleic Acids Res.* 2022, 50, 7529–7544) was constructed using the trajectory data from the last 300 ns of the production simulations. Two reaction coordinates—root-mean-square deviation (RMSD) from the initial structure and radius of gyration (R_g)—were used to represent the conformational space of the protein. Trajectory frames were extracted at 10 ps intervals and aligned to the reference structure to remove overall translational and rotational motions. The 2D histogram of RMSD and R_g was computed using bins of 0.05 nm × 0.05 nm, and the corresponding free energy G was calculated using the Boltzmann inversion: $G(\text{RMSD}, R_g) = -k_B T / \ln P(\text{RMSD}, R_g)$, where k_B is the Boltzmann constant, T is the absolute temperature (310 K), and $P(\text{RMSD}, R_g)$ is the normalized probability distribution of the system in the 2D conformational space. The analysis was performed using GROMACS tools (gmX rms, gmX gyrate, and gmX sham), and the FEL was visualized as a contour map using Origin2019b.

To evaluate the relative motion between two structural domains within the same protein, the center-of-mass (COM) distance between the domains was calculated over the course of the molecular dynamics simulation. The simulation trajectory was preprocessed using gmx trjconv to remove periodic boundary conditions and align the protein to a common reference frame. The interdomain distance at each time point was determined as the Euclidean distance between the COM coordinates. Analysis was performed on the final 300 ns of the production simulation, sampled every 10 ps, and the resulting distance profile was visualized using Origin2019b.”

GoogleScholar shows that CPDiffusion and ProtSSN have been published previously. If so, extended figure 1 on which describes the methods, isn't necessary.

Reply 7: Following the referee’s suggestion, we have removed Extended Data Figure 1 in the revised main text.

Authors' Response:

We thank the editor and referees for the constructive comments and for the opportunity to submit our revised paper in response to the comments.

Per the editor's request and *Nature's* guidance to conform to the Article format, including shortening the maintext (up to 4,300 words), reducing the number of Figures (up to 5 figures) and Extended Data Figures (up to 10 A4 pages), and reorganizing all materials into the required structure, we substantially revised the manuscript. Specifically, we shortened the Abstract to ~200 words, the Maintext to ~4,300 words, and reduced the number of References to ~50. For figures in the manuscript, we merged the original Figs. 1 and 2 into a new Fig. 1, moved the original Fig. 4 to the Supplementary Data Fig. 4, the original Extended Data Figs. 1-8, 11 to the Supplementary Information. Full details of the design rationale, results, discussion, limitations, and future directions for the CytoTape system since the last revision are now provided in the Supplementary Information.

The responses below are structured as follows. We begin by providing a general overview of how the revised manuscript has been further strengthened relative to the previous submission. We then address specific comments from the editor and the referees (highlighted in blue below) in full. In the revised manuscript, all the modifications are highlighted in red.

1. Validation of biological insights identified by CytoTape on the decoupling between pCREB and Fos activities: Six cellular markers (Cleaved Caspase-3, GRP78/BiP, Ki-67, TOMM20, Ethidium homodimer-1, and Calcein-AM), four activity-dependent transcriptional markers (Egr1, pElk-1, pCREB, and Fos), and one signaling marker (pERK1/2), collectively demonstrated that CytoTape does not alter the viability, proliferation, physiological state, transcriptional activities, or signaling of HEK cells. In addition, control experiments without stimulation demonstrate that the CytoTape fibers do not alter pCREB or Fos activities. Both fluorescence reporter and immunostaining assays showed consistent decoupling between pCREB and Fos activities, confirming that this decoupling is not caused by CytoTape expression. Thus, the observed pCREB-Fos decoupling by CytoTape is not an artifact of the *in vitro* system.

2. CytoTape-based testing of the hypothesis of pCREB–Fos activity decoupling: We investigated how cAMP/PKA-driven pCREB activation is transduced or fails to be transduced into Fos induction at single-cell level. We hypothesized that ERK signaling may contribute to the pCREB–Fos decoupling as a parallel pathway to pCREB signaling, since the *c-fos* promoter contains both CRE element regulated by pCREB signaling and the SRE element regulated by ERK signaling. We observed that inhibition of ERK signaling enhanced pCREB–Fos decoupling, while activation of ERK signaling during pCREB activation resulted in markedly strong Fos induction but did not decrease the fraction of cells that show pCREB–Fos decoupling. These results suggest that ERK signaling is necessary but not sufficient for robust Fos induction and that other mechanism(s) beyond ERK signaling may contribute to pCREB–Fos decoupling. These results suggest a signal integration mechanism in IEG regulation, further demonstrating the utility of CytoTape in dissecting gene regulation dynamics.

3. Functional validation of neurons and neural circuitry *in vitro* and *in vivo*:

Immunohistochemical characterization of molecular and cellular physiological markers, electrophysiological analysis of neuronal physiology and synaptic transmission, calcium

imaging-based analysis of spontaneous neuronal network dynamics, and animal behavioral analyses, including open field test, novel object recognition, and contextual fear conditioning, demonstrated that the CytoTape system does not affect neuronal or circuit function *in vitro* and *in vivo*.

4. Demonstration of sub-daily temporal resolution at weeks-long temporal scale: New data demonstrate that CytoTape can achieve sub-daily resolution while maintaining weeks-long recording capability in neurons.

Editor's comments and authors' responses, Page 4-8
Referee #1's comments and authors' responses, Pages 9-11
Referee #2's comments and authors' responses, Pages 12-32
Referee #3's comments and authors' responses, Pages 33
Referee #4's comments and authors' responses, Pages 34-35

Editor's comments:

Among the limitations, it will be important to address the concern regarding whether the decoupling represents an artificial finding *in vitro*.

Reply: To address this concern, we first examined the cellular and physiological states of HEK cells expressing CytoTape. As shown in **Fig. R1a**, the majority of HEK cells expressed CytoTape fibers. We performed immunohistochemical characterization and fluorescence-based viability assays and found that CytoTape expression did not alter cellular or physiological state markers (**Fig. R1b-g**), including Cleaved Caspase-3 (apoptotic marker), GRP78/BiP (endoplasmic reticulum stress marker), Ki-67 (cell proliferation marker), TOMM20 (mitochondrial integrity marker), Ethidium Homodimer-1 (membrane permeability and DNA damage marker), and Calcein-AM (live-cell viability indicator). We then examined the transcriptional activities and signaling of HEK cells expressing CytoTape, using markers including Egr1 (early growth response 1; IEG marker), pElk-1 (ERK-dependent transcription factor activation marker), pCREB (cAMP/PKA-dependent transcription factor activation marker), Fos (IEG marker), and pERK1/2 (ERK/MAPK pathway activation marker). **Fig. R1h-i** shows that CytoTape expression did not alter the transcriptional or signaling activities of HEK cells.

We integrated all CytoTape-recorded signals across cells (**Fig. R2a**) and found that, on average, pCREB activation precedes Fos expression, consistent with the expected upstream role of pCREB in the transcriptional cascade. To validate the observed decoupling between pCREB and Fos, we performed orthogonal single-cell assays, including immunostaining of pCREB and Fos, as well as fluorescence reporter assays using pCREB-EGFP and Fos-mRuby3 constructs. Immunostaining and fluorescence imaging were conducted at 1 hour and 24 hours after forskolin stimulation. As shown in **Fig. R2b-e**, a substantial portion of cells showed pCREB–Fos decoupling, in agreement with the results from CytoTape recording. In addition, we performed two independent control experiments: CytoTape recording without forskolin stimulation, and immunostaining for pCREB and Fos in HEK cells with and without CytoTape expression. These results confirmed that CytoTape expressions do not alter pCREB or Fos activities (**Fig. R2f, Fig. R1j,k**). Together, the decoupling of pCREB and Fos observed by CytoTape is not an artifact of the *in vitro* system. We have added Fig. R1-R2 and associated discussion in the revised maintext.

Fig. R1 Immunohistochemical and fluorescence-based characterization of cellular and physiological state markers, as well as transcriptional activity and signaling markers, in HEK cells expressing CytoTape. (a) Representative confocal image of HEK cells expressing CytoTape. Majority of HEK cells express CytoTape fibers. HEK cells with or without CytoTape expression were stained with (b-e) antibodies against specific cellular or physiological markers, (f,g) fluorescent chemical dyes, (h-k) antibodies against specific transcriptional activity, and (l) antibodies against signaling markers. CytoTape fibers were labeled with JF₅₈₅ and nuclei were stained with DAPI. Fluorescence intensities were imaged volumetrically using a 10x objective on a spinning disk confocal microscope under identical imaging conditions. (b,e,h-l) Mean fluorescence intensities were quantified in ImageJ as the averaged signal of the fluorescent secondary antibodies over imaged fields of view. Scale bars, 50 μ m. (b-l, top) Representative confocal images of HEK cells transfected with pUC19 (Ctrl) or expressing CytoTape, stained with the indicated antibodies or imaged with the indicated fluorescent dyes. Scale bars, 50 μ m. (b-l, bottom) Scatter plots of the quantifications for each marker between control and CytoTape-

expressing groups (n = 10 fields of view from 2 cultures). Middle line, median; whiskers, interquartile range; black dots, individual data points. n.s., not significant; Mann–Whitney U test. (b) Cleaved Caspase-3 (apoptotic marker). (c) GRP78/BiP (endoplasmic reticulum stress marker). (d) Ki-67 (cell proliferation marker). (e) TOMM20 (mitochondrial integrity marker). (f) Ethidium homodimer-1 (membrane permeability and DNA damage marker). (g) Calcein-AM (live cell marker). (h) Egr1 (Egr1 expression marker). (i) pElk-1 (Elk activity marker). (j) pCREB (CREB activity marker). (k) Fos (Fos expression marker). (l) pERK1/2 (ERK activity marker).

Fig. R2 pCREB and Fos exhibit complex, temporally decoupled dynamics following pCREB activation, as validated by protein immunostaining and fluorescence protein reporter assays across HEK cells. (a) Averaged curve of pCREB and Fos signals relative change from baseline plotted against recovered time after *TransIT-X2* transfection. Thick centerline, mean; darker boundary, s.e.m. Confocal images of pCREB and Fos protein immunostaining in cells fixed at (b) 1 hour and (c) 24 hours after the completion of a 1-hour stimulation with 50 μ M FSK. Confocal images of pCREB and Fos protein reported by EGFP and mRuby3 in cells fixed at (d) 1 hour and (e) 24 hours after the completion of a 1-hour stimulation with 50 μ M FSK. pCREB and Fos intensities in (b-e) were quantified in 30 cells from 2 cultures. Each dot represents a single HEK cell. Scale bar, 50 μ m. (f) Profiles of pCREB and Fos signals intensities along CytoTape in the control group (no stimulation). Thick centerline, mean; darker boundary, s.e.m; thin lines, single traces. pCREB and Fos intensities were quantified in 8 cells from 2 cultures.

Format changes:

STATISTICS: We have confirmed that detailed information on statistical tests, sample sizes, replicates, and error bar definitions has been included in the figure legends and Methods, as well as in the Supplementary Information and Supplementary Table 2.

REPRODUCIBILITY: Reporting summary and the Code and software Checklist, have been updated to reflect the revisions and are included with the resubmission.

LENGTH: Maintext has been revised to ~4300 words and limited to 5 main figures.

TITLE: Title contains 74 characters and does not include any punctuation.

SUMMARY PARAGRAPH: Summary Paragraph has been condensed to 205 words, includes references, and adheres to the organizational structure outlined in the editorial instructions.

MAIN TEXT: Maintext has been streamlined for clarity and conciseness. The introduction now comprises ~500 words. Subheadings within the maintext have been limited to ~40 characters.

REFERENCES: References have been reduced to 52.

FIGURE LEGENDS: All figure legends have been revised to ensure completeness and consistency with Nature's standards.

METHODS: Methods section has been fully revised to provide step-by-step details sufficient for replication. All experimental parameters, reagents, and analysis procedures have been explicitly described. References cited in the Methods continue the numbering from the maintext.

ETHICS STATEMENT: All animal experiments were approved by the Institutional Animal Care and Use Committee of University of Michigan, Icahn School of Medicine at Mount Sinai, and Massachusetts Institute of Technology, and were performed in accordance with institutional and national guidelines and regulations. No human subjects, human embryos, or human stem cells were involved in this study.

MAIN TEXT STATEMENTS: Acknowledgements section lists all sources of funding and institutional support. Author Contributions statement specifies the experimental and analytical roles of each author. Competing Interests statement has been added to disclose that the authors have applied for patent(s) related to this work.

DATA AND CODE AVAILABILITY STATEMENTS: Data Availability Statement has been included in the maintext specifying that all data supporting the findings of this study are available within the paper and supplementary files. Code Availability Statement has also been added.

DISPLAY ITEMS: All figures have been evaluated and reduced to present only data central to the study's conclusions. Maintext includes five figures, each provided in an editable format (.pdf) and assembled into rectangular layouts consistent with Nature's figure specifications. Figure panel organization and scaling were optimized to ensure clarity at reduced print sizes.

FIGURE FORMATTING: All figure labels use a uniform sans-serif font (≥ 5 pt at final print size) and standardized lower-case lettering. Axes and units follow SI conventions with appropriate spacing. Figures were prepared in accordance with Nature's image-presentation and data-integrity guidelines.

EXTENDED DATA: A total of eleven Extended Data figures have been prepared, each submitted as individual .pdf files of print-quality resolution. Their legends are listed sequentially after the maintext and all are cited appropriately within the maintext.

SUPPLEMENTARY INFORMATION: We have prepared and formatted the Supplementary Information according to the provided guidelines.

SOURCE DATA (GRAPHS): For transparency, source data underlying each figure related with animal model have been provided as individual Excel files. All Source Data files are referenced in the respective figure legends.

RAW DATA (GELS): The electrophoretic separation experiment is not performed in this study.

THIRD PARTY RIGHTS: The scientific illustrations shown in Fig. 1a and Fig. 5a,b were created with BioRender. We have obtained a license from BioRender.

ORCID: The corresponding author has linked the ORCID to his account on Nature's MTS.

Referees' comments:

Referee #1 (Remarks to the Author):

This substantially revised manuscript is an impressive work that now explicitly demonstrates the advantages of the technology that were hinted at in the original submission. Whether an improvement in an existing approach is appropriate for this particular journal is an editorial decision. Either way, I do think that this technology has clear value to the field.

Reply 1: We thank the referee for the positive evaluation and for recognizing that the revised manuscript explicitly demonstrates the advantages of our technology.

I applaud the new experiments. I may be biased since it was performed at my suggestion, but I think the new results reported in Figure 5f-l are the best demonstration of the power of the approach. The recordings sharply show temporally precise recordings over an impressive span of time. I would only ask that the authors provide some example images to accompany these experiments in a supplemental figure given that I strongly think this will be a critical result that is discussed in labs around the world when evaluating the use of the approach, and while interpreting results if they adopt the technology. I also appreciate the more direct demonstration of advantages of standard timelapse FP imaging, highlighting the utility of a record the moves with the cell, as well as the *in vivo* data.

Reply 2: We thank the referee for the experimental suggestions and the positive evaluation on the strength of the new experiments in Fig. 5f-i. Following the referee's suggestion, we have included example images corresponding to these experiments in the maintext (**Fig. R3**). We also appreciate the referee's agreement of the advantages over time-lapse fluorescent protein imaging and the inclusion of *in vivo* data.

Fig. R3 CytoTape enables weeks-long recording of *c-fos*-promoter driven gene expression in cultured neurons. (a,d) Schematic of the constructs transfected into cultured neurons, (b,e) the experimental timeline (top panel), and the expected dye distributions along the CytoTape (bottom panel). (c,f) Images of neurons expressing the constructs shown in a,d. (b) 15 μM FSK was added to the culture on day 3 and day 6, with an incubation time of 1 h. (e) 15 μM FSK was added to the culture on day 3 and day 15, with an incubation time of 1 h for the first stimulation and 1.5 h for the second stimulation. (c,f) First row, the image in the timestamp channels; Second row, the image in the *c-fos*-promoter signal channel.

My last remaining concern is semantic, but it is something that I think is very important for the integrity of the literature. This technology is an improved version of a previously developed approach called XRI. Yes, the fibers are more controlled and that yields much better recording

and potential for multiplexing and time-stamping. However, the technology is presented by the authors as though it is a fully new approach and given a new name, distinct from XRI. I takes an unnecessary amount of effort to determine how related the two approaches are since one was developed from the other, and that is bad for the health of the field of technology development and a user's ability to clearly evaluate which tools they should be using. I don't even mind that there is a new name. I think CytoTape is, much like the technology, better than the original. However, there is no reason to disguise the relationship. I would like to see one statement in the abstract and one in the introduction that clearly states something like "We improved upon a previous protein recording technology called XRI to create more temporally precise and multiplexable recording approach, which we now call CytoTape."

Reply 3: We agree with the referee on the importance of clearly stating the technological lineage in the paper for the integrity of the literature and the health of the technology development field. Our rationale was to use a name that accurately describes the current technology while also encouraging future technology development in the field, as described below.

The name *eXpression Recording Islands* (XRI) originated from the fact that XRI structures are "island-like" in live cells (shorter and less flexible than CytoTape fibers), inspired by our earlier work on *Signaling Reporter Islands* ("SiRI"; *Cell* 2020, 183, 1682-1698) that creates puncta-like structures in live cells for multiplexed live imaging of fluorescent biosensors/reporters. However, the term "islands" no longer precisely describes the morphology of the new CytoTape fibers, which are long, thin, and flexible, best described as a "tape" or a "thread". Because "tape" more aligns with our goal of building a continuous recording system, as in the terms "tape recorders" and "ticker tapes", we decided to use "tape" for the name of the technology. Moreover, because XRI cannot function properly in dividing cells, while CytoTape can (as confirmed in HEK, HeLa, and glial cells), we therefore believe using the word "Cyto" could further clarify the fact that CytoTape can function across diverse cell types. We understand the potential risk of such a new name, as clearly pointed out by the referee, that it might make the technology development history harder to track (we hope this can be largely mitigated by the revised texts in the new Abstract and Introduction, copied below, following the referee's suggestion). Nevertheless, we also recognize the risk of confusion by still naming the thread/tape-like structure as "islands", since it does not precisely describe the distinct morphological features of the new structure. Furthermore, we envision the thread/tape-like structures represent a promising (and perhaps critical) future direction in this field, due to their many advantages that break through the limitations of short, thick, or rigid structures as we experimentally observed in this work. Therefore, we also hope the name CytoTape could encourage future technology development efforts in the field towards thread/tape-like recorders. Essentially, the choice between "XRI2" and "CytoTape" for the name represents whether one would like to emphasize past efforts or to encourage future efforts. Because protein tape recording is a new field with, we believe, large future potential to achieve multiplexed recording across large spatial and temporal scales for new kinds of biology and medicine research, we think choosing the latter name would best benefit the health of this field of technology development and encourage future efforts.

We have added a clear description of the relationship between XRI and CytoTape in the Abstract, as well as the technical gap between previous protein ticker tape methods (XRI and

iPAK4) in the Introduction of the revised maintext. We respectfully note that although we were unable to include extensive details in the maintext due to the word limit required by *Nature's* formatting guidelines, the detailed technological lineage between CytoTape and XRI is discussed in the Supplementary Information.

Abstract: “Gene expression is dynamically regulated by gene regulatory networks comprising multiple regulatory components to mediate cellular functions. An ideal tool for analyzing these processes would track multiple-component dynamics with both spatiotemporal resolution and scalability within the same cells, a capability not yet achieved. Here, we present CytoTape, a genetically encoded, modular protein tape recorder for multiplexed and spatiotemporally scalable recording of gene regulation dynamics for up to three weeks, physiologically compatible, with single-cell, minutes-scale resolution. **CytoTape employs a flexible, thread-like, elongating intracellular protein self-assembly engineered via computationally assisted rational design, built on earlier XRI technology.** We demonstrated its utility across multiple mammalian cell types, achieving simultaneous continuous recording of five transcription factor activities and gene transcriptional activities. CytoTape reveals that divergent transcriptional trajectories correlate with transcriptional history and signal integration, and that distinct immediate early genes (IEGs) exhibit complex temporal correlations within single cells. We further extended CytoTape into CytoTape-vivo for scalable, spatiotemporally resolved single-cell recording in the living brain, enabling simultaneous weeks-long recording of doxycycline- and IEG promoter-dependent gene expression histories across up to 14,123 neurons spanning multiple brain regions per mouse. Together, the CytoTape toolkit establishes a versatile platform for scalable and multiplexed analysis of cell physiological processes *in vitro* and *in vivo*.”

Introduction: “... Thus, a scalable technology enabling long-term, multiplexed measurements of cellular activities with spatiotemporal resolution is critically needed. The recently proposed “protein tape recorder” concept (XRI and iPAK4) allows recording of cellular activities along intracellular linear protein self-assemblies for post-fixation scalable readout. **However, these protein tape recorders do not support simultaneous recording of multiple cellular signals or time-resolved recording over weeks, because long-term growth of linear protein assemblies is constrained by cell size.**

Here, we introduce CytoTape, an intracellular protein tape recorder for scalable and multiplexed continuous recording with minute-scale temporal precision and weeks-long duration, built on the earlier XRI system. CytoTape employs flexible, thread-like, physiologically compatible protein self-assembly that can grow longer than the cell size with multiple molecular tags, each encoding a distinct cellular signal, to achieve long-term, multiplexed recording. ...”

Referee #2 (Remarks to the Author):

The revised manuscript and the authors' responses are carefully reviewed. Unfortunately, this reviewer is not convinced that the revised manuscript represents a significant advance in either technical or scientific dimensions. Although the authors demonstrate that CytoTape outperforms XRI, its core concept remains essentially unchanged; CytoTape appears to be an improved version of the XRI system rather than a new technology.

Reply 1: We respectfully note that CytoTape is not merely an engineered structural monomer but rather a comprehensive and powerful system that represents substantial advances in multiple technical and scientific aspects, including *in vivo* recording, mapping and multiplexing at large spatiotemporal scales (Fig. 5), stabilized protein monomer structure and dynamics (Fig. 1, Extended Data Fig. 1), flexible fiber morphology (Fig. 1), timestamp strategy and time axis recovery (cell culture and *in vivo*) (Fig. 2 and 5), seven transcriptional recorders (Extended Data Fig. 4), minutes-scale temporal resolution and multi-weeks-long temporal scalability (Fig. 2), single-cell spatial resolution and multi-brain-region scalability (Fig. 5), recording performance across diverse dividing and non-dividing cell types (Fig. 3), 8-color, 5-signal multiplexing capability per fiber (Fig. 3), and biological findings on transcriptional regulation dynamics (Fig. 4). Collectively, these multi-level innovations and new capabilities enable the CytoTape system to surpass the XRI system in scope, precision, and versatility, and thus the CytoTape system represents a fundamental and substantial advance over the XRI system.

Here, we outline the advances of the CytoTape system over the XRI system in both technical and scientific aspects (see the hexagon comparison (Fig. R4) and summarized Table (Table R1)).

Fig. R4 Comparison of XRI and CytoTape system across key performance dimensions. Radar plot illustrating major capabilities of the XRI (green) and CytoTape system (purple). Axes represent recording performance metrics, including (clockwise from top) recording in neurons, recording in dividing cells, recording *in vivo*, timestamps along single fibers, maximum temporal resolution, maximum temporal scale, number of unique recorders, multiplexing capacity *in vitro*

and *in vivo*, multiple recorded brain regions *in vivo*, and maximum number of recorded neurons per mouse. Radial grids denote quantitative or categorical scales for each feature. For “recording in neurons,” “recording in dividing cells,” “recording *in vivo*,” and “timestamps along a single fiber,” a value of 0 indicates no physiological signal recording capability or single-cell time axis, while a value of 1 indicates the presence of such capability. For “multiple recorded brain regions *in vivo*”, a value of 0 indicates that the system lacks this capability, whereas a value of 1 indicates that the capability is present. A value of 0.5 for “recording *in vivo*” denotes the system preserves temporal information but lacks an explicit single-cell time axis. All the XRI results are obtained from *Nat. Biotechnol.* 41, 640–651 (2023).

Table R1 Comparison of XRI and CytoTape in cell culture and *in vivo*

Tech Specs	XRI	CytoTape (this work)
Morphology in cultured neurons	fiber (functional)	fiber (functional)
Morphology in HEK cells	puncta and fiber (not functional)	fiber (functional)
Morphology in HeLa cells	intertwined fiber (not functional)	fiber (functional)
Morphology in cultured glial cells	none	fiber (functional)
Time recovery method in cell culture	time calibration via tamoxifen- induced Cre/FLEX system	timestamps via HaloTag and dye switches
Timestamp precision in cell culture (hour)	24	0.5
Longest recording duration reported in cultured neurons (day)	7	21
Longest recording duration reported in HEK cells (day)	none	5
Longest recording duration reported in HeLa cells (day)	none	4

Number of developed recorders for distinct cell physiological activities	1 (c-fos promoter activity)	7 (c-fos , Arc , Egr1 , and HSPA1A promoter activities; CREB, Npas4 , and Fos protein activities)
Reported number of simultaneously recorded distinct cell physiological activities in cell culture	1	5
Spatial mapping of in vivo transcriptional recordings	none	yes
Multiplexing signals in vivo	none	yes
Time recovery method in in vivo	none	Dox-inducible system (Tet-On)
Maximum neurons recorded in vivo	835	14,123
Recorded regions of the mouse brain	hippocampal CA1	hippocampal CA1, hippocampal DG, and posterior parietal cortex
Biological findings	none	yes

Our view is that, like other technology development efforts in the field of molecular tools for cellular recording, such as **DNA ticker tapes** (*Science* 2016, 353, aag0511; *Science* 2017, 358, 1457-1461; *Science* 2018, 361, 870-875; *Nature* 2022, 608, 98-107; *Nature* 2024, 632, 1073-1081, etc.), **fluorescent calcium indicators** (*Nature* 1997, 388, 882-887; *Nature* 2013, 499, 295-300; *Cell* 2019, 177, 1346-1360.e24; *Nature* 2023, 615, 884-891, etc.), **fluorescent voltage indicators** (*Science* 2011, 333, 345-348; *Science* 2015, 350, 1361-1366; *Cell* 2016, 166, 245-257; *Cell* 2018, 174, 481-496; *Cell* 2019, 179, 1590-1608; *Nature* 2019, 574, 413-417; *Science* 2019, 365, 699-704; *Cell* 2022, 185, 3408-3425; *Cell* 2023, 186, 543-559, etc.), and **fluorescent neurotransmitter indicators** (*Science* 2018, 360, eaat4422; *Cell* 2018, 174, 481-496; *Cell* 2020, 183, 1986-2002; *Science* 2025, 388, eadt7705, etc.), which have evolved through successive generations of engineering refinement to achieve higher sensitivity, stability, and broader applicability, CytoTape extends the capabilities of its predecessor by enabling multi-week, multi-event, and single-cell-resolved transcriptional recording in a spatiotemporally scalable manner in both *in vivo* and *in vitro*. We believe these advances substantially expand

both the scope and the utility of intracellular protein tape recorders beyond what was previously possible with XRI.

From a scientific standpoint, the reported decoupling of pCREB and c-Fos in some neurons is unexpected, but its biology mechanism is addressed. Without a clearly posed question, a testable hypothesis, and experiments designed to verify it, presenting only a descriptive observation is premature for publication and does not represent a significant scientific breakthrough. Importantly, there is also no evidence that this phenomenon is not an artifact of the in-vitro system.

Reply 2: We respectfully note that the pCREB–Fos decoupling observations were from HEK cells. Multiple hypotheses could be attributed to the decoupling of pCREB and Fos in mammalian cells. For example, decoupling could be due to mechanisms from alternative pathway(s) that co-regulates Fos in addition to the CREB pathway, such as insufficient CBP/p300 recruitment despite CREB-Ser133 phosphorylation (*Cell* 1997, 91, 741-752), CRTC (TORC) family co-activator regulation via Ca^{2+} /calcineurin and SIK/PP2A (*Nat. Rev. Mol. Cell. Biol.* 2011, 12, 141-151), ERK–RSK/PP2A–CRTC3 cross-talk (*Current Biol.* 1995, 5, 1191-1200), CaMKII/IV-dependent modulation of CREB signal transduction (*Mol. Cell. Biol.* 2000, 20, 9409-9422), PKC isoform–mediated regulation of CBP (*Biochem. J.* 2014, 458, 469-479), Epac–Rap1–B-Raf–ERK signaling (*Biochim. Biophys. Acta Mol. Cell Res.* 2015, 1853, 539-548), stress kinases (JNK/p38) or PI3K/Akt pathways (*EMBO J.* 1993, 2, 5097-104), failure of ERK-dependent co-activation via Elk-1/SRF at the SRE element in the *c-fos* promoter (*Curr. Biol.* 1995, 5, 1191-1200), mismatched or asynchronous timing between PKA/CREB and ERK/Elk-1 signaling inputs (*Cell* 2008, 134, 624-633), and reduced chromatin accessibility or promoter refractory states at the Fos locus due to prior IEG activation (*Nat. Neurosci.* 2007, 10, 1238–1240). Each of these signaling or transcriptional axes represents a complex, multi-layered regulatory network, and comprehensively testing all these possibilities would require extensive, multi-year mechanistic studies beyond the scope of this technology development work. The focus of this work is to establish and validate the CytoTape platform and demonstrate its technological capability to dissect gene regulation dynamics. We envision that future studies will leverage the new kind of measurement enabled by the CytoTape system, together with targeted perturbations of these pathways, to systematically elucidate the molecular basis of this phenomenon and many other phenomena. We would also like to note that we are currently collaborating with Dr. Denise Cai's lab at Icahn School of Medicine at Mount Sinai and Dr. William Hwang's lab at Harvard Medical School on mechanistic studies of gene regulation dynamics in brain and cancer cells using the CytoTape system.

In light of these many possible mechanisms and the referee's request for a hypothesis-driven analysis, we first validated that the decoupling of pCREB and Fos is not an artifact of the *in vitro* system, and then formulated a biological question, proposed a hypothesis, and performed experiments to test this hypothesis.

Safety test and validation of the decoupling of pCREB and Fos revealed by CytoTape: we first examined the cellular and physiological states of HEK cells expressing CytoTape. As shown in **Fig. R1a**, the majority of HEK cells expressed CytoTape fibers. We performed immunohistochemical characterization and fluorescence-based viability assays and found that CytoTape expression did not alter cellular or physiological state markers (**Fig. R1b-g**), including

Cleaved Caspase-3 (apoptotic marker), GRP78/BiP (endoplasmic reticulum stress marker), Ki-67 (cell proliferation marker), TOMM20 (mitochondrial integrity marker), Ethidium Homodimer-1 (membrane permeability and DNA damage marker), and Calcein-AM (live-cell viability indicator). We then examined the transcriptional activities and signaling of HEK cells expressing CytoTape, using markers including Egr1 (early growth response 1; IEG marker), pElk-1 (ERK-dependent transcription factor activation marker), pCREB (cAMP/PKA-dependent transcription factor activation marker), Fos (IEG marker), and pERK1/2 (ERK/MAPK pathway activation marker). **Fig. R1h-i** shows that CytoTape expression did not alter the transcriptional or signaling activities of HEK cells.

We integrated all CytoTape-recorded signals across cells (**Fig. R2a**) and found that, on average, pCREB activation precedes Fos expression, consistent with the expected upstream role of pCREB in the transcriptional cascade. To validate the observed decoupling between pCREB and Fos, we performed orthogonal single-cell assays, including immunostaining of pCREB and Fos, as well as fluorescence reporter assays using pCREB-EGFP and Fos-mRuby3 constructs. Immunostaining and fluorescence imaging were conducted at 1 hour and 24 hours after forskolin stimulation. As shown in **Fig. R2b-e**, a substantial portion of cells showed pCREB–Fos decoupling, in agreement with the results from CytoTape recording. In addition, we performed two independent control experiments: CytoTape recording without forskolin stimulation, and immunostaining for pCREB and Fos in HEK cells with and without CytoTape expression. These results confirmed that CytoTape expressions do not alter pCREB or Fos activities (**Fig. R2f, Fig. R1j,k**). Together, the decoupling of pCREB and Fos observed by CytoTape is not an artifact of the *in vitro* system. We have added Fig. R1-R2 and associated discussion in the revised maintext.

CytoTape provides new insights into temporal principles of gene regulation dynamics: CREB and Fos are two transcriptional factors involved in signaling-dependent gene-expression programs in mammalian cells (*Annu. Rev. Biochem.* 1999, 68, 821-861). We applied CytoTape to investigate how CREB activation is transduced into Fos induction over time (Fig. 4a-c), a process that informs how cells integrate signaling to control gene expression (*Cell* 2013, 152, 945-956). In HEK cells, FSK-induced CREB activation preceded Fos expression, consistent with the upstream role of CREB in the transcriptional cascade (Extended Data Fig. 5a). We computed time-lagged correlations between the two temporal trajectories, and dimensional reduction of the correlation profiles revealed two distinct cell clusters (Fig. 4d): Type 1 cells (“decoupled mode”) exhibited little or no correlation between CREB and Fos activities at any time lag, whereas Type 2 cells (“coupled mode”) displayed strong positive correlations near zero lag, indicating coordinated CREB–Fos dynamics (Fig. 4e). Correspondingly, Type 2 cells showed a canonical CREB–Fos cascade (Fig. 4g), whereas Type 1 cells showed irregular Fos fluctuations despite comparable CREB activation (Fig. 4f).

We hypothesized that this divergence arises from differences in both transcriptional state and signal integration (*Nat. Commun.* 2016, 7, 12057; *Nat. Commun.* 2017, 8, 255). Supporting this hypothesis, Fos activity fluctuated modestly more in Type 1 cells before CREB activation, suggesting a history of active Fos regulation may place the gene in a refractory state, limiting its response to new CREB activation (Fig. 4h,i). Immunostaining against pCREB and Fos proteins (Extended Data Fig. 5b,c) and fluorescent protein reporter assays (Extended Data Fig. 5d,e) at 1 h and 24 h after FSK stimulation likewise showed heterogeneous and often uncorrelated

pCREB and Fos signals, confirming that CREB phosphorylation alone is insufficient to ensure Fos induction.

To investigate the molecular basis of CREB–Fos decoupling, we asked whether other signaling inputs influence whether CREB phosphorylation is transduced into Fos transcription (**Fig. R5**). Because the Fos promoter contains both the cAMP response element (CRE, regulated by cAMP→PKA→pCREB signaling) and the serum response element (SRE, regulated by MEK→ERK→pElk-1/SRF signaling), we hypothesized that CREB–Fos decoupling arises when ERK signaling is absent or ineffective to integrate inputs (*Mol. Cell. Biol.* 1999, 19, 136–146). Consistent with this hypothesis, CytoTape showed that inhibition of ERK signaling by U0126, a MEK inhibitor, abolished Fos induction under FSK stimulation, whereas activation of ERK signaling by epidermal growth factor (EGF) resulted in markedly strong Fos induction under FSK stimulation (**Fig. R5c,d**). In both conditions, CREB remained robustly activated by FSK. These results were independently validated by protein immunostaining against pCREB and Fos (**Fig. R5e,f**). Compared to FSK alone, FSK+U0126 increased the portion of cells in the decoupled state while FSK+EGF did not decrease this portion (**Fig. R5g,h**), indicating that ERK signaling is necessary but not sufficient for robust Fos induction and that other mechanism(s) beyond ERK signaling may contribute to CREB–Fos decoupling. These results suggest a signal- and state-dependent mechanism in IEG regulation, demonstrating the utility of CytoTape in dissecting temporal cellular signals.

We have added Fig. R1,R2,R5 and discussions in the revised maintext.

Fig. R1 Immunohistochemical and fluorescence-based characterization of cellular and physiological state markers, as well as transcriptional activity and signaling markers, in HEK cells expressing CytoTape. (a) Representative confocal image of HEK cells expressing CytoTape. Majority of HEK cells express CytoTape fibers. HEK cells with or without CytoTape expression were stained with (b-e) antibodies against specific cellular or physiological markers, (f,g) fluorescent chemical dyes, (h-k) antibodies against specific transcriptional activity, and (l) antibodies against signaling markers. CytoTape fibers were labeled with JF₅₈₅ and nuclei were stained with DAPI. Fluorescence intensities were imaged volumetrically using a 10x objective on a spinning disk confocal microscope under identical imaging conditions. (b,e,h-l) Mean fluorescence intensities were quantified in ImageJ as the averaged signal of the fluorescent secondary antibodies over imaged fields of view. Scale bars, 50 μ m. (b-l, top) Representative confocal images of HEK cells transfected with pUC19 (Ctrl) or expressing CytoTape, stained with the indicated antibodies or imaged with the indicated fluorescent dyes. Scale bars, 50 μ m. (b-l, bottom) Scatter plots of the quantifications for each marker between control and CytoTape-

expressing groups (n = 10 fields of view from 2 cultures). Middle line, median; whiskers, interquartile range; black dots, individual data points. n.s., not significant; Mann–Whitney U test. (b) Cleaved Caspase-3 (apoptotic marker). (c) GRP78/BiP (endoplasmic reticulum stress marker). (d) Ki-67 (cell proliferation marker). (e) TOMM20 (mitochondrial integrity marker). (f) Ethidium homodimer-1 (membrane permeability and DNA damage marker). (g) Calcein-AM (live cell marker). (h) Egr1 (Egr1 expression marker). (i) pElk-1 (Elk activity marker). (j) pCREB (CREB activity marker). (k) Fos (Fos expression marker). (l) pERK1/2 (ERK activity marker).

Fig. R2 pCREB and Fos exhibit complex, temporally decoupled dynamics following pCREB activation, as validated by protein immunostaining and fluorescence protein reporter assays across HEK cells. (a) Averaged curve of pCREB and Fos signals relative change from baseline plotted against recovered time after *TransIT-X2* transfection. Thick centerline, mean; darker boundary, s.e.m. Confocal images of pCREB and Fos protein immunostaining in cells fixed at (b) 1 hour and (c) 24 hours after the completion of a 1-hour stimulation with 50 μ M FSK. Confocal images of pCREB and Fos protein reported by EGFP and mRuby3 in cells fixed at (d) 1 hour and (e) 24 hours after the completion of a 1-hour stimulation with 50 μ M FSK. pCREB and Fos intensities in (b-e) were quantified in 30 cells from 2 cultures. Each dot represents a single HEK cell. Scale bar, 50 μ m. (f) Profiles of pCREB and Fos signals intensities along CytoTape in the control group (no stimulation). Thick centerline, mean; darker boundary, s.e.m; thin lines, single traces. pCREB and Fos intensities were quantified in 8 cells from 2 cultures.

Fig. R5 CytoTape resolves pCREB–Fos decoupling under pathway perturbations in single HEK cells. (a) Schematics of the constructs transfected into HEK cells. (b) Experimental timelines (left) and expected JF dyes distributions along CytoTape (right). Averaged curve of pCREB and Fos signal relative change from baseline plotted against recovered time under (c) 50 μ M FSK + 10 μ M U0126 and (d) 50 μ M FSK + 20 μ M EGF stimulation. Thick centerline, mean; darker boundary, s.e.m. Confocal images of pCREB and Fos protein immunostaining in cells fixed at 1 hour (left) and 24 hours (right) after the completion of a 1-hour stimulation with (e) 50 μ M FSK + 10 μ M U0126 stimulation and (f) 50 μ M FSK + 20 μ M EGF. Scale bar, 50 μ m. (g) UMAP plot of time-lagged correlations between pCREB and Fos activities across single cells under 50 μ M FSK ($n = 77$ CytoTapes from 77 cells, 5 cultures), 50 μ M FSK + 10 μ M U0126 ($n = 30$ CytoTapes from 30 cells, 2 cultures), and 50 μ M FSK + 20 μ M EGF ($n = 29$ CytoTapes from 29 cells, 2 cultures). (h) Heatmaps showing time-lagged correlations between pCREB and Fos signals for decoupled (upper) and coupled (lower) HEK cells under 50 μ M FSK (left), 50 μ M FSK + 10 μ M U0126, and 50 μ M FSK + 20 μ M EGF stimulation. x-axis represents a time-lagged shift window of ± 0.4 days. y-axis represents HEK cell numbers. Data of HEK cells under 50 μ M FSK is from Fig. 4.

The newly added in-vivo data in Fig. 7 closely mirror the strategy and presentation of Fig. 4e–f in the published XRI Nature Biotechnology paper, not offering new advances.

Reply 3: We respectfully disagree with the referee’s comment regarding advances of the *in vivo* results enabled by CytoTape, explained below.

CytoTape enables large-scale, multiplexed, single-cell-resolved recording of gene expression dynamics across multiple brain regions over weeks-long timescales, incorporating non-invasive doxycycline-based timestamps for temporal reconstruction for *in vivo* application, which XRI did not provide. Time-resolved recording with single-cell time axis is especially critical to support causal inference for *in vivo* studies, which CytoTape, but not XRI, offers this capability. In the *in vivo* experiments of XRI (Fig. 4 in *Nat. Biotechnol.* 2023, 41, 640–651), the data served only as a proof of concept demonstrating that XRI can preserve a temporal event in the mouse brain under 4-OHT stimulation. In addition, XRI analyzed only a single signal in 835 neurons, limited to hippocampal CA1 region, and did not constitute true temporal recording due to the absence of the time axis (Fig. 4e-f of the XRI paper, the X-axis shown in Fig. 4f of the XRI paper represents the location along the fiber rather than the actual time axis). In contrast, CytoTape-*vivo* enables multiplexed recording of *c-fos*-promoter and doxycycline (Dox)-induced transcriptional dynamics across large neuron populations (8,639 neurons in a single mouse; 30,000+ neurons across 3 mice in total) in the hippocampus-CA1, hippocampus-DG, and posterior parietal cortex (PPC) in the same brain across various time scales (**Fig. R6e-l**). This allows us to use the Dox ON and OFF signals controlled by drinking water as non-invasively introduced timestamps to achieve large-scale, multi-brain-region, spatiotemporally-resolved *c-fos*-promoter recording following KA-induced seizures *in vivo* over 14 days (**Fig. R6j**, 4,541 neurons were analyzed from a single mouse brain) and 18 days (**Fig. R6k**, 14,123 neurons were analyzed from a single mouse brain) of CytoTape-*vivo* expression. We also observed that neurons in the DG showed higher fold changes of *c-fos*-promoter-driven signal monomer level (compared to pre-Dox/KA baseline) than those in CA1 and PPC (**Fig. R6f**). In addition, we developed a robust, user-friendly computational platform for high-throughput segmentation and analysis of fibers from large cell populations across brain regions *in vivo*. Together, these results demonstrate that CytoTape-*vivo* provides substantial new advances beyond the *in vivo* results of XRI. We would also like to note that the number of cells recorded *in vivo* using CytoTape is substantially greater than that recorded by other molecular recording technologies *in vitro*, such as the DNA ticker tape (DNA typewriter, 3,257 cultured cells in *Nature* 2022, 608, 98-107).

To further highlight the advances of CytoTape-*vivo* over the XRI *in vivo* data, we recovered and spatially mapped *c-fos*-promoter signals in the hippocampal CA1 region. As shown in **Fig. R6m**, CytoTape-*vivo* enables recording of *c-fos*-promoter activity across weeks-long timescales and large spatial domains, covering many thousands of neurons. This result demonstrates that CytoTape-*vivo* can continuously record and read out physiological gene expression dynamics through self-assembling protein assemblies across extended temporal and broad spatial scales, with single cell resolution—a capability not achievable with any prior *in vivo* recording technology. We have added Fig. R6m in the revised maintext.

Fig. R6 CytoTape-vivo simultaneously records Dox-dependent and c-fos signal monomer expression histories over weeks in single cells across multiple brain regions *in vivo*. (a) Schematic of the CytoTape-vivo system for multiplexed, scalable recording of cellular signals in living mouse brains. (b) Left, experimental pipeline for *in vivo* application. Middle, low-magnification coronal-view confocal images from a mouse brain expressing CytoTape-vivo. Right, computational segmentation of neuronal soma and CytoTape-vivo fibers from high-magnification DG volumes. (c) Experimental timeline for multiplexing *in vivo*. (d) Left, low-magnification image of a representative brain section. Scale bar, 500 μ m. Middle, 3.2- μ m-thick maximum-intensity projection in DG. Scale bars, 50 μ m. Right, confocal images showing c-fos-promoter-driven and Dox-dependent signal monomer expression in DG neurons. (e) Dox and c-fos signals in posterior parietal cortex (left), hippocampal CA1 (middle), and hippocampal DG (right) (n = 604, 1,835, and 11,205 fibers from 600, 1,697, and 6,342 neurons, respectively; one mouse). (f) c-fos promoter signal amplitudes from (e). ***, P < 0.001; n.s., not significant; Kruskal-Wallis analysis with Dunn's post hoc tests. (g) Experimental timeline for c-fos promoter

signal recording *in vivo*. **(h)** Left, Dox signal change over 14 days in control and experimental mice. Centerline, mean; shaded area, interquartile range (IQR). Right, Dox signal amplitudes change in control (Ctrl; no Dox, no KA) and experimental (Exp; described in **(g)**) mice. ***, $P < 0.001$; n.s., not significant; Kruskal–Wallis analysis with Dunn’s post hoc tests. **(i)** Averaged *c-fos* promoter signals change in control and experimental mice (control: 717 neurons; experimental: 4,514 neurons, both DG, one mouse). ***, $P < 0.001$; n.s., not significant; Kruskal–Wallis analysis with Dunn’s post hoc tests. **(j)** Recorded *c-fos* promoter signal change in DG neurons under KA-induced seizure (described in **(g)**). $n = 4,514$ neurons from one mouse. Dashed line, seizure induction. **(k)** Experimental timeline for *c-fos* promoter signal recording *in vivo*. **(l)** Recorded *c-fos* promoter signal intensity change in CA1 neurons under KA-induced seizure (described in **(k)**). $n = 14,123$ neurons from one mouse. Dashed line, seizure induction. **(m)** Spatiotemporal maps of *c-fos* promoter signals over weeks. Color represents *c-fos* promoter signal intensity from **(l)**. Box plots in **(f,h,i)**: middle line, median; box boundary, interquartile range; whiskers, 10-90 percentile.

Comments on specifics:

- Figure R13 shows only a marginal difference between C-4 and B1; three outliers in B1 drive a small shift, which is unconvincing.

Reply 4: To address this concern, we expanded our analysis to include a larger dataset (51 fibers each for B1 and C4). The updated results show that the difference between B1 and C4 is not driven by outliers (**Fig. R7**). B1 and C4 constructs represent part of intermediate designs that demonstrate reducing the linker length contributes to decreased fiber thickness. In addition, our goal is not only to obtain a thinner structure, but also to build a full molecular recording system.

Fig. R7 Statistical analysis of protein assembly width for B1 and C4 in cultured neurons at day 7. A total of 51 fibers from B1 and 51 fibers from C4 were analyzed. *, $P < 0.05$; Mann–Whitney U test. Middle line in bar plot, median; whiskers, minimum and maximum; black dots, individual data points.

- The revised summary is clearer.

Reply 5: We appreciate the positive feedback.

- Figure R3 does not demonstrate recording beyond day 14 or long-term test.

Reply 6: Following the referee’s suggestion, we performed experiments to show that CytoTape can achieve sub-daily resolution while maintaining weeks-long recording capability (**Fig. R8**). We have added Fig. R8 in the revised maintext.

Fig. R8 CytoTape offers sub-daily temporal resolution over weeks-long recordings in cultured neurons. (a) Schematic of the constructs transfected into cultured neurons. (b) Time points of JF₅₈₅, JF₆₃₅, and JF₅₀₃ addition and cell fixation (left panel) and expected dye distributions along the protein fiber (right panel). (c) Left, confocal images of timestamped fibers labeled with different dye-switching intervals for long-term recording. Middle, enlarged views of dye switches along the CytoTape fiber. Right, fluorescence line profiles from the experiments described in (b) (n = 10 CytoTapes from 10 neurons, 2 cultures). Dashed lines in (c) indicate the transient dye switches. Thick centerline, mean; darker boundary, s.e.m.; lighter boundary, s.d.; light gray thin lines, data from individual CytoTapes.

- The authors claim CytoTape grows only horizontally, while XRI expands in all directions. Yet, in their published Nature Biotechnology paper (Fig. 1), they showed that XRI also grows horizontally, whereas iPOK(293Y) grows in all directions. The rationale behind these versions is essentially the same, and the progress is incremental: iPOK(293Y) is version 1 (First paper, Dr. Levy group, <https://www.nature.com/articles/nature23320>), XRI is version 2 (the authors’ Nat. Biotechnology), and CytoTape is version 3 (or maybe 2.5). These similar iterations with different names make it difficult for readers to trace the previously published work.

Reply 7: We respectfully clarify below. 1POK(E239Y) exhibits expansion in all directions in neurons. In the case of XRI, the use of an MBP fusion reduced lateral growth within the first 7 days in neurons; however, lateral expansion became significant after 7 days, and XRI cannot form stable structures in dividing cells. In contrast, CytoTape was structurally redesigned to eliminate significant lateral growth for 21 days and to function robustly in cultured neurons, dividing cells, and *in vivo*. Therefore, CytoTape structural monomer is not only optimized to suppress lateral fiber growth for extended recording duration in neurons, but is also engineered for flexibility, compatibility with diverse dividing cells, and long-term recording.

We also clarify that the rationale behind 1POK(E239Y), XRI, and CytoTape is fundamentally different and the advances are substantial. **Scientific purpose of Dr. Levy’s paper:** 1POK(E239Y), which named by PDB ID (<https://www.rcsb.org/structure/1POK>), was used to study protein–protein interactions within protein assemblies, aiming to identify the amino acids required for assembly formation rather than to perform physiological recording in live cells. Furthermore, 1POK(E293Y) cannot robustly assemble into regular fibers in mammalian cells and cannot record, as shown in Fig. 1 of the XRI paper. Therefore, 1POK(E239Y) does not

belong to the protein ticker tape recording system and cannot be considered as version 1, as its concept and application are fundamentally different.

Rationale and progress of 1POK(E239Y), XRI, and CytoTape: XRI uses the addition of MBP to reduce lateral growth within 7 days to enable recording (otherwise 1POK(293Y) cannot record, as shown in Fig. 1 of the XRI paper). However, CytoTape employs a computationally assisted, multi-front rational design not only to reduce lateral growth over extended durations, but also to increase fiber flexibility and enable single-cell, long-term recording across multiple cell types both *in vitro* and *in vivo*. For the underlying mechanism, the increased conformational stability of the CytoTape monomer may have contributed to the formation of thinner protein fibers, as limiting the protein conformational fluctuations and transitions could in principle minimize unintended binding surfaces and favor only the longitudinal binding surfaces by design. This low free-energy landscape of CytoTape also makes it less susceptible to perturbations from the surrounding microenvironment, supporting its ability to maintain structural integrity and compatibility across cell types. Thus, the design rationale, underlying mechanism, recording concept, and overall performance of CytoTape differ fundamentally from those of XRI (**Fig. R4, Table R1**).

In addition, monomer design is one part of our comprehensive efforts presented in this work to achieve difference-of-a-kind progress in molecular recording. Compared to the XRI system, we have comprehensively redesigned and developed it to substantial advances in multiple scientific and technical aspects, including *in vivo* recording, mapping and multiplexing at large spatiotemporal scales (Fig. 5), stabilized protein monomer structure and conformational dynamics (Extended Data Fig. 1), flexible fiber morphology (Fig. 1), timestamp strategy and time axis recovery (cell culture and *in vivo*) (Fig. 2 and 5), seven transcriptional recorders (Extended Data Fig. 4), minutes-scale temporal resolution and multi-weeks-long temporal scalability (Fig. 2), single-cell spatial resolution and multi-brain-region scalability (Fig. 5), recording in diverse dividing and non-dividing cell types (Fig. 3), 8- color, 5-signal multiplexing capability per fiber (Fig. 3), and biological findings on transcriptional regulations (Fig. 4). Therefore, the CytoTape system represents a fundamentally different form of progress compared to the XRI system and therefore cannot be simply regarded as versions 2 and 3.

Name choice of CytoTape: The name *eXpression Recording Islands* (XRI) originated from the fact that XRI structures are “island-like” in live cells (shorter and less flexible than CytoTape fibers), inspired by our earlier work on *Signaling Reporter Islands* (“SiRI”; *Cell* 2020, 183, 1682-1698) that creates puncta-like structures in live cells for multiplexed live imaging of fluorescent biosensors/reporters. However, the term “islands” no longer precisely describes the morphology of the new CytoTape fibers, which are long, thin, and flexible, best described as a “tape” or a “thread”. Because “tape” more aligns with our goal of building a continuous recording system, as in the terms “tape recorders” and “ticker tapes”, we decided to use “tape” for the name of the technology. Moreover, because XRI cannot function properly in dividing cells, while CytoTape can (as confirmed in HEK, HeLa, and glial cells), we therefore believe using the word “Cyto” could further clarify the fact that CytoTape can function across diverse cell types. We understand the potential risk of such a new name, as clearly pointed out by the referee, that it might make the technology development history harder to track (we hope this can be largely mitigated by the revised texts in the new Abstract and Introduction). Nevertheless, we also recognize the risk of confusion by still naming the thread/tape-like structure as “islands”, since it

does not precisely describe the distinct morphological features of the new structure. Furthermore, we envision the thread/tape-like structures represent a promising (and perhaps critical) future direction in this field, due to their many advantages that break through the limitations of short, thick, or rigid structures as we experimentally observed in this work. Therefore, we also hope the name CytoTape could encourage future technology development efforts in the field towards thread/tape-like recorders. Essentially, the choice between “XRI2” and “CytoTape” for the name represents whether one would like to emphasize past efforts or to encourage future efforts. Because protein tape recording is a new field with, we believe, large future potential to achieve multiplexed recording across large spatial and temporal scales for new kinds of biology and medicine research, we think choosing the latter name would best benefit the health of this field of technology development and encourage future efforts.

- The decoupling of pCREB and c-Fos remains unexpected. More studies are needed to determine whether it is naturally occurring or induced by the artificial system, and question/hypothesis driven biology.

Reply 8: Please see our detailed responses of validation, biological question, hypothesis, hypothesis experiments, and validation in **Reply 2**.

- Only systems with long-term, sub-daily resolution, and no functional influence on neurons and circuits are suitable for behavioral studies such as memory tracing. Functional validation in neurons/circuitry is needed before publishing.

Reply 9: Immunohistochemical characterization of molecular and cellular physiological markers, electrophysiological analysis of neuronal physiology and synaptic transmission, calcium imaging-based analysis of spontaneous neuronal network dynamics, and animal behavioral analyses, including open field test, novel object recognition, and contextual fear conditioning, demonstrated that the CytoTape system does not affect neuronal or circuit function *in vitro* and *in vivo*. In addition, 1POK based protein assembly has been reported to be inert in live cells without altering proteomics (*Mol. Syst. Biol.* 2025, 21,1306–1324).

Immunohistochemical characterization *in vivo*: We performed immunohistochemical characterization of mouse brains expressing CytoTape (**Fig. R9**) and found CytoTape-*in vivo* expression in cell populations *in vivo* does not alter cellular and synaptic physiology markers, including NeuN as a neuronal marker, cleaved Caspase-3 as an apoptotic marker, GFAP as an astrocyte marker, Synaptophysin as a synaptic protein marker, Iba1 as a microglial marker, γ H2AX as a DNA damage marker, Hsp70 and Hsp27 as cell physiological stress markers, and *c-fos* protein as a cellular activity marker.

Electrophysiology, synaptic transmission, calcium activity, and neural network dynamics in primary neuron cultures: We analyzed the electrophysiological properties, synaptic transmission, spontaneous calcium activity, and neural network dynamics of cultured neurons expressing CytoTape and found that CytoTape expression does not alter these physiological processes (**Fig. R10**).

Contextual fear conditioning test *in vivo*: we found weeks-long CytoTape expression across hippocampi in both hemispheres retains normal contextual fear learning, as the freezing rate during contextual fear recall had no significant difference between control and CytoTape-*in vivo* mouse groups (**Fig. R11b,c**).

Open field test *in vivo*: No significant differences were detected in total traveling distance, mean speed, time spent in the center, distance traveled in the center, and average speed in the center between control and CytoTape-*vivo* mouse groups, indicating weeks-long CytoTape expression across hippocampi in both hemispheres retains normal locomotor functions (**Fig. R11d-j**).

Novel objective recognition test *in vivo*: Mice expressing CytoTape-*vivo* exhibited typical exploration patterns during both the familiarization and test phases, with no significant differences in total exploration time or discrimination index, indicating weeks-long CytoTape expression across hippocampi in both hemispheres retains normal memory functions (**Fig. R11k-n**).

Thus, these orthogonal and comprehensive assays demonstrate that the CytoTape system does not exert functional influence on neurons or circuits. We have added Fig. R9-R11 and discussion in the revised maintext.

Fig. R9 Immunohistochemical characterization of cellular and synaptic physiology markers in mouse brains expressing CytoTape-vivo. (a) Representative confocal images of brain slices from adult mice expressing CytoTape-vivo in the hippocampus in the right cerebral hemisphere following AAV injection. 3-month-old mice were injected with AAV9-CytoTape-vivo-HA at the CA1 region in the hippocampus of the right cerebral hemisphere. When the mice reached 14 days after AAV injection, they were perfused with 4% PFA, and brains were sliced coronally at 40 μ m in 1X PBS, and stained with antibodies against one of the cellular and synaptic markers below (see b-j) and against HA tag to label CytoTape-vivo fibers, together with DAPI to label nuclei. Staining intensities of cellular and synaptic markers in the hippocampus were imaged volumetrically using a 40x objective on a spinning disk confocal microscope, with identical imaging conditions, measured in ImageJ as the averaged fluorescent intensities of the fluorescent secondary antibodies over imaged fields of view (400 μ m \times 400 μ m \times 40 μ m for each fields of view), and compared between the left hemisphere and the right hemisphere.

Scale bars, 500 μm . **(b-j, top)** Representative confocal images of the CA1 region in the non-injected hemisphere (Ctrl) and CytoTape-vivo-injected hemisphere (CytoTape-vivo) in the brain slices stained with antibodies against each of the cellular and synaptic markers indicated, and DAPI. Scale bars, 50 μm . **(b-j, bottom)** Box plots of the quantifications for each of the cellular and synaptic markers between the non-injected hemisphere (Ctrl) and CytoTape-vivo-injected hemisphere (CytoTape-vivo); for each marker in each hemisphere, $n = 10$ fields of view (FOVs) from 3 mice. Middle line in box plot, median; box boundary, interquartile range; whiskers, minimum and maximum; black dots, individual data points. n.s., not significant; Wilcoxon signed-rank test. **(b)** NeuN (a neuronal marker). **(c)** Cleaved Caspase-3 (an apoptotic marker). **(d)** GFAP (an astrocyte marker). **(e)** Synaptophysin (a synaptic protein marker). **(f)** Iba1 (a microglial marker). **(g)** γH2AX (a DNA damage marker). **(h,i)** Hsp70 and Hsp27 (cell physiological stress markers). **(j)** *c-fos* protein (a cellular activity maker).

Fig. R10 Electrophysiological properties and calcium dynamics of cultured neurons expressing CytoTape. Scatter plots of the electrophysiological properties in cultured neurons with or without CytoTape expression, in terms of (a) resting potential, (b) membrane capacitance, (c) membrane resistance, (d) holding current at -65 mV, and (e) action potential amplitude. Middle line, median; error bar, interquartile range; gray dots, individual data points. Cultured neurons in the CytoTape group were transduced with UBC-CytoTape-P2A-mEGFP on day in vitro 7 (DIV 7) and assessed by whole-cell patch clamp on DIV 15-18 alongside neurons transduced with CAG-mEGFP (Ctrl group). Neurons expressing CytoTape were identified by GFP before whole-cell recording. $n = 9$ neurons for the control group; $n = 10$ neurons for the CytoTape group. n.s., not significant; Mann-Whitney U test. Scatter plots of synaptic transmission properties, including (f) EPSC amplitude and (g) EPSC inter-event interval, in cultured neurons with or without CytoTape expression. Middle line, mean; error bar, interquartile range; gray dots, individual data points. Experimental group, cultured neurons transduced with CytoTape; Control group, cultured neurons without transduction. f: $n = 8$ neurons for the control

group; n = 8 neurons for the CytoTape group. **g**: n = 8 neurons for the control group; n = 7 neurons for the CytoTape group. n.s., not significant; Mann–Whitney U test. Single traces of calcium dynamics of neurons (**h**) with or (**i**) without expressing CytoTape. Scatter plots of calcium imaging–based network activity metrics in control and CytoTape-expressing cultured neurons, including (**j**) amplitude of calcium spike, (**k**) frequency of calcium spikes, (**l**) synchronous firing rate, and (**m**) coefficient of variation. Middle line, median; error bars, interquartile range; dots, individual neurons or fields of view; n.s., not significant; Mann–Whitney U test.

Fig. R11 Behavioral assays of mice with and without CytoTape-vivo expression, under contextual fear conditioning, open field test, and novel object recognition. (a) Representative confocal images of brain slices from adult mice expressing CytoTape-vivo over weeks in the hippocampi of both the left and the right cerebral hemispheres following AAV injection. Scale bars, 500 μ m. **(b)** Experimental pipeline of contextual fear conditioning experiment. **(c)** Freezing rate during contextual fear recall (n = 8 mice per group). n.s., not significant; Mann–Whitney U test. Middle line, median; whiskers, interquartile range; black dots, data from individual mice. **(d,e)** Representative movement traces of mice with or without our CytoTape-vivo expression in the open-field arena. Quantification of locomotor parameters,

including **(f)** total travel distance, **(g)** mean speed, **(h)** time spent in the center, **(i)** distance traveled in the center, and **(j)** average speed in the center, comparing the control and CytoTape-vivo groups (n = 10 mice per group). n.s., not significant; Mann–Whitney U test. Middle line, median; whiskers, interquartile range; black dots, data from individual mice. **(k)** Representative exploration heatmaps of mice with or without CytoTape-vivo expression during the familiarization (left) and test (right) phases of the novel object recognition task. Dashed outlines indicate predefined object zones. Quantification of novel object recognition performance, including **(l)** percentage of total exploration time during familiarization of object 1 (FO1) and object 2 (FO2), **(m)** percentage of exploration time for familiar (FO1) and novel (NO) objects during the test phase, and **(n)** discrimination index comparing between the control and CytoTape-vivo groups (n = 10 mice per group). In **l**, n.s., not significant; Wilcoxon signed-rank test. In **m**, *, P < 0.05; **, P < 0.01; Wilcoxon signed-rank test. In **n**, n.s., not significant; Mann–Whitney U test. Middle line in bar plots, median; whiskers, interquartile range; black dots, data from individual mice.

Referee #3 (Remarks to the Author):

I co-reviewed this manuscript with one of the reviewers who provided the listed reports.

Referee #3 (Remarks on code availability):

I co-reviewed this manuscript with one of the reviewers who provided the listed reports.

Reply: We thank Referee #3 for co-reviewing this manuscript.

Referee #4 (Remarks to the Author):

The authors addressed my comments satisfactorily. Reading the other reviewers' remarks about precedents, I agree with them that the manuscript gives the sense of a new method, e.g., in the abstract. It's impressive work, even as an "upgrade" of an existing method, and it may be better to explain the gap the previous method left to be filled here.

Reply: We thank the referee for their positive assessment and for finding our revisions satisfactory and the work impressive. We agree with the referee on the importance of clearly stating the technological lineage in the paper. Our rationale was to use a name that accurately describes the current technology while also encouraging future technology development in the field, as described below.

The name *eXpression Recording Islands* (XRI) originated from the fact that XRI structures are "island-like" in live cells (shorter and less flexible than CytoTape fibers), inspired by our earlier work on *Signaling Reporter Islands* ("SiRI"; *Cell* 2020, 183, 1682-1698) that creates puncta-like structures in live cells for multiplexed live imaging of fluorescent biosensors/reporters. However, the term "islands" no longer precisely describes the morphology of the new CytoTape fibers, which are long, thin, and flexible, best described as a "tape" or a "thread". Because "tape" more aligns with our goal of building a continuous recording system, as in the terms "tape recorders" and "ticker tapes", we decided to use "tape" for the name of the technology. Moreover, because XRI cannot function properly in dividing cells, while CytoTape can (as confirmed in HEK, HeLa, and glial cells), we therefore believe using the word "Cyto" could further clarify the fact that CytoTape can function across diverse cell types. We understand the potential risk of such a new name, as clearly pointed out by the referee, that it might make the technology development history harder to track (we hope this can be largely mitigated by the revised texts in the new Abstract and Introduction, copied below, following the referee's suggestion). Nevertheless, we also recognize the risk of confusion by still naming the thread/tape-like structure as "islands", since it does not precisely describe the distinct morphological features of the new structure. Furthermore, we envision the thread/tape-like structures represent a promising (and perhaps critical) future direction in this field, due to their many advantages that break through the limitations of short, thick, or rigid structures as we experimentally observed in this work. Therefore, we also hope the name CytoTape could encourage future technology development efforts in the field towards thread/tape-like recorders. Essentially, the choice between "XRI2" and "CytoTape" for the name represents whether one would like to emphasize past efforts or to encourage future efforts. Because protein tape recording is a new field with, we believe, large future potential to achieve multiplexed recording across large spatial and temporal scales for new kinds of biology and medicine research, we think choosing the latter name would best benefit the health of this field of technology development and encourage future efforts.

We have added a clear description of the relationship between XRI and CytoTape in the Abstract, as well as the technical gap between previous protein ticker tape methods (XRI and iPAK4) in the Introduction of the revised maintext. We respectfully note that although we were unable to include extensive details in the maintext due to the word limit required by *Nature's* formatting guidelines, the detailed technological lineage between CytoTape and XRI is discussed in the Supplementary Information.

Abstract: “Gene expression is dynamically regulated by gene regulatory networks comprising multiple regulatory components to mediate cellular functions. An ideal tool for analyzing these processes would track multiple-component dynamics with both spatiotemporal resolution and scalability within the same cells, a capability not yet achieved. Here, we present CytoTape, a genetically encoded, modular protein tape recorder for multiplexed and spatiotemporally scalable recording of gene regulation dynamics for up to three weeks, physiologically compatible, with single-cell, minutes-scale resolution. **CytoTape employs a flexible, thread-like, elongating intracellular protein self-assembly engineered via computationally assisted rational design, built on earlier XRI technology.** We demonstrated its utility across multiple mammalian cell types, achieving simultaneous continuous recording of five transcription factor activities and gene transcriptional activities. CytoTape reveals that divergent transcriptional trajectories correlate with transcriptional history and signal integration, and that distinct immediate early genes (IEGs) exhibit complex temporal correlations within single cells. We further extended CytoTape into CytoTape-vivo for scalable, spatiotemporally resolved single-cell recording in the living brain, enabling simultaneous weeks-long recording of doxycycline- and IEG promoter-dependent gene expression histories across up to 14,123 neurons spanning multiple brain regions per mouse. Together, the CytoTape toolkit establishes a versatile platform for scalable and multiplexed analysis of cell physiological processes *in vitro* and *in vivo*.”

Introduction: “... Thus, a scalable technology enabling long-term, multiplexed measurements of cellular activities with spatiotemporal resolution is critically needed. The recently proposed “protein tape recorder” concept (XRI and iPAK4) allows recording of cellular activities along intracellular linear protein self-assemblies for post-fixation scalable readout. **However, these protein tape recorders do not support simultaneous recording of multiple cellular signals or time-resolved recording over weeks, because long-term growth of linear protein assemblies is constrained by cell size.**

Here, we introduce CytoTape, an intracellular protein tape recorder for scalable and multiplexed continuous recording with minute-scale temporal precision and weeks-long duration, built on the earlier XRI system. CytoTape employs flexible, thread-like, physiologically compatible protein self-assembly that can grow longer than the cell size with multiple molecular tags, each encoding a distinct cellular signal, to achieve long-term, multiplexed recording. ...”